# COMPASS for rapid combinatorial optimization of biochemical pathways based on artificial transcription factors

Gita Naseri [1,2], Jessica Behrend[1], Lisa Rieper[1] & Bernd Mueller-Roeber [2,3,4]

Balanced expression of multiple genes is central for establishing new biosynthetic pathways or multiprotein cellular complexes. Methods for efficient combinatorial assembly of regulatory sequences (promoters) and protein coding sequences are therefore highly wanted. Here, we report a high-throughput cloning method, called COMPASS for COMbinatorial Pathway ASSembly, for the balanced expression of multiple genes in *Saccharomyces cerevisiae*. COMPASS employs orthogonal, plant-derived artificial transcription factors (ATFs) and homologous recombination-based cloning for the generation of thousands of individual DNA constructs in parallel. The method relies on a positive selection of correctly assembled pathway variants from both, in vivo and in vitro cloning procedures. To decrease the turn-around time in genomic engineering, COMPASS is equipped with multi-locus CRISPR/Cas9-mediated modification capacity. We demonstrate the application of COMPASS by generating cell libraries producing β-carotene and co-producing β-ionone and biosensor-responsive naringenin. COMPASS will have many applications in synthetic biology projects that require gene expression balancing.

[1] University of Potsdam, Cell2Fab Research Unit, Karl-Liebknecht-Str. 24-25, 14476 Potsdam, Germany. [2] University of Potsdam, Department Molecular Biology, Karl-Liebknecht-Str. 24-25, House 20, 14476 Potsdam, Germany. [3] Max-Planck Institute of Molecular Plant Physiology, Plant Signalling Group, Am Mühlenberg 1, D-14476 Potsdam-Golm, Germany. [4] Center of Plant Systems Biology and Biotechnology (CPSBB), Department Plant Development, Ruski Blvd. 139, 4000 Plovdiv, Bulgaria. Correspondence and requests for materials should be addressed to B.M.-R. (email: bmr@uni-potsdam.de)

The yeast *Saccharomyces cerevisiae* is a widely used microorganism for the production of high-value chemicals[1]. Generating an optimal microbial cell factory can be affected by several factors such as gene sequences[2,3], the strength of gene expression regulators[4], the host's genetic background[5], and the expression system[4,6,7]. For the expression of heterologous enzymes, regulation at the transcriptional level is often critical[8,9]. Several types of orthogonal artificial transcription factors (ATFs) have therefore been developed recently[10–12], including our library of 106 inducible, plant-derived ATFs of varying strengths[4].

Importantly, high-level expression of pathway genes often increases metabolic burden[13–18]. To overcome this, methods for balancing metabolic flux have been developed for *Escherichia coli*, including pORTMAGE[19] and ePathOptimize[20], and *S. cerevisiae*, including CRISPR-AID[21] and VEGAS[22]. However, combinatorial optimization approaches typically rely on the constitutive expression of pathway genes[22,23] likely affecting metabolic performance of the cell[24], while inducible ATFs allow a conditional expression of genes[4,18,25].

Several methods for combinatorial assembly have been developed, including Golden Gate (GG) cloning[26] and methods derived thereof, including MoClo[26] and VEGAS[22]. GG relies on type IIs restriction enzyme (RE)-based DNA assembly. The most significant limitation of RE-based assembly methods is their sequence dependency, as RE IIs recognition sites must be eliminated from all DNA sequences to be assembled ("DNA domestication"). Moreover, GG-based methods are entirely based on construction steps performed in *E. coli*, which often is not straightforward, in particular for assemblies of genes of long pathways; amplification of large plasmids is not easily done in *E. coli*[27].

Here, we present COMPASS, a unique, high-throughput method for both, combinatorial gene assembly and pathway optimization; it employs inexpensive in vivo and in vitro overlap-based cloning methods to build large DNA constructs while at the same time eliminating unwanted scar sequences. COMPASS employs the isopropyl β-D-1-thiogalactopyranoside (IPTG)-inducible *GAL1* promoter for the control of plant-derived regulators[4,25], which then control the expression of pathway genes. COMPASS makes use of positive selection protocols, which severely reduces the need for checking individual constructs and strongly improves the efficiency of detecting correct assemblies in cloning reactions[28].

Plasmid-based systems are commonly used for pathway engineering (i.e., ePathOptimize[20] and VEGAS[22]), although they often lack sufficient robustness due to segregational and structural instability[29]. Therefore, we adopted COMPASS for both, plasmid- and genomic integration-based pathway assemblies.

To establish complex construct libraries leading to high levels of metabolic product output, the level of the wanted product must be screenable. The diversity of a library can, e.g., be assessed by screening individual colonies for the production of colored products[22], such as β-carotene[22,30]. However, most chemicals are uncolored and their detection therefore requires alternative methods. Several high-throughput screening methods have recently been developed for the detection of chemicals, e.g., a biosensor for the detection of naringenin (NG) in single-yeast cells[31], which we employ here within the framework of our construction method. Collectively, COMPASS implements multiple features as a high-throughput combinatorial cloning tool for yeast synthetic biology applications.

## Results

### General outline of the combinatorial COMPASS strategy.
COMPASS allows the rapid combinatorial assembly of up to ten pathway genes, each transcriptionally controlled by nine inducible ATF/binding site (BS) units (in short: "plant regulators") differing in expression strength. In COMPASS, construct libraries are built in three successive cloning levels (Fig. 1). At Level 0, (i) ATF/BS units and (ii) CDS units (consisting of the enzyme coding sequence (CDS), a yeast terminator, and an *E. coli* promoter of a selection marker gene) are constructed in 1 week. Level 1 serves the combinatorial assembly of ATF/BS units upstream of up to ten CDS units to generate complete ATF/BS-CDS modules, in a further week. At Level 2, up to five ATF/BS-CDS modules are combinatorially assembled into a single vector in 4 weeks. At Level 1 and 2, successful assemblies are selected by plating cells on appropriate selection media. Levels 0–2 allow multiple parallel assemblies. Modules in Levels 1 and 2 can be integrated into the genome to generate stable yeast strains, facilitated by CRISPR/Cas9-mediated modification that allows one-step integration of multiple groups of cassettes into multiple loci[25]. COMPASS thus provides advantages over alternative combinatorial cloning and optimization methods (Supplementary Data 1). Details of our approach are described in the following.

### Plant-derived ATF/BS units.
To construct orthogonal plant-based transcription regulatory units, we selected nine ATF/BS combinations from our previously reported library of 106 genome-integrated ATF/BSs to cover weak (NLS-GAL4AD-RAV1/4×, NLS-DBD$_{JUB1}$-GAL4AD/2×, and ANAC102-NLS-VP64AD/4×; 300–700 arbitrary units (AU), determined using yEGFP as reporter[4]), medium (NLS-GAL4AD-GRF7/4×, NLS-GAL4AD-ANAC102/4×, and NLS-JUB1-EDLLAD-EDLLAD/4×; 1100–1900 AU) and strong (NLS-GAL4AD-ATAF1/2×, NLS-ATAF1-GAL4AD/2×, and NLS-JUB1-GAL4AD/2×; 2500–4000 AU) transcriptional outputs. The ATFs span an expressional activity ranging from ~0.4- to ~5-fold of that observed for the *TDH3* promoter[32], whereby the regulators were all integrated into the *ura3-52* locus of yeast[4] (Supplementary Fig. 1). Here, we tested whether the genomic locus into which the nine regulators are integrated affects their transcriptional activity. To this end, we placed them into a second locus, *XII-5*, and quantified the transcriptional output and noise level. Expression of yEGFP under the control of the constitutive yeast *TDH3* promoter was used as control. The expressional activity of the ATFs ranged from ~0.05- to ~4.3-fold of that of the *TDH3* promoter[32]. In support of Chen and Zhang[7], the genomic position affected the transcriptional output of the plant regulators. However, the expressional effect exerted by the different regulators was well preserved. The strongest regulators always led to the highest expression mean of the reporter protein, regardless of where the regulator was placed in the genome (Supplementary Note 1, Supplementary Fig. 2, and Supplementary Data 2).

### COMPASS vectors.
To optimize pathway generation using COMPASS, we designed Entry vector X as well as Acceptor and Destination vectors (Supplementary Note 2 and Supplementary Data 3). Entry vector X (Supplementary Fig. 3) is used to assemble (i) ATF/BS units and (ii) CDS units. Acceptor and Destination vectors are designed in two Sets and are used to assemble (i) the library of ATF/BS units upstream of CDS units at Level 1 and (ii) the library of ATF/BS-CDS modules at Level 2. Set 1 includes Destination vector I (Supplementary Fig. 4a) and Acceptor vectors A–D (Supplementary Fig. 4b); while Set 2 includes Destination vector II (Supplementary Fig. 5a) and Acceptor vectors E–H (Supplementary Fig. 5b). Destination vector I can be integrated into the *URA3* or *LYS2* locus of yeast, while Destination vector II can be integrated into the *ADE2.a*

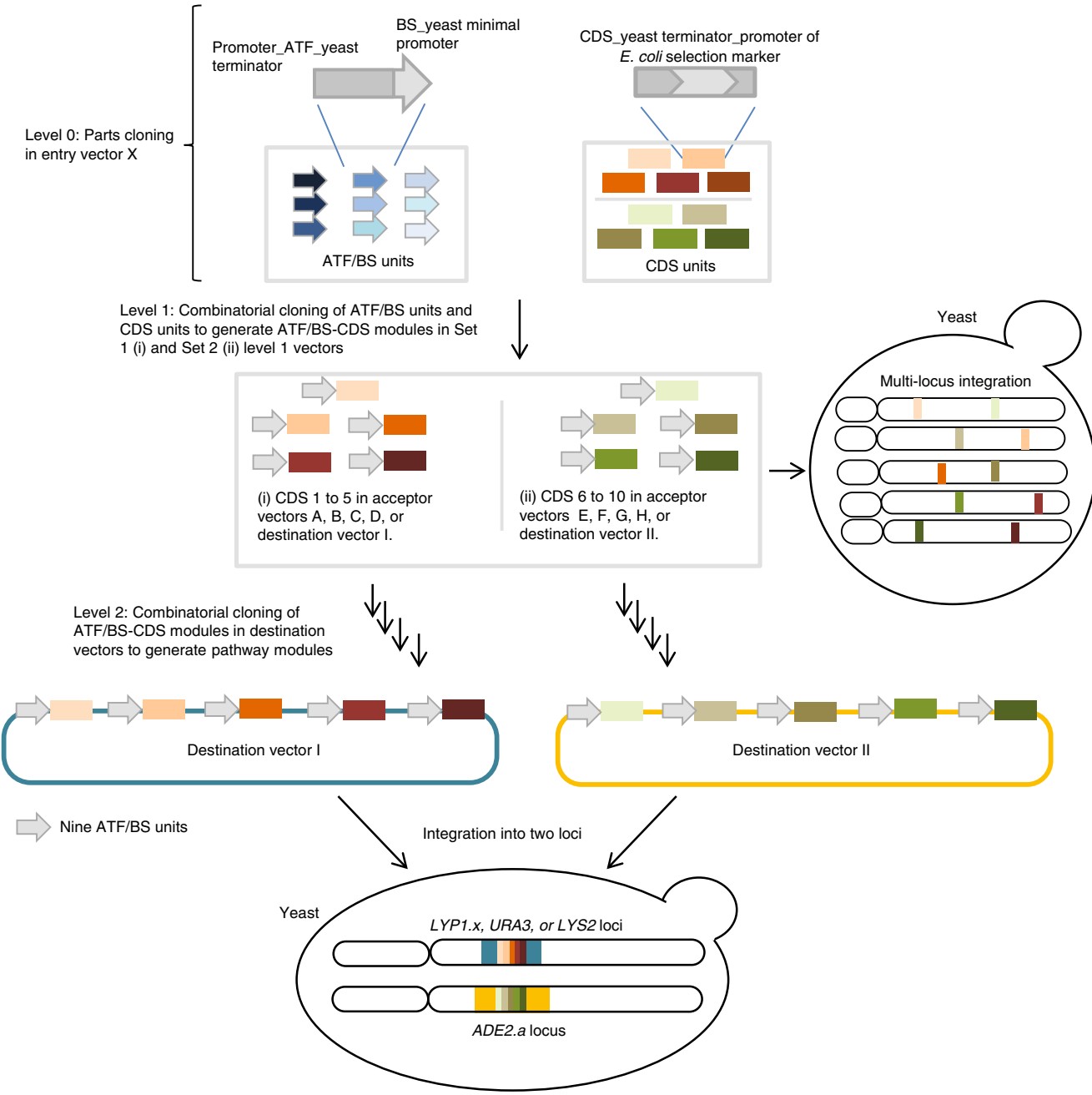

**Fig. 1** The COMPASS workflow. The workflow encompasses three assembly levels. At Level 0, nine ATF/BS units are assembled into Entry vector X. Each ATF/BS unit harbors an inducible promoter to drive ATF expression, the ATF CDS, a yeast terminator, and promoter fragments containing one or more copy(s) of the ATF's BS within a minimal *CYC1* promoter. In separate reactions, each of the ten CDSs is combined with a yeast terminator and the promoter of an *E. coli* selection marker (which defines the vector of the next assembly level) into Entry vector X to generate CDS units. At Level 1, two parallel combinatorial clonings are performed: (i) The nine different ATF/BS units (gray arrows), five CDS units (colored rectangles), and a set of linearized auxotrophic marker vectors (Acceptor vectors A–D, and Destination vector I) are subjected to overlap-based cloning in a single reaction tube (using NEBuilder HiFi). (ii) Similarly, nine ATF/BSs, five other CDSs, and a set of dominant-marker vectors (Acceptor vectors E–H, and Destination vector II) are combined. Using this approach, COMPASS generates ten groups of vectors, each containing nine plasmids (90 plasmids in total). Thereafter, the ten groups of ATF/BS-CDS modules of Level 1 may be integrated into ten defined loci of the yeast genome. At Level 2, ten libraries of ATF/BS-CDS modules are established in Destination vector I and Destination vector II (five libraries in each vector). The Destination vectors I are integrated into the *LYS2*, *URA3*, (or *LYP1.x*) loci, while Destination vectors II are integrated into the *ADE2.a* locus

locus[33]. In addition, Destination vector I.1 (Supplementary Fig. 6) allows integration into the *LYP1.x* locus[33].

**Level 0 of library construction.** For the construction of the ATF/BS library, we employed combinatorial cloning (Supplementary Methods) which established the nine combinations of ATFs and

BSs in Entry vector X. For example, to build NLS-DBD$_{JUB1}$-GAL4AD/2×, NLS-JUB1-EDLLAD-EDLLAD/4×, and NLS-JUB1-GAL4AD/2×-ATF units, we triplex-polymerase chain reaction (PCR)-amplified the *Pro$_{mGAL1-LacI}$-JUB1*-derived ATF fragments (from expression plasmids[4]) and duplex-PCR-amplified *Pro$_{CYC1}$* containing two and four copies of the BS

fragments (from reporter plasmids[4]). The primers include homology regions (HRs) allowing overlap-based recombinational cloning (Supplementary Methods). Overlap-based recombination theoretically leads to six different ATF/BS combinations (Fig. 2a), of which three (numbers 1–3) were needed here. Colony-PCR (Fig. 2b), followed by sequencing, allowed identifying the respective ATF-BSs combinations.

To build units of pathway genes, the CDS, a yeast terminator and a promoter of an *E. coli* selection marker are assembled in *Pac*I-digested Entry vector X (Fig. 2c, Supplementary Methods and Supplementary Data 4). Introducing rare RE cleavage sites in the CDS units (see the protocol for primer design in Supplementary Methods) allows replacing the CDS-terminators with other CDS-terminators in COMPASS vectors at a later stage, if needed.

**Level 1 for construction of ATF/BS-CDS modules**. To construct the library of the ATF/BS-CDS modules (Fig. 1, Fig. 3a), (i) nine ATF/BS units are combinatorially assembled upstream of five CDS units in Set 1 vectors and (ii) five other CDS units are assembled in Set 2 vectors. To conduct five clonings in Set 1 vectors simultaneously (Fig. 3b), equal amounts of the nine ATF/BS units, five freely selected CDS units, and linearized Set 1 vectors are mixed in a single reaction tube for in vitro overlap-based cloning. By providing the missing promoter of *E. coli* selection markers within the assembly units, successful assemblies are identified by plating *E. coli* cells transformed with the reaction cocktail on media containing antibiotics. This creates a complexity of $9 \times 5 = 45$ different plasmids in a successful experiment. Step-by-step protocols and information about primer design are given in Supplementary Methods. Using the same strategy, five other CDS units can be assembled with the nine ATF/BS units in Set 2 vectors.

Moreover, CDS units (without the upstream ATF/BS units) can be generated through combinatorial assembly of five CDSs, five yeast terminators and promoters of five antibiotic resistance genes in either Set 1 or Set 2 vectors (see Supplementary Methods and Supplementary Fig. 7).

**General strategy for multilocus integration of gene modules**. To decrease the turnaround time in metabolic engineering projects, we implemented the CRISPR/Cas9 strategy for multi-locus integration into loci reported to exhibit high-integration efficiency[33]. Multiple groups of donors are integrated at multiple loci, whereby each group is integrated into a single locus of different yeast cells. Each donor contains a yeast selection marker, an inducible ATF/BS, a CDS, and a yeast terminator amplified from Level 1 vectors, in addition to 50-bp upstream and downstream HRs allowing integration into predesigned genomic loci. The resulting library of yeast strains grows on plates containing appropriate yeast selection markers.

**Level 2 for assembly of the pathway library**. COMPASS is designed for multi-step combinatorial cloning of the multi-libraries of CDS modules ($Pro_{yeast\_auxotrophic/dominant\_marker}$-$Pro_{m-GAL1}$-ATF/BSs-CDS-$Ter_{yeast}$-$Pro_{E.\ coli\_selection\_marker}$-$CDS_{E.\ coli\_selection\_marker}$-$Ter_{E.\ coli}$) in the Destination vectors (library of one CDS module in each step), as decreasing the number of inserts in an assembly reaction increases cloning efficiency[34]. Another foremost advantage of this design is that the combinatorial libraries of each five-pathway-gene-module can be constructed in parallel in the two Destination vectors, which considerably reduces the time needed for subsequent combinatorial cloning. Our approach is based on the positive selection of successful constructs from both, in vitro and in vivo cloning procedures. Transformation-associated recombination (TAR) is the preferred method over

methodologies that employ *E. coli*. COMPASS allows ATF/BSs to control the expression of up to five CDSs in Destination vector I (Fig. 4a), and up to five CDSs in Destination vector II (Supplementary Fig. 8a). Integration of the ten CDS modules occurs at sites *p1* (CDS1), *p2* (CDS2 and CDS7), *p3* (CDS3 and CDS8), *p4* (CDS4 and CDS9), *p5* (CDS5 and CDS10), and *p6* (CDS6) (Supplementary Figs. 4 and 5).

As shown in Fig. 4b (and described in Supplementary Methods), nine ATF/BS-CDS2 modules are assembled at site *p2* of nine Destination vectors I-CDS1 to generate a library of 81 Destination vectors I-CDS1-CDS2. Cells with successful constructs grow on SC-Ura/-His medium, as the backbone and insert carry the *URA3* and *HIS3* selection markers, respectively. The plasmid library is recovered from the yeast cells, transformed into *E. coli*, and spread on LB agar plates containing ampicillin, where only cells harboring correct assemblies will grow. In three rounds of combinatorial cloning, ATF/BS-CDS3, ATF/BS-CDS4, and ATF/BS-CDS5 modules are successively assembled in sites *p3*–*p5*, correspondingly, of Destination vectors I-CDS1-CDS2 to generate the complete Destination vectors I-CDS1-CDS2-CDS3-CDS4-CDS5 library (Fig. 1). The number of constructs (library size) after each cloning step is $X_n = Y^n$, with $n$ = cloning step (1, 2, 3, 4, or 5), Y = number of regulators. Using the same pair of primers (Supplementary Methods), libraries of ATF/BS-CDS7, ATF/BS-CDS8, ATF/BS-CDS9, ATF/BS-CDS10 modules are assembled successively at sites *p2*–*p5*, correspondingly, starting with the Destination vectors II-CDS6 library (Fig. 1, Supplementary Methods, and Supplementary Fig. 8b).

**Single-locus integration of pathways**. In addition to multi-locus integration of Level 1 modules, COMPASS allows integrating completely assembled pathways (from a Level 2 Destination vector) into a single, defined locus of the yeast genome. Thereby, the position effect observed for transcriptional outputs of ATF/BS regulators positioned at multiple loci (Supplementary Note 1) is eliminated. This is important, when regulated expression of metabolic pathway genes is required.

Digestion of the library of Destination vector I with *Bam*HI or *Not*I allows integration of the library into the yeast *LYS2* or *URA3* genomic locus, respectively, by using the yeast's native homologous recombination system (Supplementary Fig. 4a). Moreover, Destination vector I.1 (Supplementary Fig. 6) and II (Supplementary Fig. 5a) were equipped with *Pme*I and *Pci*I recognition sequences flanked by 45-bp HRs to the left and right for integration into the *LYP1.x* and *ADE2.a* locus, respectively[33], by CRISPR/Cas9-mediated modification. *LYP1.x* and *ADE2.a* were selected for single-locus integration, because (i) gRNA-mediated targeting of *LYP1.x* and *ADE2.a* is possible with 100% disruption efficiency as previously reported[33] and the *LYP1*[35] and *ADE2* mutations[36] can be screened.

**One-step disruption of yeast markers by HI-CRISPR**. To further improve COMPASS—more specifically, to extend its genetic engineering capacity toward pathways with more than ten genes—we implemented Homology-Integrated CRISPR−Cas (HI-CRISPR) and its previously reported design principles[33], which allow the one-step disruption of multiple auxotrophic and dominant marker genes employed in the positive selection scheme of COMPASS. Plasmids pCOM002 (Supplementary Fig. 9) and pCOM009 (Supplementary Fig. 10) harbor a homology-integrated crRNA cassette for the disruption of five auxotrophic and dominant selection genes in a single step (in addition to iCas9 and tracrRNA). The plasmids are transformed into yeast to achieve frame-shift mutations of marker genes. Plasmid pCOM002 was used to generate the background parental

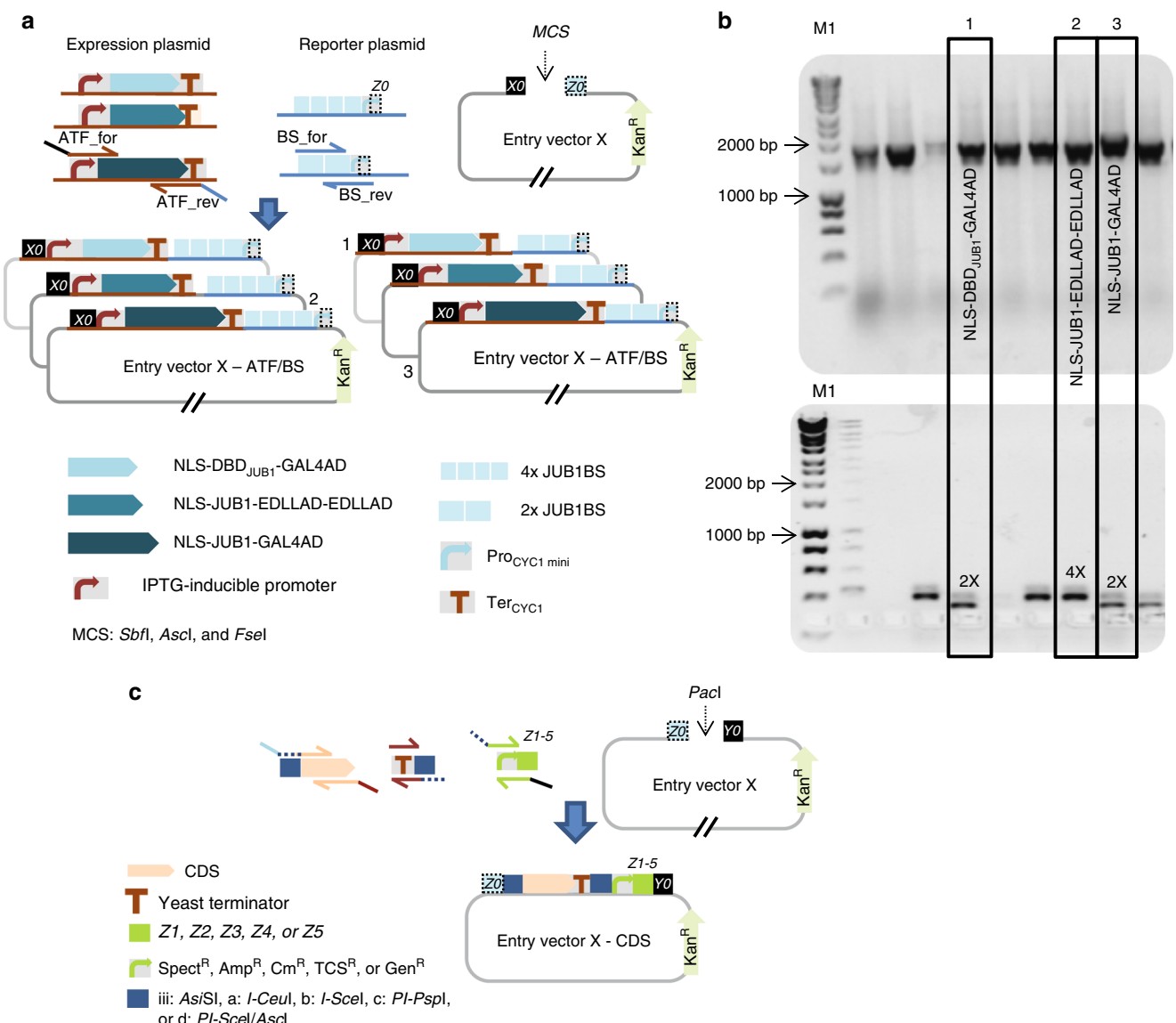

**Fig. 2** Combinatorial assembly of ATF/BS or CDS units in Entry vector X. **a** Combinatorial insertion of JUB1-derived ATF/BS units in Entry vector X. Three-partite fragments, harboring an IPTG-inducible promoter, the ATF, and the *CYC1* terminator are PCR-amplified from expression plasmids NLS-DBD$_{JUB1}$-GAL4AD (light blue arrows), NLS-JUB1-EDLLAD-EDLLAD (blue arrows), and NLS-JUB1-GAL4AD (dark-blue arrows)[4]. Two synthetic promoters, harboring two (two light-blue squares) and four (four light-blue squares) copies of the JUB1BS, are PCR-amplified from reporter plasmids[4]. ATFs and BSs amplicons are mixed in 1:10 molar ratio and cloned into *Fse*I/*Asc*I-linearized Entry vector X. The diagram shows six possible outcomes. The three required JUB1-derived ATF/BSs, NLS-DBD$_{JUB1}$-GAL4AD/2×, NLS-JUB1-EDLLAD-EDLLAD/4×, and NLS-JUB1-GAL4AD/2×, are numbered 1–3, correspondingly. Brown bent arrow, IPTG-inducible promoter. Brown "T", yeast *CYC1* terminator. **b** Gel electrophoresis to identify required JUB1-derived ATF/BSs. The verification of construct numbers 1–3 was done using colony PCR on ATF/BS fragments. M1: HyperLadder 1 kb (Bioline). **c** Insertion of CDS units into Entry vector X. CDSs (light-orange arrows) with rare RE sites (blue squares; iii: *Asi*SI, a: *I-Ceu*I, b: *I-Sce*I, c: *PI-Psp*I, d: *PI-Sce*I/*Asc*I) in their 5′ regions, a yeast terminator (brown "T") carrying similar RE sites (blue squares) in the 3′ region, and promoters of *E. coli* marker genes (green bent arrow) fused to *Z1, Z2, Z3, Z4,* or *Z5* are cloned into *Pac*I-digested Entry vector X. The *Z* and RE sites define based on Level 1 vectors (Destination vectors I/II: *Z1* and iii; Acceptor vectors A/E: *Z2* and a; B/F: *Z3* and b; C/G: *Z4* and c; and E/F: *Z5* and d). The HRs *X0, Z0–Z5,* and *Y0* are explained in footnote to Supplementary Data 3

strain auxotrophic for yeast dominant markers required to establish the Narion library (see: Coproduction of β-ionone and naringenin, and Methods). Transformed cells are selected for the presence of plasmids, followed by the selection for disrupted yeast selection markers; such cells do not grow on selective, but only on nonselective media.

**Building a library for controllable β-carotene production.** We sought to confirm the optimization ability of COMPASS for metabolic engineering applications by testing a well-known phenotype, the production of β-carotene[22,37].

At Level 0, we assembled CDSs required for β-carotene synthesis (i.e., *McrtI, BTS1,* and *McrtYB*)[37], in Entry vector X (Supplementary Data 5). At Level 1, the library of ATF/BS units was assembled upstream of the CDS units in Acceptor vectors A, B, and Destination vector I (Supplementary Fig. 11 and Supplementary Data 6). We studied β-carotene production in three approaches: (1) integration of the modules into three loci of the yeast genome (Fig. 5a); (2) transformation of Destination vector I containing the pathway modules into yeast cells without genome integration (Fig. 5b); and (3) integration of Destination vector I into the genome (Fig. 5c). We tested two yeast strains

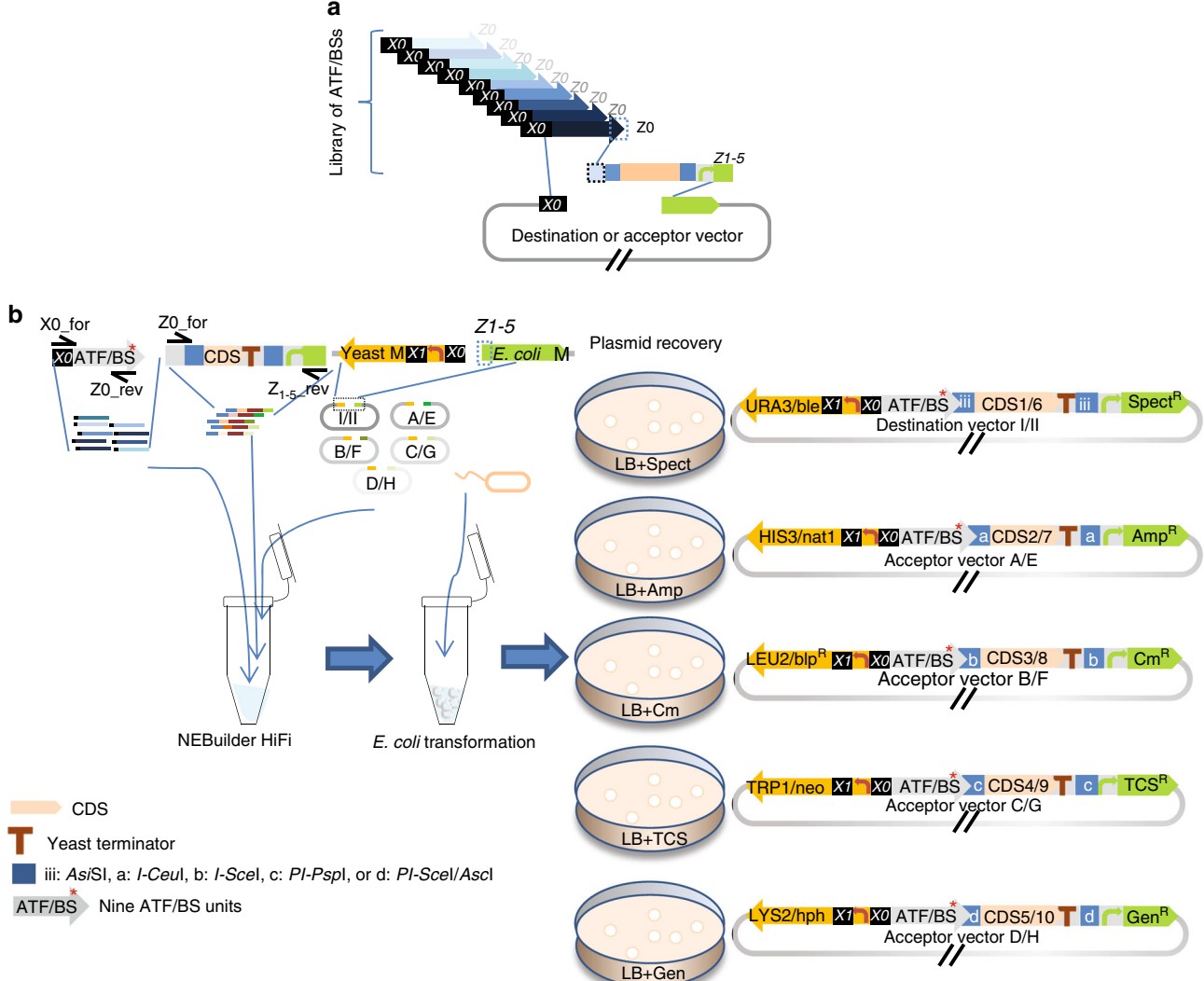

**Fig. 3** Combinatorial assembly of ATF/BS and gene units in the modules. **a** Combinatorial assembly of ATF/BS units upstream of CDS units. Nine ATF/BS units and CDS units from the Entry vectors X are PCR-amplified. The 5′ regions of ATF/BS units overlap with the *X0* sequence of the vector backbone, while their 3′ regions are identical to the 30 bp of the 3′ end of the minimal *CYC1* promoter (*Z0*) and overlap with the forward primer amplifying the CDS units. The 3′ regions of the CDS units overlap with *Z1–Z5* of five linearized vectors. **b** The COMPASS workflow to generate ATF/BS-CDS modules in vectors of either Set 1 or Set 2 (1/2). The PCR-amplified ATF/BS unites (primers X0_for/Z0_rev, on Entry vectors X-ATF/BS), five CDS units (primers Z0_for/Z1_rev/Z2_rev/Z3_rev/Z4_rev/Z5_rev at molar ratio 5:1:1:1:1:1, on mixed Entry vectors X-CDS) and five linearized Destination vectors I/II and Acceptor vectors A/E, B/F, C/G, and D/H are mixed at molar ratio 2:2:1 for in vitro overlap-based cloning in a single tube to generate different ATF/BS-CDS modules in the diverse vectors by providing the missing promoter sequences of the *E. coli* markers within the assembly units. Therefore, libraries of Destination vectors I-CDS1/II-CDS6, Acceptor vectors A-CDS2/E-CDS7, B-CDS3/F-CDS8, C-CDS4/G-CDS9, or D-CDS5/H-CDS10 are generated. X0_for overlaps with *X0*, Z0_rev overlaps with Z0_for, while primers Z1_rev–Z5_rev overlap with the downstream (right) HR of the linearized vector (called *Z1–Z5*). Light-orange arrow, CDS. Brown "T", yeast terminator. Blue squares, iii: *Asi*SI, a: *I-Ceu*I, b: *I-Sce*I, c: *PI-Psp*I, d: *PI-Sce*I / *Asc*I. Gray arrows, nine ATF/BS units. For simplicity, IPTG-inducible promoters and terminators are not included in the figure. *X0, Z0–Z5* are explained in footnote to Supplementary Data 4

with a CEN.PK background: (i) Gen 0.1 (Fig. 5d), which had been optimized for production of farnesyl pyrophosphate (FPP), a precursor of β-carotene (Methods, Supplementary Data 7); and (ii) IMX672.1 (Fig. 5e), lacking optimized FPP production (Methods, Supplementary Data 7). The constitutive yeast promoter *TDH3*[32] controlling the expression of the β-carotene CDSs served as positive control (Gen 0.1, Fig. 5f; IMX672.1, Fig. 5g). We then identified the combinations of ATF/BS fragments (Fig. 5h) in three colonies from each approach showing most intense β-carotene accumulation, and analyzed one of them by high-performance liquid chromatography (HPLC) (Fig. 5i). We observed that approaches 2, 1, and 3

resulted in the highest to lowest amount of β-carotene accumulation.

In approach 1 (Fig. 5a, Methods), a wide spectrum of colors ranging from light yellow to deep orange was observed in the colonies obtained (Gen 0.1, Fig. 5d, left plate; IMX672.1, Fig. 5e, left plate). Sequencing ATF/BS fragments (Fig. 5h) revealed that weak/medium, medium/strong, and medium/strong ATF/BS units, respectively, were assembled in *BTS1*, *McrtI*, and *McrtYB* modules in the Gen 0.1 background (CB1–CB3). In strain IMX672.1, weak/medium, strong, and medium ATF/BS units, respectively, were assembled in *BTS1*, *McrtI*, and *McrtYB* modules (CC1–CC3). We additionally tested the effect of strong

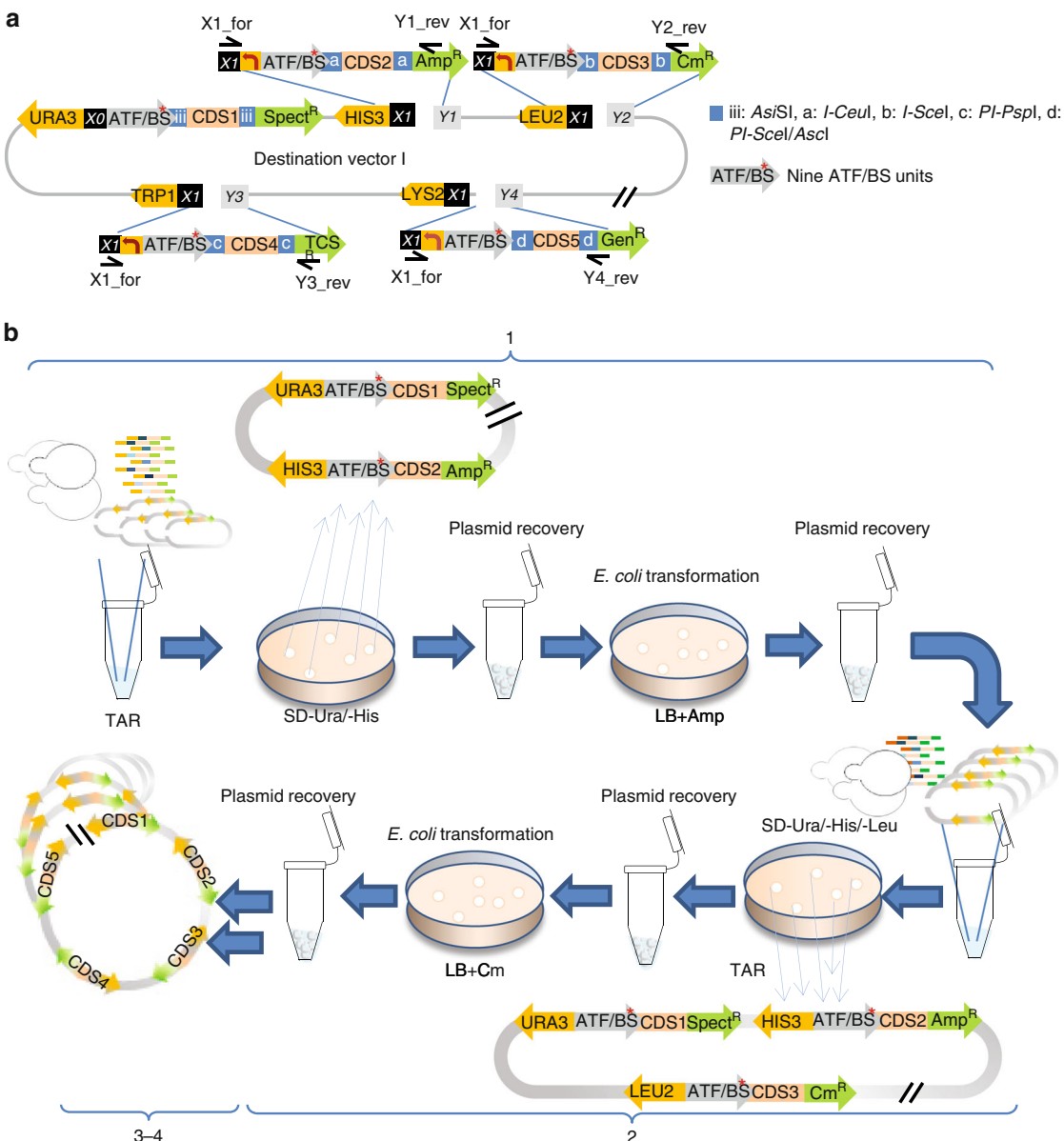

**Fig. 4** Combinatorial cloning of pathway modules into Destination vector I. **a** Combinatorial cloning of pathway genes into Destination vector I. The libraries of PCR-amplified (i) $Pro_{HIS3}$-ATF/BS-CDS2-$Pro_{AmpR}$-AmpR-$Ter_{AmpR}$ (primers X1_for/Y1_rev, on Acceptor vector A), (ii) $Pro_{LEU2}$-ATF/BS-CDS3-$Pro_{CmR}$-CmR-$Ter_{CmR}$ (primers X1_for/Y2_rev, on Acceptor vector B), (iii) $Pro_{TRP1}$-ATF/BS-CDS4-$Pro_{TCSR}$-TCSR-$Ter_{TCSR}$ (primers X1_for/Y3_rev, on Acceptor vector C), and (iv) $Pro_{LYS2}$-ATF/BS-CDS5-$Pro_{GenR}$-GenR-$Ter_{GenR}$ (primers X1_for/Y4_rev, on Acceptor vector D) modules are successively assembled in sites p2–p5, correspondingly, starting with the Destination vectors I-CDS1 library (see Fig. 3), in four rounds of combinatorial cloning. **b** The COMPASS workflow for combinatorial assembly of ATF/BS and gene modules into Destination vector I. The mixed ATF/BS-CDS2 modules are assembled using TAR in site p2 of Destination vectors I-CDS1. Yeast cells with successful constructs grow on SC-Ura/-His medium. Cells are scraped from the plates, the plasmid library is extracted to obtain a pool of all randomized members, transformed into E. coli, and cells are grown on LB plates containing ampicillin. Cells are scraped from the plates to extract the plasmid library (1). The ATF/BS-CDS3 modules are assembled in site p3 of the Destination vectors I-CDS1-CDS2 library. Yeast cells with successful constructs grow on SC-Ura/-His/-Leu medium (2). The libraries of ATF/BS-CDS4 and ATF/BS-CDS5 modules are cloned into sites p4 and p5, respectively (3 and 4). Gray arrows, nine ATF/BS units. Blue squares, iii: AsiSI, a: I-CeuI, b: I-SceI, c: PI-PspI, d: PI-SceI / AscI. For simplicity, the IPTG-inducible promoters and terminators are not included in the figure. X0 and X1 are explained in footnote to Supplementary Data 4. Y1–Y4 overlap with the last 30 bp of terminators of the $Amp^R$, $Cmr^R$, $TCS^R$, and $Gen^R$ genes

regulators (CB4 and CC4). Importantly, we only observed yellow colonies, indicating insufficient metabolic flux toward β-carotene or activation of alternative pathways. As a further control, modules expressing the three genes from the *TDH3* promoter were integrated in the same loci of Gen 0.1 (Fig. 5f, left plate) and IMX672.1 (Fig. 5g, left plate). Almost all colonies were yellow. The HPLC data (Fig. 5i) demonstrate that more β-carotene was produced in the optimized strain than the wild type (WT) (CB1, $0.43 \pm 0.03$ mg g$^{-1}$ cdw; CC1; $0.41 \pm 0.03$ mg g$^{-1}$ cdw). Moreover, the colonies with ATF/BS control modules produced 1.2- to 1.5-fold more β-carotene than colonies with *TDH3* promoters.

We next used flow cytometry for high-precision growth rate measurements of the modified strains (Supplementary Note 3, Supplementary Fig. 12a, Supplementary Data 8). Strain CB1 did

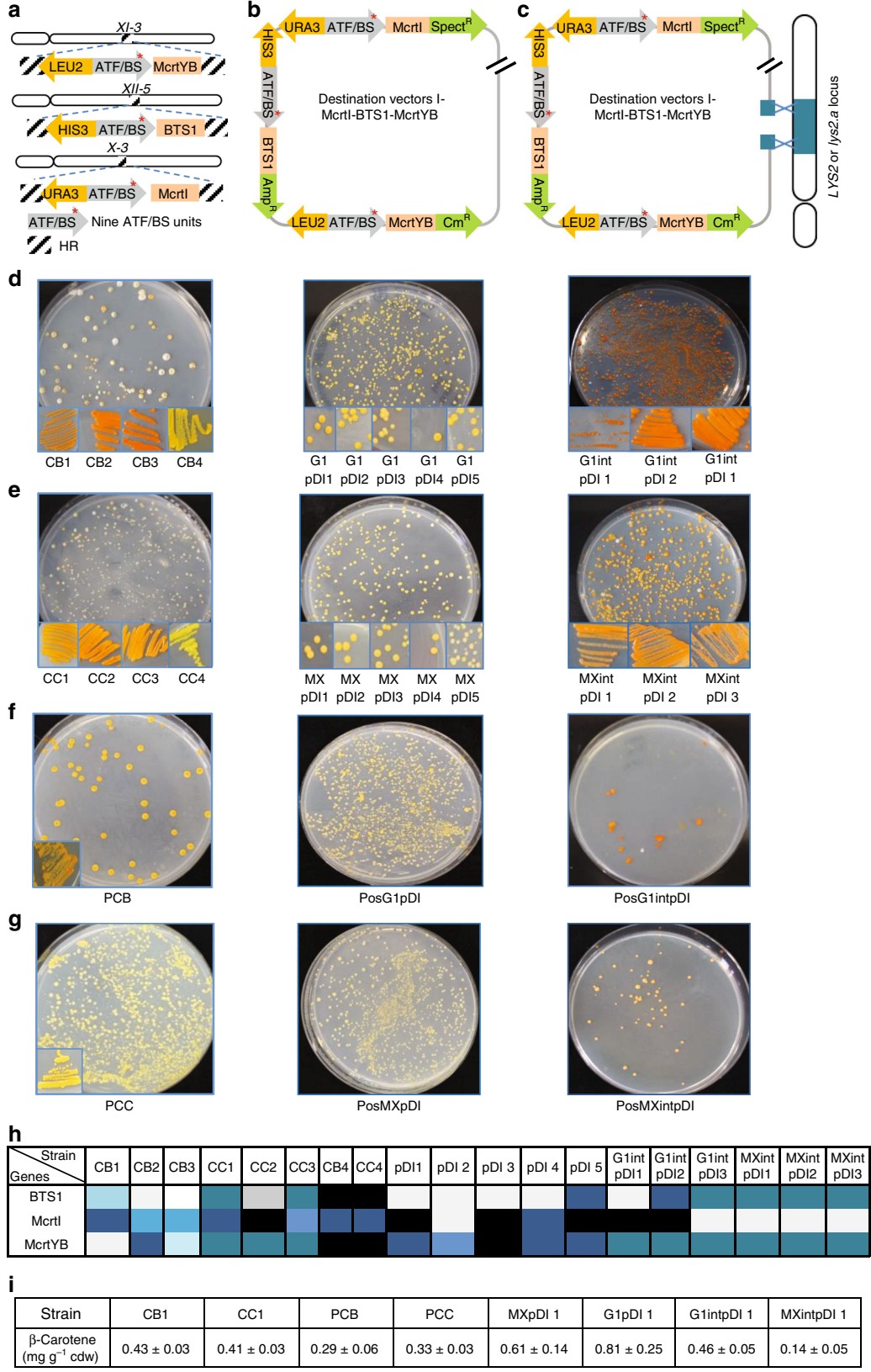

h

| Genes \ Strain | CB1 | CB2 | CB3 | CC1 | CC2 | CC3 | CB4 | CC4 | pDI1 | pDI 2 | pDI 3 | pDI 4 | pDI 5 | G1int pDI1 | G1int pDI2 | G1int pDI3 | MXint pDI1 | MXint pDI2 | MXint pDI3 |
|---|---|---|---|---|---|---|---|---|---|---|---|---|---|---|---|---|---|---|---|
| BTS1 | | | | | | | | | | | | | | | | | | | |
| McrtI | | | | | | | | | | | | | | | | | | | |
| McrtYB | | | | | | | | | | | | | | | | | | | |

i

| Strain | CB1 | CC1 | PCB | PCC | MXpDI 1 | G1pDI 1 | G1intpDI 1 | MXintpDI 1 |
|---|---|---|---|---|---|---|---|---|
| β-Carotene (mg g⁻¹ cdw) | 0.43 ± 0.03 | 0.41 ± 0.03 | 0.29 ± 0.06 | 0.33 ± 0.03 | 0.61 ± 0.14 | 0.81 ± 0.25 | 0.46 ± 0.05 | 0.14 ± 0.05 |

not show a reduction in growth compared to wild type (WT), when the regulators and pathway genes were not expressed (noninducing medium) (Supplementary Fig. 12b, c, Supplementary Data 8). In contrast, strain PCB which constitutively expresses the β-carotene biosynthesis genes from the yeast

*TDH3* promoter, showed 3.6% reduction in growth rate compared to WT in noninducing medium (Supplementary Fig. 12d, Supplementary Data 8). When regulators were expressed (inducing medium), strain CB1 showed 15% reduction in the growth rate compared to WT (Supplementary Fig. 12b, c,

**Fig. 5** β-Carotene pathway optimization using COMPASS. **a** Overview of the three-gene β-carotene pathway library with integration of the genes into the *X-3* (McrtI), *XI-3* (McrtYB), and *XII-5* (BTS1) loci (approach 1), **b** assembled in Destination vector I (pDI) (approach 2), or **c** assembled in pDI followed by integration into the *LYS2/lys2.a* locus (approach 3). *URA3*, *LEU2*, and *HIS3* allow selection on SC-Ura/-Leu/-His media, while *E. coli* markers *Spect^R*, *Amp^R*, and *Cm^R* allow selection on LB media containing ampicillin, chloramphenicol, and spectinomycin. For simplicity, the IPTG-inducible promoters upstream of ATF/BSs, terminators, and cleavage sites flanking pathway genes are not shown. **d** β-Carotene production controlled by the nine ATF/BS units in strain Gen 0.1 or **e** IMX672.1. CB1–3 and CC1–3 (left plate), G1intpDI1–3 and MXintpDI1–3 (right plate) represent colonies with deep-orange colors from libraries containing randomized ATF/BSs upstream of β-carotene CDSs, while CB4 and CC4 represent modules expressing β-carotene CDSs from strong ATF/BSs. The plasmid libraries obtained with approach 2 were transformed into both background strains. Plasmid libraries extracted from the yeasts were transformed into *E. coli*. Five sequenced plasmids (pDI1–5, middle plate) were retransformed into both background strains to achieve strains G1pDI1–5 and MXpDI1–5. **f** Modules containing β-carotene CDSs under the control of the constitutive yeast *TDH3* promoter were engineered into strains Gen 0.1 or **g** IMX672.1 to obtain PCB and PCC (left plate), PosG1pDI and PosMXpDI (middle plate), PosG1intpDI and PosMXintpDI (right plate). **h** Identification of ATF/BS units. The color code is given in Supplementary Fig. 1. **i** HPLC analysis of for three different colonies of strains. Gray arrows, nine ATF/BS units. Black/white-striped squares, homology regions. Data are means ± SD from three biological replicates

Supplementary Data 8). We furthermore investigated the orthogonality of the system and the productivity of the IPTG-inducible system utilized in the current version of COMPASS (Supplementary Note 4). In CB1, induction of the regulators led to 8.5-fold more β-carotene than in cells grown in non-inducing medium (Supplementary Fig. 13, Supplementary Data 9). This is 1.5-fold more than in strain PCB (Fig. 5i, Supplementary Fig. 13, Supplementary Data 9). As the inducible plant regulators employed in COMPASS allow decoupling of growth from metabolite production phases, carbon sources can be redirected towards the production of high levels of targeted compounds at the desired time points.

At Level 2, the pDI-McrtI-BTS1-McrtYB library was transformed into strains Gen 0.1 or IMX672.1 (Supplementary Note 5, middle plate in Fig. 5). Because colony color may result from the presence of several plasmids within a given yeast cell, the plasmids were recovered from the cells and transformed into *E. coli*. The ATF/BS fragments of five Destination vectors were sequenced. Single plasmids were then transformed into strains Gen 0.1 or IMX672.1. The results showed that combining a weak ATF/BS (*BTS1*) with two strong ATF/BSs (*McrtI* and *McrtYB*) results in superior β-carotene accumulation in both strains. Moreover, we observed that expressing all genes from strong ATF/BS units (pDI 5, middle plate in Fig. 5) did not result in high β-carotene accumulation. HPLC analysis revealed no significant difference in β-carotene level between the background-optimized strain (G1pDI 1; 0.81 ± 0.25 mg g$^{-1}$ cdw) and the WT (MXpDI 1; 0.61 ± 0.14 mg g$^{-1}$ cdw).

Next, the Destination vector I library was integrated into the *lys2.a* locus of strains Gen 0.1 or IMX672.1 (Supplementary Note 6, right plate in Fig. 5). Weak/medium/strong, weak/strong, and medium ATF/BS regulators, respectively, were employed in the *BTS1*, *McrtI*, and *McrtYB* modules in the Gen 0.1 background, while weak/medium and weak/strong ATF/BS regulators controlled the expression of *BTS1* and *McrtI*, respectively, and all *McrtYB* expressing modules contained strong ATF/BS regulators in the IMX672.1 background. Moreover, 3.3-fold more β-carotene was produced in the background-optimized strain Gen 0.1 (G1intpDI 1) than in the nonoptimized IMX672.1 strain (MXintpDI 1). Compared to WT, strain MXintpDI 1 did not show a growth defect in noninducing medium, while it showed a 5.5% reduction in growth rate in inducing medium (Supplementary Fig. 12e, f, Supplementary Data 8).

Overall, the top producer, generated by approach 2, yielded 2.1-fold more β-carotene than IME167[38] tested in our experiment (Supplementary Note 7 and Supplementary Data 10). Moreover, our results strongly indicate that a combination of weak, medium, and strong ATF/BSs is required for high-level β-carotene production. In some cases, e.g., CC1, CC3, CC4, CB4, and pDI 5 (Fig. 5h), the same ATF/BS regulator controls expression of two

different pathway genes. An ATF may compete for binding of the respective target site(s) within synthetic promoters. Previous studies showed that specifically bound TFs affect the transcriptional output, while the concentration of the TF seems less important[39]. Moreover, we previously showed that a higher transcriptional output is often obtained by increasing the copy number of a BS of a plant regulator[4], suggesting that the abundance of the ATF expressed from an IPTG inducible promoter is likely sufficient to target more than a single copy of the BS. However, our results (Supplementary Note 8, Supplementary Fig. 14, Supplementary Data 11) demonstrate that a combined expression of two fluorescent reporter proteins, AcGFP1 and DsRED, each controlled by the same regulator (NLS-JUB1-EDLLAD-EDLLAD/4×), leads to a lower output compared to strains individually expressing the reporters from the same regulator (AcGFP1: 66%, DsRED: 79% of the levels achieved upon separate expression). The result is in accordance with the model that cells allocate their limited resources to different tasks including the production of regulators that might create an increased metabolic burden leading to a drop in the biosynthetic performance[8–13].

**Building a library for controllable β-ionone production.** We next established a pathway for the biosynthesis of β-ionone in the best β-carotene producers achieved by the three approaches reported above (Supplementary Note 9). RiCCD1 converts β-carotene to β-ionone which leads to yeast cells that are less intensely colored than β-carotene-producing cells. We used approaches 1 (Supplementary Fig. 15), 2 (Supplementary Fig. 16) and 3 (Supplementary Fig. 17), as described in Supplementary Notes 10–12, respectively. We found that approach 3, 1, and 2 resulted in high, medium and low amounts of β-ionone production, respectively. Our data demonstrate that (i) in 86% of the cases a medium or high expression of *RiCCD1* was required to produce a high level of β-ionone in high-β-carotene accumulators and (ii) even more β-ionone accumulated in strain IMX672.1 compared to Gen 0.1, demonstrating that superior β-carotene accumulation is not per se sufficient for a high-level accumulation of β-ionone. However, our top β-ionone producer, strain MXintR1pDI 1 (Supplementary Fig. 17), yielded 3.3-fold more β-ionone than control strain RiCCD1, tested in our experiment (Supplementary Note 7 and Supplementary Data 10). Strain RC1 did not show a growth defect compared to WT in noninducing medium, while it showed 9.5% reduction in growth rate compared to WT in inducing medium (Supplementary Fig. 12e, g, Supplementary Data 8).

**Co-production of β-ionone and naringenin.** To test the versatility and optimization capacity of COMPASS, we co-engineered

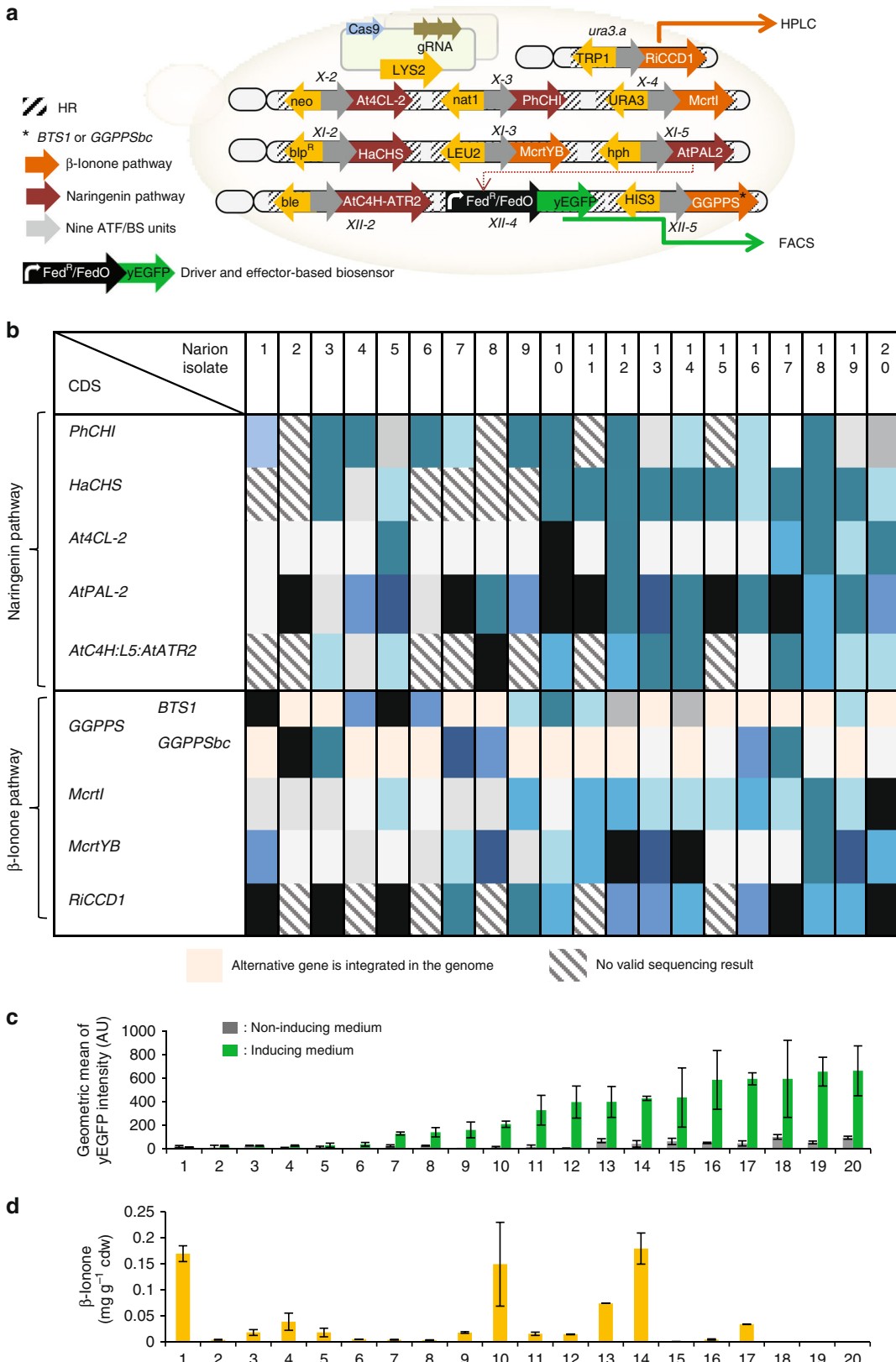

the pathways for β-ionone (four genes)[38] and naringenin (NG; five genes; *AtC4H:L5:AtATR2, PhCHI, HaCHS, At4CL-2,* and *AtPAL-2;* Fig. 6)[31] production in *S. cerevisiae*. We detected NG accumulation using a cellular biosensor[31]. To produce β-ionone, the library of *McrtI, McrtYB, BTS1* (or *GGPPSbc*), and *RiCCD1* CDSs from Level 1 was used (see above and Methods). For NG biosynthesis, five CDS-terminators[31] and promoters of *E. coli* selection markers were assembled in the five Set 2 vectors

**Fig. 6** Diversity of β-ionone and NG production from a randomized ATF/BS library. **a** Schematic overview of the multi-locus integration of β-ionone and NG pathway genes. The *McrtI-, CrtE-, McrtYB-, RiCCD1-, PhCHI-, HaCHS-, At4CL-2-, AtPAL-2-,* and *AtC4H-ATR2*-CDS donors were integrated into the *X-4, XII-5, XI-3, ura3.a, X-3, XI-2, X-2, XI-5, XII-2* loci of MXFde0.2, respectively, via CRISPR/cas9-mediated multi-locus integration. Plating cells on SC-Ura/-Leu/-His/-Trp/-Lys media containing G418, hygromycin B, phleomycin, bleomycin, and nourseothricin allows maintaining the Cas9/sgRNA encoding plasmids and screening integrated cassettes. Subsequently, ATFs are expressed from an IPTG-inducible promoter in the presence of inducers and target their BSs upstream of pathway genes. β-Ionone production is quantified by HPLC, while the production of NG is detected using the FdeR biosensor. This results in yEGFP expression is detectable by flow cytometry. Black/white-striped squares, homology regions. Black asterisk, BTS1 or GGPPSbc. Orange arrow, β-ionone pathway genes. Brown arrow, naringenin pathway genes. Gray arrows, nine ATF/BS units. Black arrow, driver and effector-based biosensor. Green arrow, yEGFP fluorescence reporter. For simplicity, the IPTG-inducible promoter, terminator, and cleavage sites are not included in the figure. **b** Sequencing results of ATF/BS units present upstream of each CDS. The color code is given in Supplementary Fig. 1. Beige boxes indicate that an alternative gene was integrated into the genome. Black/white striped boxes indicate that no valid sequencing results were obtained. **c** Screening of NG production. Twenty colonies were pre-cultured in SC medium with appropriate selection markers and subsequently used to monitor yEGFP output in the absence (YPDA, 2% (w/v) glucose) and presence of inducer (YPDA, 2% (w/v) galactose and 20 μM IPTG). Gray, noninduction medium; green, induction medium. Data are geometric means ± SD ($n = 9$) of the fluorescence intensity obtained from three cultures, each derived from an independent yeast colony and determined in three technical replicates. AU arbitrary units. Full data are given in Supplementary Data 13. **d** HPLC analysis for β-ionone production. Data are means ± SD from three biological replicates. Full data are given in Supplementary Data 13

(Methods, Supplementary Methods). Subsequently, the five-gene pathway and the nine ATF/BS units were assembled in Destination vector II and Acceptor vectors E–H in a single-reaction tube to construct a library of 45 plasmids (Supplementary Fig. 7; Supplementary Data 6). The nine regulator gene module libraries were integrated into the nine genomic loci of strain MXFde0.2, which harbors the NG biosensor (Methods and Supplementary Fig. 9). The selection for correct integrations was carried out on SC medium containing appropriate selection markers (Fig. 6a). The resulting library was called Narion and each colony was expected to contain one out of $2 \times 9^9$ possible combinations of nine ATF/BS units upstream of the nine CDSs. Twenty colonies (representing 0.0000025% of the theoretical complexity of the library and ~0.05% of its actual complexity) were randomly selected to identify ATF/BS sequences driving expression of the CDSs (using primers listed in Supplementary Data 12). Of note, all colonies were unique with respect to the combinations of ATF and CDS sequences (Fig. 6b). Flow cytometry revealed that approximately 30% of the library members show no- or low-NG production (Narion 1–6; Fig. 6c, Supplementary Data 13). Half of the Narion isolates were categorized as midrange producers in which weak and medium ATF/BSs control expression of most of the NG genes. The better producers (Narion 16–20) harbor mostly medium ATF/BS units upstream of NG genes (except Narion 17 and 20, which both contain strong ATF/BS units upstream of *AtPAL-2*). HPLC analyses (Fig. 6d, Supplementary Data 13) showed that 40% of the library isolates (Narion 2, 6–8, 15–16, 19, and 20) produced β-ionone at a level of less than 0.01 mg g$^{-1}$ cdw, while half of the strains (Narion 3–5, 9, 11–13, 17, and 18) produced 0.01–0.1 mg g$^{-1}$ cdw of β-ionone. Approximately, 15% of the library isolates produced 0.1–0.2 mg β-ionone g$^{-1}$ cdw and harbor a combination of weak, medium and strong ATF/BSs upstream of β-ionone genes. The highest β-ionone yield (0.18 ± 0.015 mg g$^{-1}$ cdw) was observed in Narion 14. Surprisingly, Narion 19 and 20, top NG accumulators, produced the lowest amount of β-ionone, and Narion 1, producing the lowest amount of NG, was the second-best β-ionone accumulator. Our HPLC data demonstrate that the presence of medium and strong ATF/BSs in *McrtI* expressing modules results in low-level production of β-ionone. Previously, Ding et al.[2] reported that GGPPSbc leads to an improved GGPP supply over BTS1. However, we observed that *BTS1* expressing yeasts produce more β-ionone than *GGPPSbc* expressing cells in 20 characterized Narion isolates. Taken together, we identified Narion 14 as the best producer strain as it produces a medium level of NG and the highest level of β-ionone. In Narion 14, the expression of NG and β-ionone pathway genes is controlled by weak/medium and weak/medium/strong ATF/BS regulators, respectively. Overall, through checking 0.0000025% of the theoretical complexity of the library, Narion 14 (Fig. 6d) was identified that yielded 4.2-fold more β-ionone than RiCCD1[38], tested in our experiment (Supplementary Data 10).

## Discussion

Projects in synthetic biology often require the expression of multiple genes[8]. Typically, the required expressional activities of those genes are not known a priori. As a consequence, a large number of synthetic constructs must be constructed and tested in the target organism. The complexity of such libraries rapidly increases with the number of promoters and genes combined.

Establishing and testing complex libraries requires: (i) A high-throughput and reliable method for the assembly of large numbers of diverse constructs[40]. (ii) Methods to determine product output[41]. Often, cell colonies are screened for the formation of colored products (at least in test cases)[22], but can also be screened by, e.g., HPLC. A more advanced approach employs biosensors that detect the compounds at the single-cell level[31].

An important aspect of COMPASS is that it only requires 11 core vectors (one Entry, eight Acceptor, and two Destination vectors) to establish combinatorial libraries of thousands to millions of constructs for ten-gene pathways. To extend the capacity of COMPASS even further—for pathways with more than ten genes—we included two additional plasmids (pCOM001 and pCOM009) in the toolkit. In the current version of COMPASS, we are able to generate libraries of stable yeast variants with a complexity of theoretically 3,486,784,401 different members through only four cloning reactions followed by the decupled integration of the constructs into the genome. To achieve this large number of constructs we employ only nine of the 106 plant-derived ATF/BS pairs (plant regulators)[4] and ten enzyme-encoding open reading frames. The depth at which COMPASS generates diversity is defined by the number of regulators (Y) and open reading frames (N) (size of the library = Y$^N$). To demonstrate a useful application of COMPASS, we generated a library of β-carotene producers. Our results clearly show that a combination of different ATF/BSs units, leading to different expression levels of the enzyme-encoding CDSs, was needed for high-level β-carotene production (0.81 ± 0.025 mg β-carotene g$^{-1}$ cdw). In our second setup, we coproduced the biosensor-responsive chemical NG and the colorless product β-ionone. Analyzing less than 0.0000025% of the library revealed that approximately 30% of the library members show no or low levels of both chemicals,

highlighting the importance of developing combinatorial optimization approaches. In addition, we found a strain producing $0.18 \pm 0.015$ mg β-ionone g$^{-1}$ cdw (4.2-fold more β-ionone than RiCCD1[38]) demonstrating the optimization capacity of COMPASS.

Most current methods for pathway engineering rely on growth-coupled biosynthesis with constitutive promoters controlling the expression of pathway genes. However, growth-coupled production limits metabolite production as the carbon source is consumed for both, biomass formation and production of metabolites triggered by the newly introduced pathway. Therefore, COMPASS employs modular and inducible transcription factor-based controllers that allow decoupling growth and production phases. Thereby, biomass can accumulate rapidly during the growth phase and metabolite production (and then limited growth) is obtained during the production phase. Our growth data indicate that growth rates of high-β carotene producers are not impaired as long as the plant regulators are not expressed in non-inducing medium, while after their induction growth rates decrease by 7–16%. Thus, yeast cells harboring COMPASS regulatory machineries can first be cultivated in noninducing medium to gain high-cell densities, and then induced for metabolite production.

Currently, we utilize COMPASS in three technical settings. Approach 1 is a fast method for fine-tuning gene expression output through only four cloning reactions, followed by the decupled integration of the plant regulatory-CDS modules into the genome to generate large libraries of stable yeast variants. Approach 2 establishes plasmid-based systems which may be favorable in some cases over genomic integration due to their easy manipulation. Notably, approach 3, where multigene constructs are integrated in a single genomic locus, is a suitable alternative for applications requiring a predicted behavior of the regulators.

Further improvements may include the utilization of other inducers (e.g., light[42]), adding dynamic regulation to the system[39], and adopting COMPASS to a wider range of hosts. In the present study, we used the COMPASS toolkit to optimize the production of biochemical compounds. However, COMPASS can facilitate many other projects in synthetic biology, including, e.g., the building of multisubunit protein complexes, the engineering of sophisticated gene-regulatory networks, or the construction of entire synthetic organelles. COMPASS thus has a great potential in many areas of synthetic biology.

## Methods

**General**. The list of *S. cerevisiae* strains used in this study and their genotypes are given in Supplementary Data 7. Plasmids were constructed by NEBuilder HiFi DNA assembly (New England Biolabs, Frankfurt am Main, Germany) and SLiCE cloning[43]. Plasmids and primer sequences are given in Supplementary Data 14 and Supplementary Data 15. PCR amplifications of DNA fragments were done using high-fidelity polymerases: Phusion Polymerase (Thermo Fisher Scientific), Q5 DNA Polymerase (New England Biolabs) or PrimeSTAR GXL DNA Polymerase (Takara Bio, Saint-Germain-en-Laye, France) according to the manufacturer's recommendations. Amplified DNA parts were gel-purified prior to further use, except when noted otherwise. Moreover, multiplex PCR-amplified fragments were not gel-purified (Supplementary Methods). All primers and oligonucleotides were ordered from Eurofins Genomics (Ebersberg, Germany). All constructs were confirmed by sequencing (LGC Genomics, Berlin, Germany).

**Bacterial and yeast strains**. Plasmids were transformed into *E. coli* NEB 5α or NEB 10β cells (New England Biolabs), or into ElectroSHOX Competent Cells (Bioline, Luckenwalde, Germany). Strains were grown in Luria–Bertani medium with appropriate selection marker at 28 °C (triclosan, 14.5 µg/ml) or 37 °C (spectinomycin, 50 µg/ml; ampicillin, 50 µg/ml; chloramphenicol, 25 µg/ml; gentamicin, 50 µg/ml).

*S. cerevisiae* strains YPH500 (ATCC: 76626), IMX672 (Euroscarf, #Y40595), and SCIGS22a[30] were used. Information regarding yeast strains constructed in this work is presented in Supplementary Data 7. Generation of competent yeast cells and genetic transformation of plasmids or linearized DNA fragments were done using either the LiAc/SS carrier DNA/PEG method[44] or The Frozen-EZ Yeast Transformation II Kit (Zymo Research, Freiburg, Germany). All strains were grown at 30 °C in yeast extract peptone dextrose adenine (YPDA)-rich medium or in appropriate synthetic complete (SC) media lacking one or more amino acids to allow selection for transformed cells. Dominant selection markers were used in YPDA medium at the indicated final concentrations: G418 (200 µg/ml), hygromycin B (200 µg/ml), phleomycin (20 µg/ml), bleomycin (100 µg/ml), and nourseothricin (100 µg/ml). Verification of transformation was done using colony PCR followed by sequencing. Either Zymoprep Yeast Plasmid Miniprep II kit (Zymo Research) or a method for isolating high-quality circular plasmid DNA in microgram quantities[27] were used to recover plasmid DNA from positive-yeast clones.

**Construction of pCOM plasmids for strain generation**. pCOM001: Donor and direct repeat sequences required for disruption of yeast auxotrophic selection markers were generated by gene synthesis and inserted into plasmid pEXA2 by MWG Eurofins to create plasmids pCOMA and pCOMB. The plasmids served as templates to PCR-amplify fragments A (primers COMA_for/COMA_rev, on pCOMA) and B (primers COMB_for/COMB_rev, on pCOMB). The two fragments were individually assembled into *Bsa*I-digested pCRCT plasmid (Addgene #60621), and the resulting plasmids were named pCOMC (containing fragment A) and pCOMD (containing fragment B). Subsequently, PCR-amplified fragment B (primers COMD_for/COMD_rev, on pCOMD) was assembled into *Bsa*I-digested pCOMC. The resulting plasmid was named pCOM001 and used for the disruption of *LEU2*, *HIS3*, *LYS2*, and *TRP1* marker genes.

pCOM002: To construct pCOM002, a PCR-amplified *neo* gene (primers KAN_for/KAN_rev, on pTAJAK92[45]), a fragment containing *CYC1* terminator - structural RNA - *STU4* terminator (tCYC1_for/tCYC1_rev, on BY4741 genomic DNA), and an *SNR52* fragment (primers SNR_for/SNR_rev, on pCRCT, Addgene #60621) were assembled into *Bsa*I-digested pCOM001. The resulting plasmid was named pCOM002.

pCOM003: The *CEN/ARS* origin of replication (CEN_for/CEN_rev, on pGN006[4]) was cloned into *Pme*I-digested pGN003B[4] to construct pGN003BM. Next, plasmid pCOM003 was constructed by Gibson assembly, amplifying the Cre-EBD transcription unit from pDL12[32], and its insertion into the *Eco*RI/*Sac*II-digested pGN003BM[4].

pCOM004: The *LYS2* encoding fragment (LYSA_for/LYSA_rev, on pYC6Lys-TRP1URA3, Addgene #11010) and iCas9 encoding fragment (CASA_for/CASA2_rev, on pCRCT, Addgene #60621) were cloned into *Nco*I/*Not*I-digested pTAJAK-92[45].

pCOM005: The *LYS2* encoding fragment (LYSA_for/LYSA_rev, on pYC6Lys-TRP1URA3, Addgene #11010) and iCas9 encoding fragment (CASA_for/CASA_rev, on pCRCT, Addgene #60621) were cloned into *Nco*I/*Sph*I-digested pCfB3052 (Addgene #73294).

pCOM006: The *LYS2* encoding fragment (LYSB_for/LYSA_rev, on pYC6Lys-TRP1URA3, Addgene #11010), iCas9 encoding fragment (CASA_for/CASB_rev, on pCRCT, Addgene #60621), and HphR encoding fragment (HYG_for/HYG_rev, on Acceptor vector H, see COMPASS vector section) were cloned into *Sfo*I-digested pTAJAK-105[46].

pCOM007: The *LYS2* encoding fragment (LYSA_for/LYSA_rev, on pYC6Lys-TRP1URA3, Addgene #11010) and iCas9 encoding fragment (CASA_for/CASA_rev, on pCRCT, Addgene #60621) were cloned into *Nco*I/*Sph*I-digested pCfB3051 (Addgene #73293).

pCOM008: The *LYS2* encoding fragment (LYSA_for/LYSA_rev, on pYC6Lys-TRP1URA3, Addgene #11010) was cloned into *Nco*I/*Sph*I-digested pCfB3053 (Addgene #73295).

pCOM009: To construct pCOM009, fragments C (primers COMC1_for and COMC1_rev, on annealed single-stranded oligonucleotides COMC2_for/COMC2_rev) and D (primers COMD1_for/COMD2_rev, on annealed single-stranded oligonucleotides COMD2_for/COMD2_rev) were PCR-amplified. Subsequently, the *Bsa*I-digested fragments C and D were inserted into *Bsa*I-digested pCRCT plasmid (Addgene #60621). The resulting plasmid was named pCOMCD. Next, fragment E (primers COME1_for/COME1_rev, on annealed single-stranded oligonucleotides COME2_for/COME2_rev) was PCR-amplified followed by digestion with *Bsa*I and cloning into *Bsa*I-digested pCOMCD. The resulting plasmid was named pCOM009 and used for *ble*, *nat1*, *blp*R, *neo*, and *hph* gene disruption.

**Construction of yeast strains**. Strain Gen 0.1: The *S. cerevisiae* strain SCIGS22a[30,47] was used in this work to generate strain Gen 0.1. Strain SCIGS22a has a CEN.PK background with additional modifications in the genome for the overaccumulation of FPP[30,47] and is auxotrophic for *URA3*. COMPASS employs a positive-selection scheme that involves five auxotrophic marker genes. The marker´s coding sequences are provided within the vector backbones, while their corresponding promoters are part of the assembly fragments used in the TAR reaction. The promoter drives expression of the corresponding auxotrophic marker only in successfully assembled constructs. Therefore, SCIGS22a needed to be auxotrophic for *HIS3*, *LEU2*, *TRP1*, and *LYS2*, in addition to *URA3*. Hence, we used the Homology-Integrated CRISPR−Cas (HI-CRISPR) system and its design principles for one-step multiple auxotrophic gene disruption[33]. HI-CRISPR uses plasmid

pCRCT (Addgene, #60621), a high-copy plasmid harboring iCas9, a variant of wild-type (WT) Cas9 that increases the gene disruption efficiency, trans-encoded RNA (tracrRNA), and a homology-integrated crRNA cassette. It relies on the insertion of a 100-bp dsDNA mutagenizing homologous recombination donor between two direct repeats in the case of each target gene. Multiple donors and corresponding guide sequences can be introduced in pCRCT. For each gene disruption, a gRNA targeting 20-bp unique sequence was selected via BLAST searches against the *S. cerevisiae* S288c genome (NCBI Taxonomy ID: 559292) to minimize off-target effects. In the next step, plasmid pCOM001 for quadruple auxotrophic gene disruption (see above and Supplementary Fig. 9a) was transformed into *S. cerevisiae* SCIGS22a cells. Hence, *leu2.a* (519-bp downstream of the *LEU2* start codon), *his3.a* (265-bp downstream of the *HIS3* start codon), *lys2.a* (799-bp downstream of the *LYS2* start codon), and *trp1.a* (245-bp downstream of the *TRP1* start codon) were targeted to achieve frame-shift mutations. The transformed cells were inoculated in liquid SC-Ura culture overnight. After 4 days, 200 μl of a $10^4$-fold diluted cell culture were plated on SC-Ura plates. After 2 days, a total of 50 colonies were randomly selected and each single colony was streaked out onto four different selective plates (i.e., SC-Leu, SC-His, SC-Lys, and SC-Trp). After 2 more days, cells of a colony that did not grow on either of the four selective plates were streaked out on nonselective YPDA agar medium to eliminate the pCOM001 plasmid. After four rounds of re-streaking single colonies onto new YPDA plates, we recovered a colony that was also not able to grow on SC-Ura medium. The new strain was named Gen 0.1.

Strain IMX672.1: To achieve a *lys2.a* frame-shift mutation in strain IMX672 (derived from strain CEN.PK), plasmid pCOM001 was transformed into the strain and selection was done as described for Gen 0.1 above. Thirty colonies were randomly streaked out on SC-Lys and YPDA plates. The colony that did not grow on SC-Lys plate carries the *lys2.a* frame shift mutation. To remove pCOM001, the corresponding colony was re-streaked several rounds on YPDA and SC-Ura plates, which yielded a colony that was not able to grow on SC-Ura. The corresponding strain was named IMX672.1.

Strain MXFde 0.2: The *S. cerevisiae* strain TSINO93 (FdeR-based reporter strain)[31] has a CEN.PK background with $Pro_{TDH3}$-FdeR-URA3 and $Pro_{CYC1-FdeO}$-GFP-LoxP-HphMX-LoxP genes, and is auxotrophic for *LEU2* and *TRP1*. COMPASS employs a positive-selection scheme that involves auxotrophic and dominant marker genes. Therefore, TSINO93 needed to be auxotrophic for *HIS3*, *URA3*, and *LYS2*, in addition to *LEU2* and *TRP1*, and the *HphMX* dominant marker needed to be deleted from the strain. We employed the HI-CRISPR method for one-step multi-gene disruption[33]. To this end, a fragment-encoding a gRNA to target a sequence 133-bp downstream of the *URA3* start codon (annealed single stranded oligonucleotides URA3.A_for/URA3.A, Supplementary Fig. 9b, c) and pCOM002 plasmid (see above and Supplementary Fig. 9c) were cotransformed into strain TSINO93 to achieve the frame-shift mutation. Two-hundred microliters of a $10^4$-fold diluted cell culture were plated on YPDA plate with G418. After 2 days, a total of 50 colonies were randomly selected and each single colony was streaked out onto three different selective plates (i.e., SC-His, SC-Lys, and SC-Ura). After several rounds of restreaking single colonies onto new YPDA plates, we recovered a colony that was also not able to grow on YPDA medium with nourseothricin. After two more days, cells of a colony that did not grow on either of the three selective plates were streaked out on nonselective YPDA agar medium to eliminate the pCOM002 plasmid. The new strain was named MXFde 0.1. To generate strain MXFde 0.2, we employed β-estradiol (EST)-induced Cre recombinase[48] to remove the *HphMX* CDS flanked by *loxPsym* sites. Plasmid pCOM003 was constructed as described and transformed into MXFde 0.1 cells. An EST-Cre cell culture was plated on SC-Trp medium. After 3 days, single colonies were inoculated in liquid SC-Trp medium and grown in darkness for 6 h at 30 °C and 230 rpm, then induced with 2 μM β-estradiol (Sigma-Aldrich, Munich, Germany) and grown for another 24 h. Two-hundred microliters of a $10^4$-fold diluted cell culture were plated on YPDA plates containing 200 μg/ml hygromycin B. After several rounds of re-streaking single colonies onto new YPDA plates, we recovered a colony that was also not able to grow on YPDA medium with hygromycin B. The positive colony was checked by PCR for deletion of the *HphMX* CDS. After two more days, cells of a colony that did not grow on selective plates were streaked out on non-selective YPDA agar medium to eliminate the pCOM003 plasmid. The new strain was named MXFde 0.2.

**Construction of COMPASS plasmids.** All cloning steps are performed using overlap-based methods. To this end, regions homologous to neighboring fragments are included in the primers used to PCR amplify the DNA fragments to be joined.

Entry vector X: pGN003B[4] was digested with *Pme*I and religated to construct pCOMPASS01. Next, pL0A_0_1[32] was cut with *Asc*I/*Fse*I and used in an assembly reaction using PCR amplified $Pro_{TEF1}$-LacI-$Ter_{ADH1}$ (primers LAC_for/LAC_rev, on pCOMPASS01) to generate pCOMPASS02. *Not*I-digested pCOMPASS02 was used in an assembly reaction using PCR amplified $Pro_{TRP1}$ (primers PTRP_for/PTRP_rev, on BY4741 genomic DNA). Thereby, *Bam*HI and *Sal*I sites were introduced downstream of $Pro_{TRP1}$, while *Pac*I site was introduced upstream of $Pro_{TRP1}$. The resulting plasmid was called pCOMPASS03. The pCOMPASS03 plasmid was digested with *Sbf*I, treated with T4 DNA polymerase to remove 3′ overhang end and religated. The resulting plasmid was called pCOMPASS04. *Bam*HI/*Sal*I-digested pCOMPASS04 was used in an assembly reaction with

annealed single-stranded oligonucleotides CYCM_for/CYCM_rev introducing *MCS* and *X0*, the last 30 bp of $Pro_{CYC1mini}$, between $Pro_{TRP1}$ and *E. coli_{ori}*. The resulting plasmid was called Entry vector X.

Acceptor vectors A–D: To construct the first set (Set 1) of Acceptor vectors containing auxotrophic selection markers, *Bam*HI/*Not*I-digested pCOMPASS04 was used in an assembly reaction using PCR-amplified DNA parts as follows: (i) pL0A_0_1[32] was cut with *Not*I. A two-way Gibson cloning of $Pro_{KanaR}$ (primers PKANA_for/PKANA_rev, on pCR4-topo, Invitrogen, Karlsruhe, Germany), Amp[R] (primers AMP_for/AMP_rev, on pGN005B[4]) and $Ter_{KanaR}$ (primers TKANA_for/TKANA_rev, on pCR4-topo, Invitrogen) was done. The resulting plasmid was called pCOMPASS05. $HIS3$ and $Ter_{HIS3}$ (primers HISTERA_for/HISTERA_rev, on pGN005B[4]), $Pro_{HIS3}$ (primer PHIS_for/PHIS_rev, on pGN005B[4]), $Amp^R$ and $Ter_{KanaR}$ (primers AMPTER_for/AMPTER_rev, on pCOMPASS05) to result in Acceptor vector A. (ii) $LEU2$ and $Ter_{LEU2}$ (primers LEUTER_for/LEUTER_rev, on pGAD424, TAKARA Bio, GenBank #U07647), $Pro_{LEU2}$ (primer PLEU_for/PLEU_rev, on pGAD424, TAKARA Bio, GenBank #U07647), $Cm^R$ and $Ter_{CmR}$ (primers CMRTER_for/CMRTER_rev, on pLD_3_4[32]) to result in Acceptor vector B. (iii) $TRP1$ and $Ter_{TRP1}$ (primer TRPTERC_for/TRPTERC_rev, on pGN003B[4]), $Pro_{TRP1}$ (primers PTRP_for/PTRP_rev, on pGN003B[4]), $TCS^R$ and $Ter_{TCSR}$ (primers TCSRTER_for/TSCRTER_rev, on pF2, Addgene #42520) to result in Acceptor vector C. (iv) $LYS2$ and $Ter_{LYS2}$ (primers LYSTERA_for/LYSTERA_rev, on pYC6Lys-TRP1URA3, Addgene #11010), $Pro_{LYS2}$ (primers PLYS_for/PLYS_ter, on pYC6Lys-TRP1URA3, Addgene #11010), $Gen^R$ and $Ter_{GenR}$ (primers GENTER_for/GENTER_rev, on pDEST321) to result in Acceptor vector D.

Destination vector I: pCOMPASS03 was digested with *Pac*I and *Cla*I to remove $Pro_{URA3}$-URA3-$Ter_{URA3}$ from downstream (right) of *E. coli_{ori}*. The ~5-kb vector was then used in assembly reaction using PCR-amplified fragment (primers MURA_for/MURA_rev, on pCOMPASS03). The resulting plasmid was called pCOMPASS06. To construct pCOMPASS07, *Not*I/*Pac*I-digested pCOMPASS06 was used in an assembly reaction using PCR-amplified $Pro_{URA3}$ (primers URATERI_for/URATERI_rev, on pCOMPASS04) and PCR-amplified URA3-$Ter_{URA3}$ (primer PURAI_for/PURAI_rev, on pCOMPASS04). A *Not*I site was introduced between $Pro_{URA3}$ and the URA3 CDS, while *Bam*HI and *Pac*I sites were inserted downstream of the *X0* site. To construct pCOMPASS08, *Bam*HI/*Pac*I-digested pCOMPASS07 was used in an assembly reaction using $Spect^R$-$Ter_{SpectR}$ (primers SPECTERI_for/SPECTTERI_rev, on pCR8/GW/TOPO, TOPO Cloning Kit, TAKARA). Thereby, seven nucleotides were removed from upstream of $Spect^R$ and a *Pac*I site was inserted downstream of $Ter_{SpectR}$. Moreover, *Bam*HI/*Sal*I-digested pCOMPASS04 was used in a two-way assembly reaction using PCR-amplified DNA parts as follows: (i) $LYS2$ and $Ter_{LYS2}$ (primers LYSTERI_for/LYSTERI_rev, on pYC6Lys-TRP1URA3, Addgene #11010), and annealed single-stranded oligonucleotides LYSX0_for/LYSX0_rev to result in pCOMPASS09. Thereby, the *X1* sequence, *Asc*I site, and *Y4* were introduced between $LYS2$ CDS and *Pac*I site. (ii) $TRP1$ and $Ter_{TRP1}$ (primer TRPTERI_for/TRPTERI_rev, on pGN003B[4]), and annealed single-stranded oligonucleotides TRPX0_for/TRPX0_rev to result in COMPASS11. Thereby, *X1* sequence, *Sfi*I and *PI-Psp*I sites, and *Y3* were introduced between $LYS2$ CDS and *Pac*I site. (iii) $LEU2$ and $Ter_{LEU2}$ (primers LEUTERI_for/LEUTERI_rev, on pGAD424, TAKARA Bio, GenBank #U07647), and annealed single-stranded oligonucleotides LEUX0_for/LEUX0_rev to result in COMPASS12. Thereby, *X1* sequence, *Fse*I and *I-Sce*I sites, and *Y2* were introduced between $LEU2$ CDS and *Pac*I site. (iv) $HIS3$ and $Ter_{HIS3}$ (primers HISTERI_for/HISTERI_rev, on pGN005B[4]) and annealed single-stranded oligonucleotides HISX0_for/HISX0_rev introducing HR to result in COMPASS13. Thereby, *X1* sequence, *Sbf*I and *I-Ceu*I sites, and *Y1* were introduced between the *HIS3* CDS and the *Pac*I site.

Next, to construct pCOMPASS13.1, *Pac*I-digested pCOMPASS08 was used in an assembly reaction with PCR-amplified fragment $Ter_{HIS3}$-HIS3-$X1$-I-CeuI-$Y1$ (primers HIS_for/HIS_rev, on COMPASS13). To construct pCOMPASS14, *Fse*I/*Pac*I-digested pCOMPASS13 was used in an assembly reaction with PCR-amplified fragment $Ter_{LEU2}$-LEU2-$X1$-FseI-I-SceI-$Y2$ (primers LEU_for/LEU_rev, on pCOMPASS11). To construct pCOMPASS15, *Asc*I/*Pac*I-digested COMPASS13.1 was used in an assembly reaction using PCR-amplified fragment $Ter_{TRP1}$-TRP1-$X1$-SfiI-PI-PspI-$Y3$ (primers TRP_for/TRP_rev, on pCOMPASS10). To construct Destination vector I, *Pac*I-digested pCOMPASS14 was used in an assembly reaction with PCR-amplified fragment $Ter_{TRP1}$-TRP1-$X1$-SfiI-PI-PspI-$Y3$-$Ter_{LYS2}$-LYS2-$X1$-AscI-$Y4$ (primers TRPI_for/LYSI_rev, on pCOMPASS15). To construct Destination vector I.1, annealed single-stranded oligonucleotide LYP_for/LYP_rev was digested with *Pac*I and ligated into *Pac*I-digested Destination vector I. Through this, a 45-bp LHR-*Pme*I-*Aat*II-45 bp R_HR fragment was introduced into the plasmid which allows integration of Destination vector I.1 digested with either *Pme*I or *Aat*II into the *LYP1.x*[33] site of the yeast genome.

Acceptor vectors E–H: To construct the second set (Set 2) of Acceptor vectors, containing dominant resistance markers, *Bam*HI/*Not*I-digested pCOMPASS04 was used in a three-way assembly reaction using PCR-amplified DNA parts as follows: (i) *Nat1* and $Ter_{FBA1}$ (primers NATRER_for/NATRER_rev, on pAG36, Addgene #35126), $Pro_{FBA1}$ (primers PFBA_for/PFAB_rev, on pAG36, Addgene #35126) and $Amp^R$ and $Ter_{AmpR}$ (primers AMPTERE_for/AMPTERE_rev, on pCOMPASS05) to result in pCOMPASS16. Next, pCOMPASS16 was digested with *Pci*I and *Nde*I to delete 264-bp from the *URA3* encoding sequence. The remaining ~6-kb fragment was used in an assembly reaction using annealed single-stranded oligonucleotides

DURA_for/DURA_rev. The resulting plasmid was named Acceptor vector E. (ii) *Ble* and *Ter_ble* (primers BLERTER_for/BLERTER_rev, on pCEV-G1-Ph, Addgene #46814), *Pro_PGK1* (primer PGKF_for/PGKF_rev, on *pCEV-G1-Ph*, Addgene #46814), and *Cm^R* and *Ter_CmR* (CMRTERF_for/CMRTERF_rev, on pLD_3_4[32]) to result pCOMPASS17. Next, pCOMPASS17 was digested with *Pci*I and *Nde*I to delete 273-bp from the *URA3* encoding sequence. The remaining ~6-kb fragment was used in an assembly reaction using annealed single-stranded oligonucleotides DURA_for/DURA_rev. The resulting plasmid was named Acceptor vector F. (iii) *Neo* and *Ter_neo* (primers KANARTER_for/KANARTER_rev, on pCEV-G2-Km, Addgene #46815), *Pro_TDH3* (primers TDHG_for/TDHG_rev, on pCEV-G2-Km, Addgene #46815), *TCS^R* and *Ter_TCSR* (primers TCSRGTER_for/TSCRGTER_rev, on pF2, Addgene #42520) to result in pCOMPASS17. Next, pCOMPASS18 was digested with *Pci*I and *Nde*I to delete 273-bp from the *URA3* encoding sequence. The remaining ~6-kb fragment was used in assembly reaction using annealed single-stranded oligonucleotides DURA_for/DURA_rev. The resulting plasmid was named Acceptor vector G. (iv) *Hph* and *Ter_hph* (primers HPHRTER_for/HPHRTER_rev, on pHIS3p:mRuby2-Tub1 + 3′UTR::HPH, Addgene #50633), *Pro_TEF1* (primers PTEFH_rev/PTEFH_rev, pHIS3p:mRuby2-Tub1 + 3′UTR::HPH, Addgene #50633), *Gen^R* and *Ter_GenR* (primers GENTERH_for/GENTERH_rev, TAKARA Bio) to result in pCOMPASS19. Next, pCOMPASS19 was digested with *Aci*I and *Cla*I to delete 974-bp including *Pro_URA3* and the first 749-bp of the *URA3* CDS. The remaining ~5.5-kb fragment was used in an assembly reaction using annealed single-stranded oligonucleotides DDURA_for/DDURA_rev. The resulting plasmid was named Acceptor vector H.

Destination vector II was constructed in the following way: (i) *Not*I/*I-Ceu*I-digested pCOMPASS12 was used in an assembly reaction with PCR-amplified *nat1* and *Ter_nat1* (primers NATRERII_for/NATRERII_rev, on pAG36, Addgene #35126) to result in pCOMPASS20. Thereby, *X1* sequence, *Sbf*I and *I-Ceu*I sites, and the *Y1* sequence were introduced between the *nat1* CDS and the *Pac*I site. (ii) *Fse*I/*Not*I-digested pCOMPASS11 was used in an assembly reaction with PCR-amplified *ble* and *Ter_ble* (primers BLERTERII_for/BLERTERII_rev, on *pCEV-G1-Ph*, Addgene #46814) to result in pCOMPASS21. Thereby, *X1* sequence, *Fse*I and *I-Sce*I sites, and *Y2* were introduced between the *ble* CDS and the *Pac*I site. (iii) *PI-Psp*I/*Not*I-digested pCOMPASS10 was used in an assembly reaction with PCR-amplified *neo* and *Ter_neo* (primers KANARTERII_for/KANARTERII_rev, on pCEV-G2-Km, Addgene #46815) to result in pCOMPASS22. Thereby, *X1* sequence, *Sfi*I and *PI-Psp*I sites, and *Y3* were introduced between the *neo* CDS and the *Pac*I site. (iv) *Asc*I/*Not*I-digested pCOMPASS09 was used in an assembly reaction with PCR-amplified *hph* and *Ter_hph* (primers HPHRTERII_for/HPHRTERII_rev, on pHIS3p: mRuby2-Tub1 + 3′UTR::HPH, Addgene #50633) to result in pCOMPASS23. Thereby, *X1*, *Asc*I sites, and *Y4* were introduced between the *Hph* CDS and the *Pac*I site.

Next, pCOMPASS06 was digested using *Not*I and *Pac*I to remove *Pro_Trp1*. The remaining ~5.2-kb fragment was used in an assembly reaction with PCR-amplified Spect^R and *Ter_SpectR* (primers SPECTII_for/SPECTII_rev, on pCOMPASS08) to result in pCOMPASS24. *Pac*I-digested pCOMPASS24 was used in an assembly reaction with PCR-amplified fragment *nat1*-*Ter_nat1*-*X1*-*Sbf*I-*I-Ceu*I-*Y1* (primers NATII_for/NATII_rev, on COMPASS21). The resulting plasmid was called pCOMPASS25. To construct pCOMPASS26, *Fse*I/*Pac*I-digested pCOMPASS25 was used in an assembly reaction with PCR-amplified fragment *Ble*–*Ter_ble*–*X1*–*Fse*I-*I-Sce*I–*Y2* (primers BLEII_for/BLEII_rev, on COMPASS22). To construct pCOMPASS27, *Fse*I/*Pac*I-digested pCOMPASS25 was used in an assembly reaction with PCR-amplified fragment *hph*–*Ter_hph*–*X1*-*Asc*I–*Y3* (primers HPHRII_for/HPHII_rev, pCOMPASS26). To construct pCOMPASS28, *Pac*I-digested pCOMPASS27 was used in an assembly reaction with PCR-amplified fragment *Hph* – *Ter_hph*–*Asc*I–*Y3*-neo–*Ter_neo*–*X1*-*Sfi*I-*PI-Psp*I–*Y4* (HPHRRII_for/HKANARII_rev, on COMPASS27). *Not*I-digested pCOMPASS28 was used in an assembly reaction with annealed single-stranded oligonucleotides RE_for/RE_rev. Through this, *Bam*HI and *Xho*I site were introduced between the *X0* sequence and the Spect^R CDS. The resulting plasmid was called pCOMPASS29. *Not*I-digested pCOMPASS29 was used in an assembly reaction with PCR-amplified *Pro_TEF1*–blp^R–*Ter_TEF1* (BLPR_for/BLPR_rev, on pAG31, Addgene #35124). Through this, a *Not*I site was introduced between the blp^R encoding fragment and the *X0* sequence. The resulting plasmid was called pCOMPASS30. Finally, oligonucleotides ADE2A_for and ADE2A_rev were annealed and the resulting double-strand oligonucleotide was digested with *Pac*I and ligated into *Pac*I-digested pCOMPASS30. The resulting plasmid was called Destination vector II. Through this, 45-bp *LHR*–*Pme*I-*Pci*I-45-bp *RHR* were introduced into the plasmid to allow integration of Destination vector II, after digestion with either *Pme*I or *Pci*I, into the *ADE2.a*[33] site of the yeast genome.

**Construction of the ATF/BS library.** We selected three JUB1-, two ANAC102-, two ATAF1-, one RAV1-, and one GRF7-derived ATFs. Coding sequences of ATFs were obtained by PCR using appropriate expression plasmids[4] as templates and the respective forward (ATF-for) and reverse (ATF-rev) primers (see Supplementary Methods). The corresponding BSs (*JUB1 2×*, *JUB1 4×*, *ANAC102 4×*, *ATAF1 2×*, *RAV1 4×*, and *GRF7 4×*) fused upstream to the yeast minimal *CYC1* promoter were obtained by PCR using appropriate reporter plasmids[4] as templates and the respective forward (BS-for) and reverse (BS-rev) primers (Supplementary Methods). Both fragments were inserted into Entry vector X previously digested with

*Fse*I/*Asc*I. Constructs containing the ATFs NLS-GAL4AD-RAV1 and NLS-GAL4AD-GRF7 (both in combination with four copies of their BSs upstream of the minimal *CYC1* promoter)[4] were cloned into the Entry vector X by standard overlap-based cloning. The remaining ATF/BS combinations were assembled using a combinatorial approach (see Results). For JUB1, three expression plasmids containing NLS-JUB1-GAL4AD, NLS-DBD_JUB1-GAL4AD, and NLS-JUB1-EDLLAD-EDLLAD coding sequences were mixed in 1:1:1 molar ratio, and two reporter plasmids harboring two and four copies of the JUB1 BS, respectively, were mixed in 1:1 molar ratio. Overlap-based cloning results in different combinations between the three ATFs and the two BSs in the Entry vector, including the three desired combinations. For ATAF1, DNA fragments encoding two ATAF1-derived ATFs were mixed in 1:1 molar ratio and assembled by overlap-based cloning with a promoter fragment containing two copies of the ATAF1 BS. For ANAC102, DNA fragments encoding two ANAC102-derived ATFs were mixed in 1:1 molar ratio and assembled by overlap-based cloning with a promoter fragment containing four copies of the ANAC102 BS. In this way, nine different Entry vectors X derivatives harboring the different ATF/BS regulator modules were generated. The desired constructs were identified by colony PCR followed by sequencing (ATF: primers EXSEQ-for and EXSEQ-rev; BS-*Pro_CYC1_mini* and *Pro_TDH3*: PROSEQ-for and PROSEQ-rev). Primer sequences are given in Supplementary Data 15.

**Assembly of pathway genes.** β-Carotene and β-ionone: The promoters of the *E. coli* selection marker genes, yeast terminators, relevant parts of the CDSs of the β-carotene (*BTS1* or *GGPPSbc*, codon optimized *McrtI* and *McrtYB*) and β-ionone (*RiCCD1*) biosynthesis genes were amplified from different sources of genomic DNA or plasmids. Entry vector X was digested with *Fse*I and *Asc*I and three fragments including the CDS of the gene of interest, the yeast terminator and the promoter of the *E. coli* selection marker were inserted. The primers used for amplification included overhangs with rare RE recognition sites compatible to the appropriate Acceptor vector for the next cloning steps. All parts were verified by sequencing.

Naringenin: The promoters of the *E. coli* selection marker genes, relevant parts of the CDSs of the NG biosynthetic pathway (*AtC4H:L5:AtATR2*, *PhCHI*, *HaCHS*, *At4CL-2*, and *AtPAL-2*) fused to the terminators were amplified from different sources of plasmids and mixed in equimolar ratio. The Acceptor vectors were digested with *Fse*I and *Asc*I (Acceptor vector E–H) or *Xho*I and *Bam*HI (Destination vector II) and mixed in equimolar ratio. The combined fragments containing the CDSs, promoters of the *E. coli* selection marker genes and the digested plasmids were mixed in a single tube in a ratio recommend by the manufacturer to perform the NEBuilder HiFi reaction. We plated the transformed cells onto four different LB agar media containing either spectinomycin, ampicillin, chloramphenicol, or triclosan as selection markers for the transformed plasmids. Thereby, *AtC4H::L5::AtATR2*, *PhCHI*, *HaCHS*, *At4CL-2*, and *AtPAL-2* were assembled in Destination vector II, Accepter vector E–H, respectively. All parts, their functions, and sources are given in Supplementary Data 5.

**Combinatorial expression of each single gene of pathways.** β-Carotene and β-ionone: Nine PCR-amplified ATF/BS fragments (primers X0_for and Z0_rev, PCR performed on the Entry vectors-nine ATF/BS) were mixed (see Supplementary Methods). The *McrtI*, *BTS1*, *McrtYB*, and *RiCCD1* coding sequences and their downstream terminators and the promoters of the *E. coli* selection marker genes were PCR-amplified from Entry vectors using appropriate pairs of primers (Supplementary Methods). Four vectors were digested: Destination vector I (*Sal*I/*Eco*RI) and Acceptor vector A–C (*Fse*I/*Asc*I), and mixed in equimolar ratio. The combined ATF/BS fragments, the fragments containing the CDSs, and the digested plasmids were mixed in a single tube in a ratio recommend by the manufacturer to perform the NEBuilder HiFi reaction. We plated the transformed cells onto four different LB agar medium containing either spectinomycin, ampicillin, chloramphenicol, or triclosan as selection markers for the transformed plasmids. Cells harboring the successfully assembled constructs are able to grow and form colonies on appropriate selection medium. Hence, *McrtI*, *BTS1*, *McrtYB*, and *RiCDD1* were assembled in Destination vector I and Acceptor vectors A–C, respectively. Moreover, the *GGPPSbc* coding sequence and its downstream terminator and the promoters of the *Amp^R* selection marker were PCR-amplified from Entry vector X using appropriate pairs of primers and mixed in equimolar amounts. The combined ATF/BS fragments, the fragment containing the *GGPPSbc* CDS, and the *Fse*I/*Asc*I-digested Acceptor vector A were mixed in a single tube in a ratio recommend by the manufacturer to perform the NEBuilder HiFi reaction. We plated the transformed cells onto LB agar medium containing ampicillin allowing cells harboring the successfully assembled constructs to grow. The constitutive yeast TDH3 promoter[32] was used as a positive control in all experiments. PCR-amplified *Pro_TDH3* (TDH_for/TDH_rev, on on BY4741 genomic DNA) was cloned in *Fse*I/*Asc*I-digested Entry vector X (Supplementary Data 6).

**Combinatorial expression of genes of the naringenin pathway.** Nine PCR-amplified ATF/BS fragments were mixed (see Supplementary Methods). The *AtC4H:L5:AtATR2*, *PhCHI*, *HaCHS*, *At4CL-2*, and *AtPAL-2* coding sequences and their downstream terminators and the promoters of the *E. coli* selection marker genes were PCR-amplified from Destination vector II and Acceptor vectors E–H,

respectively (Supplementary Methods). Five digested vectors, Destination vector II (XhoI/BamHI) and Acceptor vectors E–H (FseI/AscI), the ATF/BS fragments, and the fragments containing the naringenin biosynthesis CDSs were mixed in a single tube in a ratio recommend by the manufacturer to perform the NEBuilder HiFi reaction. We plated the transformed cells onto five different LB agar media containing either spectinomycin, ampicillin, chloramphenicol, triclosan, or gentamicin. Cells with successfully assembled constructs are able to grow on the appropriate selection medium. Thereby, AtC4H:L5:AtATR2, PhCHI, HaCHS, At4CL-2, and AtPAL-2 were assembled in Destination vector II and Acceptor vectors E–H, respectively (Supplementary Data 5).

**Combinatorial cloning of pathway genes.** Equal amounts of the nine Destination vectors I–McrtI were mixed and digested with I-CeuI/SbfI. Equal amounts of the nine Acceptor vectors A-BTS1 were mixed and used as a template to amplify fragments containing the promoter of the HIS3 auxotrophic marker, the IPTG-inducible promoter, the ATF/BS fragments, the BTS1 CDS, the TDH3 terminator, the promoter and CDS of the ampicillin selection marker using respective forward (X0_for) and reverse (Z2_rev) primers. Therefore, PCR-amplified fragments contain nine different modules differing in their ATF/BS units (see Supplementary Methods). Using TAR, the PCR-amplified fragments were inserted into the nine Destination vectors I-McrtI to generate a library of $9^2 = 81$ Destination vectors I-McrtI-BTS1. Yeast cells with successful constructs grow on SC-Ura/-His medium. The plasmid library was recovered from the yeast cells, transformed into E. coli, and grown on LB agar plates containing ampicillin, where only cells harboring correct assemblies will grow. In the next step, the plasmid library was digested with FseI/I-SceI. Equal amounts of the nine Acceptor vectors B-McrtYB were mixed and used as a template for a PCR reaction to amplify fragments containing the promoter of the LEU2 auxotrophic marker, the IPTG-inducible promoter, the ATF/BS fragments, the MctYB CDS, yeast synthetic terminator 3, the promoter and CDS of the chloramphenicol selection marker using respective forward (X0_for) and reverse (Z3_rev) primers. Therefore, PCR-amplified fragments contain nine different modules differing in their ATF/BS units. Using TAR, the PCR-amplified McrtYB modules were inserted into the linearized Destination vector I-McrtI-BTS1 library to generate a new library of $9^3 = 729$ Destination vectors I-McrtI-BTS1-McrtYB. Yeast cells with successful constructs grow on SC-Ura/-His/-Leu medium. The library of Destination vector I containing all three genes leads to the production of β-carotene in inducing media. The plasmid library was recovered from the yeast cells and transformed into E. coli, which was selected on LB agar medium containing chloramphenicol. The recovered library was subsequently used to integrate the pathway genes including the ATF/BS regulatory modules into the yeast genome (see below).

Plasmids producing the highest amounts of β-carotene in each background strain, IMX672.1 and Gen 0.1, were subsequently used for β-ionone production. The plasmids were digested with PI-PspI. Equal amounts of the nine Acceptor vectors C-RiCCD1 were mixed and used as a template to amplify fragments containing the promoter of the TRP1 auxotrophic marker, the IPTG-inducible promoter, the ATF/BS regulatory units, the RiCCD1 CDS, the yeast TEF1 terminator, and the promoter and CDS of the triclosan selection marker using respective forward (X0_for) and reverse (Z4_rev) primers. Therefore, PCR-amplified fragments contain nine different modules differing in their ATF/BS regulatory modules. Using TAR, the PCR-amplified fragments were inserted into the Destination vector I-McrtI-BTS1-McrtYB. Yeast cells with correctly assembled constructs grow on SC-Ura/-His/-Leu/-Trp medium. The library of Destination vector I containing all four genes leads to the production of β-ionone. The plasmid library was recovered from the yeast cells, transformed into E. coli, and grown on LB agar medium containing triclosan. Plasmids pCAROTENE-PTDH3 and pIONONE-PTDH3 (see next chapter) contain the β-carotene and β-ionone CDSs under the control of the TDH3 promoter assembled in Destination vector I.

**Positive control for β-carotene and β-ionone production.** Destination vector I was digested with EcoRI/SalI and used in a two-way assembly reaction using PCR-amplified $Pro_{TDH3}$ sequence (PROTDHMCRTI_for/PROTDHMCRTI_rev, on pCP11[32]) and McrtI CDS fussed to $Ter_{FBA1}$ (MCRTIPOS_for/MCRTIPOS_rev, on Entry vector X-McrtI) to generate pMCRTI_PTDH3. PciI/ClaI-digested pCP11[32] was used in an assembly reaction using HIS3 encoding sequence (PCPHIS_for/PCPHIS_rev, on pGN005B[4]) to generate pCOMPASS32. Next, NotI-digested pCOMPASS32 was used in an assembly reaction using BTS1 CDS (BTSPOS_for/BTSPOS_rev, on Entry vector X-BTS1) to generate pBTS1_PTDH3. PciI/ClaI-digested pCP11[32] was used in an assembly reaction using PCR-amplified LEU2 encoding sequence (PCPLEU_for/PCPLEU_rev, on pGAD424, TAKARA Bio, GenBank #U07647) to generate pCOMPASS33. Next, NotI-digested pCOMPASS33 was used in an assembly reaction using McrtYB CDS (MCRTYBPOS_for/MCRTYBPOS_rev, on Entry vector X-McrtYB) to generate pMCRTYB_PTDH3. Acceptor vector C was digested with FseI/AscI and used in a two-way assembly reaction using PCR-amplified $Pro_{TDH3}$ sequence (PROTDHRICCD_for/PROTDHRICCD_rev, on pCP11[32]) and RiCCD1 CDS-$Ter_{TEF2}$-PI-PspI-$Pro_{TCSR}$ (RICCDPOS_for/RICCDPOS_rev, on Entry vector X-RiCCD1) to generate pRICCD1_PTDH3. I-CeuI- digested pMCRTI_PTDH3 was used in an assembly reaction using PCR-amplified $Pro_{TDH3}$-BTS1-$Ter_{TDH3}$ (PTDH3_BTS1_for/PTDH3_BTS1_rev, on pBTS1-PTDH3) and AmpR encoding sequence

(AMPR_for/Y1_rev, on pGN003B[4]). The resulting plasmid was called pCOMPASS34. Then, FseI/I-SceI-digested pCOMPASS33 was used in an assembly reaction using PCR-amplified $Pro_{TDH3}$–McrtYB (PTDH3_MCRTYB_for/PTDH3_MCRTYB_rev, on pMCRTYB-PTDH3) and $Ter_{SYN3}$-CmR-$Ter_{CmR}$ (CMR_for/Y2_rev, on pLD_3_4[32]). The resulting plasmid was called pCAROTENE-PTDH3. Then, PI-PspI-digested pCOMPASS34 was used in an assembly reaction using PCR-amplified $Pro_{TDH3}$–RiCCD1-$Ter_{TEF2}$-TCSR–$Ter_{TCSR}$ (PTDH3_RICCD1_for/Y3_rev, on pRICCD1_PTDH3.). The resulting plasmid was called pIONONE-PTDH3.

**Integration of Destination vectors into the yeast genome.** Strains IMX672.1 or Gen 0.1 were transformed with the library of BamHI- or NotI-linearized Destination vectors I resulting in cassette integration into the yeast LYS2 or URA3 loci, respectively, thereby leading to the integration of the β-carotene or β-ionone pathway genes.

**Multilocus integration of pathway genes.** To simultaneously integrate the genes required for β-carotene (McrtI, BTS1, and McrtYB) or β-ionone (McrtI, BTS1, McrtYB, and RiCCD1) production, strains Gen 0.1 and IMX672.1 were cotransformed with either 1 µg of sgRNA plasmids pTAJAK-92[45] and pCOM005 (to integrate McrtI, McrtYB and BTS1, and into the X-3, XI-3, and XII-5 locus, respectively) or pCOM006 (to integrate the RiCCD1 containing module into the ura3-52 locus), plus 1 µg of each donor fragment: McrtI (primers DES1-X3_for and DES1-X3_ on Destination vector I-McrtI), BTS1 (primers ACCEPTA-XII5_for and ACCEPTA-XII5_rev on Acceptor vector A-BTS1), McrtYB (primers ACCEPB-XI3_for and ACCEPTB-XI3_rev on Acceptor vector B-McrtYB), and RiCCD1 (primers ACCEPTC-ura_for and ACCEPTC-ura_rev on Acceptor vector C-RiCCD1). To select for β-carotene producing strains, cells were inoculated in 1 ml liquid culture medium for 3 days, followed by plating 1 ml of $10^2$-diluted cells on media that selected for the presence of the sgRNA and Cas9 plasmids pTAJAK-92[45] and pCOM005 which harbor the G418 and LYS2 selection marker genes, respectively. Colonies were washed off the plate(s) and subsequently cells were plated on selective medium (SC-Leu/-Ura/-His) to screen for β-carotene producing colonies with integrated selection markers. To select for β-ionone pathway strains, cells were plated on SC-Lys to select for the presence of plasmid pCOM006. Colonies were washed off and plated on selective medium to screen for β–ionone producing colonies with integrated selection markers. When colonies appeared, the transformation plates were replicated on non-selective induction plates. Multiplex PCR was performed on single colonies; primers amplified ATF/BS regulatory units upstream of each CDS.

**Multilocus integration of pathway genes in strain MXFde 0.2.** Strain MXFde 0.2 was co-transformed with pCOM006 and 1 µg of RiCCD1 donor (primers ACCEPTC-ura_for and ACCEPTC-ura_rev on Acceptor vector C-RiCCD1) to integrate the RiCCD1 containing module into the ura3-52 site. To select for successful integration, cells were plated on SC-Lys/-Trp, as the LYS2 marker is encoded on the plasmid and the TRP1 selection marker is present on the integrated donor. When colonies appeared, the transformation plates were replicated on nonselective induction plates. Next, to integrate five genes required for naringenin production, the library of β-ionone producing stains was cotransformed with 1 µg of sgRNA plasmid pCOM004 (to integrate PhCHI and HaCHS into the X-3 and XI-2 locus, respectively), pCOM007 (to integrate AtC4H::L5::AtATR2 into XII-2), pCOM008 (to integrate At4CL-2 and AtPAL-2 into the X-2 and XI-5 locus, respectively), plus 1 µg of each donor fragment: PhCHI (primers ACCPTE-X3_for and ACCPTE-X3_rev on Acceptor vector E-CHI), HaCHS (primers ACCPTF-XI2_for and ACCPTF-XI2_rev on Acceptor vector F-CHS), AtC4H::AtATR2 (primers DESII-XII2_for and DESII-XII2_rev on Destination vector II-C4H::ATR2), At4CL-2 (primers ACCPTG-X2_for and ACCPTG-X2_rev on Acceptor vector G-4CL-2), and AtPAL-2 (primers ACCEPTH-XI5_for and ACCEPTH-XI5_rev on Acceptor vector H-PAL2). Cells were inoculated in 1 ml liquid culture medium for 3 days, followed by plating 1 ml of $10^2$-diluted cells on media that selected for the presence of the sgRNA plasmids which encodes the LYS2 selection marker. When colonies appeared, the transformation plates were replicated on selective plates (YPDA with five dominant selection markers) to (i) remove the episomal plasmids and (ii) screen for successfully integrated genes into the genome. To simultaneously integrate the three CDSs (in addition to RiCCD1) required for β-ionone production in yeast, the library of naringenin strains was co-transformed with 1 µg of triple-sgRNA plasmid pCOM005 (to integrate McrtI, McrtYB, and BTS1 or GGPPSbc into the X-4, XI-3, and XII-5, loci, respectively), plus 1 µg of the McrtI, McrtYB, and BTS1 donor fragments: McrtI (primers DES1-X4_for and DES1-X4_rev on Destination vector I-McrtI), McrtYB (primers ACCEPB-XI3_for and ACCEPTB-XI3_rev on Acceptor vector B-McrtYB), and BTS1 (primers ACCEPTA-XII5_for and ACCEPTA-XII5_rev on Acceptor vector A-BTS1). Moreover, we used the GGPPSbc donor (primers ACCEPTA-XII5_for and ACCEPTA-XII5_rev on Acceptor vector A-GGPPSbc) in addition to the BTS1 donor. Cells were inoculated in 1 ml liquid culture medium for 3 d, followed by plating 1 ml of $10^2$-diluted cells on media that selected for the presence of the sgRNA plasmids. Subsequently, when colonies appeared, the cultures of the transformation plates were replicated on non-selective induction plates. When

colonies appeared, they were scraped of the plates and used to establish liquid-media cultures thereby generating a library of isolates (called Narions) producing β-ionone and naringenin at different levels.

**Recovery of plasmids from yeast**. Assembled constructs encoding the β-carotene and β-ionone pathways were recovered from yeast using a protocol for isolating high-quality circular plasmid DNA in microgram quantities[27]. Briefly, yeast cells harboring plasmids were grown in 10 ml selective medium overnight at 30 °C, then inoculated in 400 ml selective medium and grown again overnight at 30 °C. At an $OD_{600}$ of 1.5, cells were harvested by centrifugation and resuspended in 100 ml distilled water, transferred to two 50-ml Falcon tubes, and collected by centrifugation. Cells were lysed using SPE solution (1 M sorbitol, 0.01 M sodium phosphate, 0.01 M $Na_2EDTA$, pH 7.5), Zymolyase 20T (10 mg $ml^{-1}$ in 25% (w/v) glycerol), β-mercaptoethanol, lysis buffer (0.05 M Tris-HCl, 0.02 M EDTA, 1% SDS, pH 12.8), and phenol:chloroform:isoamyl alcohol. DNA was precipitated using sodium acetate and isopropanol and plasmid DNA was extracted using the Qiagen Large-Construct Kit (Qiagen). Recovered plasmids were transformed into *E. coli* cells and plated on LB plates with appropriate *E. coli* selection markers.

**Integration of nine ATF/BS-yEGFP modules into locus XII-5**. Nine Acceptor vectors A-ATF/BS-BTS1 (Supplementary Fig. 11 and Supplementary Data 6) were digested with *I-CeuI* to remove the $BTS1$-$Ter_{TDH3}$ fragment. The *I-CeuI*-digested $yEGFP$-$Ter_{CYC1}$ fragment (primers YEGFP_for and YEGFP_rev on pGN005B[4]) was assembled in each of the nine linearized backbones to generate nine Acceptor vectors (A-yEGFP1–A-yEGFP9). The nine Acceptor vectors were identified by PCR using primers YEGFP2-for and YEGFP2-rev, followed by sequencing. Moreover, fragments containing $TDH3$ promoter–$yEGFP$–$TDH3$ terminator (primers PTDH_for and TTDHX_rev on plasmid pCP11[32]) and promoters of $Amp^R$ selection marker genes (AMPR_for/Z1_rev, on Acceptor vector A-yEGFP1) were cloned into *AscI/FseI*-digested Acceptor vector A. The resulting plasmid was called Acceptor vector A-yEGFP0.

Next, 1 μg of each $yEGFP$ donor fragment (primers ACCEPTA-XII5_for and ACCEPTA-XII5_rev, on the ten Acceptor vectors A-yEGFP0–A-yEGFP9) together with $McrtI$ (primers DES1-X3_for and DES1-X3_rev, on Destination vector I-McrtI) and $McrtYB$ (primers ACCEPB-XI3_for and ACCEPTB-XI3_rev, on Acceptor vector B-McrtYB) donors and 1 μg of triple sgRNA plasmids pTAJAK-92[45] and pCOM005 (to integrate $McrtI$, $McrtYB$, $yEGFP$ into the X-3, XI-3, and XII-5 sites, respectively) were transformed into yeast strain IMX672.1. Cells were plated on media to select for the presence of the sgRNA plasmids. Colonies were washed off the plate(s) and subsequently cells were plated on selective medium (SC-Leu/-Ura/-His) to screen for successful integration of donors with integrated selection markers. Subsequently, when colonies appeared, the transformation plates were replicated on non-selective induction YPDA plates. PCR was performed on single colonies; primers were designed to amplify ATF/BS regulatory units upstream of each CDS.

**Construction of AGREEN, BRED, and AB-GREEN-RED strains**. Acceptor vector A-NLS-JUB1-EDLLAD-EDLLAD-4×-BTS1 and Acceptor vector B-NLS-JUB1-EDLLAD-EDLLAD-4×-McrtYB (Supplementary Fig. 11 and Supplementary Data 6) were digested with *I-CeuI* and *I-SceI* to remove the $BTS1$-$Ter_{TDH3}$ and $McrtYB$-$Ter_{SYN3}$ fragment, respectively. The *I-CeuI*-digested $AcGFP1$-$Ter_{TEF2}$ fragment (primers ACGFP1_for and ACGFP1_rev, on plasmid pGREEN) and the *I-SceI*-digested $DsRED$-$Ter_{TEF2}$ fragment (primers DSRED_for and DSRED_rev, on plasmid pRED) were inserted into the linearized Acceptor vector A and B to generate plasmids AVA-GREEN and AVB-RED, respectively. AVA-GREEN and AVB-RED were identified by PCR using primers ACGFP2-for and ACGFP2-rev and DSRED1-for and DSRED1-rev, respectively, followed by sequencing. Next, donors of $AcGFP1$ (primers ACCEPTAGFP-XII5_for and ACCEPTA-XII5_rev, on AVA-GREEN) and $DsRED$ (primers ACCEPB-XI3_for and ACCEPTB-XI3_rev, on AVB-RED) were PCR-amplified. One μg of sgRNA plasmid pCOM005 (to integrate $AcGFP1$ and $DsRED$ donors into the XII-5 and XI-3 locus, respectively) together with (i) $AcGFP1$, (ii) $DsRED$, or (iii) $AcGFP1$ and $DsRED$ donors were transformed into yeast strain IMX672.1. Cells were plated on media to select for the presence of the sgRNA plasmid which harbors the LYS2 marker gene. Washed-off colonies were subsequently plated on selective medium (SC-Leu/-His). When colonies appeared, the transformation plates were replicated on non-selective induction YPDA plates. PCR was performed on single colonies using primers amplifying ATF/BS regulatory units upstream of each CDS.

**Construction of strains for growth rate measurements**. To generate constructs suitable for green fluorescent protein (GFP) and red fluorescent protein (RFP) expression, we inserted a DNA fragment containing the LYS2 gene (primers LYS_for/LYS_rev; on pYC6Lys-TRP1URA3; Addgene #11010), the yeast $TEF2$ promoter (primers PTEF2_for/PTEF2_rev; BY4741 genomic DNA), the AcGFP1 (*Aequorea coerulescens* GFP, primers ACGFP_for/ACGFP_rev; plasmid AcGFP1-N1; Addgene #54705) or DsRED (*Discosoma sp.* RFP, primers DSRED_for/ DSRED_rev; plasmid MSCV-CMV-DsRed-IRES-d24EGFP; Addgene #41944) coding sequences, and the $TEF2$ terminator (primers TTEF2_for/TTEF2_rev; BY4741 genomic DNA), into the *XbaI/PstI*-digested plasmid pL1D_hc[32]. The

resulting plasmids, called pGREEN and pRED, were digested with *XhoI*. Digested pGREEN was then integrated into the *lys2.a* site of strains Gen 0.1 and IMX672.1, while digested pRED was integrated into the *lys2.a* site of strains Gen 0.1, IMX672.1, PCB, CB1, MXintpDI 1, and RC1. The fluorescent protein reporter strains were called Green-Gen 0.1, Green-IMX672.1, and Red-Gen 0.1, Red-IMX672.1, Red-PCB, Red-CB1, Red-MXintpDI 1, and Red-RC1, respectively (Supplementary Data 7). A single colony of AcGFP1- and DsRED-labeled strains was pre-inoculated in 4 ml of noninducing SC medium with appropriate yeast selection markers for 18 h at 30 °C in a rotary shaker at 230 rpm. The subcultures were grown in non-inducing (2% (w/v) glucose) or inducing (2% (w/v) galactose, 20 μM IPTG) YPDA media. Growth competition experiments in single vessels were prepared by mixing (i) DsRED-labeled cells, when the ATF is not expressed (non-inducing medium), and corresponding AcGFP1-labeled cells; (ii) DsRED-labeled cells, when the ATF is expressed (inducing medium), and corresponding AcGFP1-labeled cells. We inoculated co-cultures at a ratio of 8:2 (DsRED-labeled cells: AcGFP1-labeled cells) at $OD_{600}$ of ~0.1 in 4 ml YPDA medium. Cells were grown for 48 h in a rotary shaker at 30 °C and 230 rpm. At time points 6, 24, and 48 h, samples were removed for $OD_{600}$ measurement and analysis by flow cytometry to determine the ratio of DsRED-positive to AcGFP1-positive cells. At each time point, cells were diluted 100-fold into fresh liquid medium for growth until the next time point. We counted DsRED-and AcGFP1-labeled cells.

**Induction experiments**. Yeast strains harboring the pathway genes were plated on non inducing SC medium (2% (w/v) glucose) with appropriate selection markers. Cells were grown at 30 °C for 3–4 days. The plates were scraped to collect the cells which were then plated on inducing SC medium plates containing 20 mM isopropyl-β-D-thiogalactopyranoside (IPTG), 2% (w/v) galactose, 1% (w/v) raffinose and the appropriate selection markers (inducing and selective media) for episomal expression, and inducing YPDA medium plates (20 mM IPTG, 2% (w/v) galactose) in the case of genome integration-based expression. Cells were grown at 30 °C for 3–4 days. Episomal plasmids were isolated from the collected colonies, transformed into *E. coli* and the assembled parts were sequenced. In the case in genome-integrated cassettes, colony PCR was performed, followed by sequencing. Three independent colonies from either integrated or episomal constructs introduced in the Gen 0.1 or IMX672 backgrounds were chosen for product (β-carotene or β-ionone) analysis by HPLC. To this end, colonies were selected based on intense orange color, indicating high level of β-carotene level, or light orange color, indicating accumulation of β-ionone. Positive controls expressed the pathway genes under the control of the constitutive yeast $TDH3$ promoter.

**High-performance liquid chromatography**. Single colonies of yeast strains were inoculated into 4 ml noninducing SC medium with appropriate selection marker (pre-culture), and grown for 18–24 h at 30 °C in a rotary shaker at 230 rpm. The pre-cultures were then used to inoculate subcultures (50 ml) inducing SC medium (20 mM IPTG, 2% galactose, 1% (w/v) raffinose) with the appropriate selection marker. All flask cultures were inoculated from pre-cultures grown on the same medium, to an initial $OD_{600}$ of 0.1. Cells were grown in 500-ml flask at 30 °C for 3 days in a rotary shaker at 230 rpm to saturation and then harvested for HPLC analysis of metabolites. Carotenoid extraction was carried out from cellular pellets according to the acetone extraction method[16], with some modifications as follows. The cell pellet was washed once with deionized water. Glass beads (400–600 μm diameter; Sigma Chemical Co.) and 500 μl of acetone were added to the cell pellet in a 2.0-ml microcentrifuge tube (Eppendorf), and vortexed to pulverize the cells, followed by incubation for 20 min at 30 °C. After breakage, the bead-cell mixture was centrifuged in a table-top micro-centrifuge at 16,300×$g$ for 5 min at 4 °C, and the acetone supernatant was collected in standard 2.0-ml microcentrifuge tubes (Eppendorf). This extraction procedure was repeated until the cell pellet was white. For β-carotene samples, the combined acetone extracts were transferred to a glass vial and dried using a speed-vac. To prevent degradation of the carotenoids by light, the glass vials were kept inside a black colored 1.5-ml microcentrifuge tubes (Eppendorf). For the determination of β-ionone, half of the combined acetone extract was dried to quantify β-carotene as described above. The other half was kept inside a pharma glass inlay and was used for β-ionone measurement. All operations were carried out under green save light to avoid degradation of carotenoids. The samples were stored at −80 °C until further use. Extracts were obtained from cells grown in three independent experiments, and HPLC analyses were performed by AppliChrom (Oranienburg, Germany). The dry samples were solved in acetone before measurement. Carotenoids were separated by HPLC using a RP-HPLC phase AppliChrom OTU DiViDo (250 × 4.6 mm) column with porous 5-μm particles (85/10/5, acetone/methanol/isopropanol, v/v/v) as the mobile phase, with a 1.0-ml/min flux. The elution profiles were recorded using Shimadzu 450 nm (D2) and Kontron 300 nm (D2) detectors.

**Absorbance-based quantification of β-carotene**. Using the method reported by Lian et al.[21], β-carotene producing strains were pre-cultured in SC medium with appropriate selection markers for approximately 2 days, inoculated into 5 ml of both, inducing (2% (w/v) galactose, 20 μM IPTG, 1% (w/v) raffinose) and non-inducing (2% (w/v) glucose) SC medium with an initial $OD_{600}$ of 0.1 in 15-ml culture tubes, and cultured (30 °C, 250 rpm) for 5 further days. Stationery phase yeast cells were collected by centrifugation at 13,000×$g$ for 1 min and cell

precipitates were resuspended in 1 ml of 3 N HCl, boiled for 5 min, and cooled in an ice-bath for 5 min. The lysed cells were washed with ddH$_2$O and resuspended in 400 μl acetone to extract β-carotene. The cell debris was removed by centrifugation. The extraction step was repeated until the cell pellet appeared white. The β-carotene containing supernatant was analyzed for its absorbance at 454 nm. The production of β-carotene was normalized to the cell density.

**Flow cytometry analysis of naringenin producing cells**. To quantify the yEGFP fluorescence output in the absence of plant-derived ATFs, single colonies of FdeR-based reporter strains were inoculated into 500 μl noninducing SC medium with appropriate selection markers in 48-well deep-well plates. Plates were incubated for 24 h at 30 °C and 230 rpm in a rotary shaker. The precultures were used to inoculate subcultures in 500 μl inducing YPDA medium to an OD$_{600}$ ~0.1. Cells were grown for 16 h in a rotary shaker at 30 °C and 230 rpm. Protein production was inhibited by adding 500 μg/ml cycloheximide. Thereafter, fluorescence output of each cell was analyzed by a BD FACSCalibur flow cytometer (BD Biosciences). The yEGFP fluorescence values were calculated for a minimum of 10,000 cells in each sample. The yEGFP fluorescence geometric mean per cell was calculated using Flowing Software version 2.5.1 (http://www.uskonaskel.fi/flowingsoftware).

**Reporting summary**. Further information on research design is available in the Nature Research Reporting Summary linked to this article.

## Data availability

The relevant data are available from the corresponding author upon request. The source data underlying Figs. 5i, 6c–d, Supplementary Figs. 1, 2b, c, 12b–g, 13c, 14b, 15g, 16e, 17h, and Supplementary Data 10 are provided as a Source Data file. The COMPASS plasmids reported in this study are available from Addgene (www.addgene.org) under the following ID numbers. Entry vector X, 126939; Acceptor vector A, 126924; Acceptor vector B, 126930; Acceptor vector C, 126931; Acceptor vector D, 126932; Acceptor vector E, 126933; Acceptor vector F, 126934; Acceptor vector G, 126935; Acceptor vector H, 126936; Destination vector I, 126937; Destination vector I.1, 127542; Destination vector II, 126938.

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

## Acknowledgements

We are thankful to the following colleagues for providing plasmids or yeast strains: Yansheng Zhang (Chinese Academy of Sciences, Beijing, China) for plasmids harboring McrtI and McrtYB coding sequences; Jules Beekwilder (Plant Research International, Wageningen, The Netherlands) for plasmid RiCCD, and the yeast strains IMC167 and RiCCD1; Michael K. Jensen and Tim Snoek (Technical University of Denmark, Lyngby, Denmark) for plasmids pROP280, pROP266, pROP273, and pTAJAK-105 and strain TSINO93; Alex T. Nielsen (Technical University of Denmark, Lyngby, Denmark) for plasmid pTAJAK-92; Verena Siewers (Chalmers University of Technology, Göteborg, Sweden) for yeast strain SCIGS22a; and Ying-Jin Yuan (Tianjin University, Tianjin, China) for plasmid pRS425-PTDH3-ts-TPGK-PPGK-GGPPSbc-TCYC1. This work was funded by the Federal Ministry of Education and Research of Germany (BMBF; Grant numbers 031A172 and 031B0223, given to Katrin Messerschmidt), the European Union H2020 project PlantaSYST (SGA-CSA No. 739582 under FPA No. 664620), and a fellowship of the Potsdam Graduate School, University of Potsdam, given to G.N. All experiments were done in the Cell2Fab lab, led by K.M. HPLC analyses were ordered by K.M. and were performed by AppliChrom® (Oranienburg, Germany).

## Author contributions

G.N. conceived and designed the COMPASS strategy, performed the experiments, and analyzed the data. J.B. and L.R. helped to construct the naringenin, fluorescent reporter plasmids, and strains. B.M.R. supervised the research conducted by G.N. and provided expert knowledge. G.N. and B.M.R. wrote the paper. All authors take full responsibility for the content of the paper.

## Additional information

**Competing interests:** The authors declare no competing interests.

