## [Peer Review File · Nature Communications]

Reviewers' Comments:

Reviewer #1:

Remarks to the Author:

The novelty of this work is related with an inducible combinatorial system, called COMPASS, that allows to rapidly generate different constructs and express them in yeast, by-passing in this way the eventual growth inhibition by metabolic burden.

There is a tremendous amount of work to demonstrate the performance of the system, with production of β -carotene and β -ionone developed as proofs of concept.

I have several concerns and questions that are required to fully appreciate the benefits of the technique:

- The method employed for constructing the vectors is, in my opinion, similar to Golden Gate, a well known and widely used combinatorial method; therefore the system is not really new. The authors do not mention Golden Gate (GG) and do not compare their technique with this method. Furthermore, GG does not need to use PCR, and this technique profusely uses PCR.
- The system requires a construction step in E coli. In this sense, it looks to be more related with MoClo.
- Their inducible system is based on the expression of artificial transcription factors, (ATF) that control the activity of the heterologous genes to be expressed. The ATF constructs are very long, therefore, the expression of these ATFs might also generate an extra metabolic demand for the cell. There is no reference in the text to this subject.
- In order to fully appreciate the performance of COMPASS, compared to other published well known methods, I propose that the authors report the growth titers obtained with their system - not only yields; and compare them with those achieved in an inducible system for comparison, not only with a constitutive system, where the metabolic burden might be higher.
- The technique requires several selection markers (antibiotic resistance genes), that are left in the vector and therefore integrate in the yeast genome.
- Among the combinations generated for the production of β -carotene, and according to my experience in the area, it does not make sense to me the combination of a weak ATF to express the BTS1 and CrtI genes, corresponding to strain G1pDI 1 that produces around 0.81mg/g. This combination in particular is surprising, as BTS1 enzyme is not the most performant one for producing GGPP and, according to the literature, this is a bottleneck in the carotenoid pathway. Thus, to achieve a limited expression of this gene does not seem to be the best option to overproduce β -carotene.
- COMPASS system is based in the combinatorial construction of metabolic pathways. Within the different steps of the technique, the last one requires to integrate the modules into the genome. To validate this technique the authors evaluated three approaches. The first one integrated each module - one ATF associated to one CDS - in one of three different loci; the second one maintained the modules in the destination vectors, without integrating them in the genome; and the third one integrated the destination vector (with the modules) in a unique locus. The authors reported that the best results were for the second approach. This apparently contradicts the first objective of the technique, which is to generate stable strains through integration in the genome of the transcriptional modules.
- It is highlighted that 9 ATFs with well characterized, differential strengths are used (from a total of 106 ATFs in a library described in a previous report). Therefore, there is still plenty of room for fine-tuning the expression of a multiple gene pathway using this technique.

Reviewer #3:

Remarks to the Author:

In this work, the authors describe the systematic design of their "COMPASS" methodology, a custom cloning workflow that's intended to facilitate assembly of synthetic transcription units into biochemical pathways for expression in *S. cerevisiae* yeast – particularly when the optimal design

for expression of a target pathway is unknown. To this end, their workflow indeed seems to be relatively easy to use and well-suited for generating low-to-moderate genetic diversity which can then be screened in a forward fashion, as they've demonstrated with their β -carotene, β -ionone, and Naringenin proof-of-principles. Their proof-of-principles also do well to demonstrate that the synthetic variants which result from their combinatorial assemblies exhibit variable and sometimes improved levels of end-product expression, and, that the regulatory components + delivery mode (multi-locus integration vs. episomal vs. single locus integration en-masse) being utilized can influence end-product expression in seemingly unpredictable ways. This is not entirely unexpected for ectopically expressed synthetic assemblies, but certainly helps to justify efforts such as theirs to streamline combinatorial assembly and in turn facilitate empirical optimization of biochemical pathway expression. As they alluded towards the end of their paper, this sort of workflow could in theory be scaled up, but this was not the focus of their manuscript and in fact they did very little to validate the actual diversity in the higher-complexity assembly that they built (the "Narion library"). Although their claims were generally well-calibrated on this matter, and a rigorous demonstration of COMPASS' scalability is arguably beyond the scope of this work, I've included a specific comment about this later in my review (under "Other Comments" #3).

The authors also describe, in detail, the design/construction of a series of plasmid-based vectors that they've tailored for streamlining each of the in vitro and in vivo (*E. coli* + *S. cerevisiae*) manipulations in their COMPASS workflow. Overall, their workflow cleverly integrates pre-existing technologies for DNA assembly, genome editing, and cloning, and I was pleased to see that the authors are quite transparent about this. Their vectors were further designed to be somewhat modular, and this could clearly reduce the labor required to test assemblies for each of their three delivery modes in parallel. However, perhaps the most novel feature of their workflow is that they've implemented a subset of their previously-tested [PMID: 28531348] "artificial transcription factor/binding site" ("ATF/BS") elements to drive expression of transcription units. The relative strength of their nine ATF/BS elements was assessed using a yEGFP reporter, and the data for this is clearly presented in Supplementary Figure 1. Based on these measurements, they draw conclusions throughout the paper about the unpredictability of their assemblies (e.g., optimal expression of a biochemical pathway's end-product sometimes required a mixture of different ATF/BS elements that scored only weakly or moderately with their yEGFP reporter). This is a compelling idea, but some experimental validation would be needed to demonstrate that their ATF/BS elements (and TDH3 "positive control") are behaving as expected in the context of actual assemblies.

Given, also, that they designed their assemblies for "controllable" expression, I was surprised to find that they had only 1 set of experiments which compared expression outputs in the absence/presence of inducers (and based on their methods section it wasn't clear to me whether this experiment was conducted in an interpretable manner). I've provided some specific comments below to elaborate on these points, and other issues I found with the manuscript (mostly minor). There also appear to be some general issues with grammar and syntax throughout the work, although I have not attempted to itemize them.

Major Comments:

1. The authors' justification for implementing ATF/BS units (and hopefully the general appeal of the paper) would be greatly strengthened by an experimental demonstration that their yEGFP reporter behaves as expected when plugged into different ATF/BS within their assemblies. It would also help to clarify whether their claims throughout the paper are justified in assuming that the ATF/BS output strengths (as determined with their yEGFP reporter in a separate set of experiments) remain constant. Generally, their assertions seemed to imply that pathway expression overall is unpredictable because the optimal titration of individual components is inherently unpredictable. While this is certainly plausible, an alternative explanation for some of

their results follows that transcriptional outputs from their ATF/BS units are unpredictable to begin with because of cryptic context-dependent effects that result from assembly (perhaps related to local chromatin structure), and this in turn is what drives the differences in pathway output. Validation experiments with yEGFP would help to clarify this, and are expected to be straightforward (given their streamlined methodology). For example, the authors could create a mock β -carotene pathway assembly into their strain of choice, where one of the β -carotene CDSs is replaced with yEGFP under control of each of their nine ATF/BS units. Do the 9x strain assemblies, once verified, generally recapitulate the yEGFP output hierarchy of Supplementary Figure 1? Ideally they would measure this for each of their three delivery modes, but 1 alone would go a long way.

2. Another concern I had was with the authors' decision to use the constitutive TDH3 promoter as a "strong" positive control in their proof-of-principle assemblies. In theory this is a satisfactory idea, but the authors showed in their previous work [PMID: 28531348] that the TDH3 promoter does not always activate yEGFP as strongly as some of their ATF/BS units in the presence of inducers. Where does the TDH3 promoter strength fall on their Supplementary Figure 1 hierarchy when hooked up to yEGFP? Ideally they would include TDH3yEGFP as a 10th strain to assemble during the validation experiment I proposed above (under Major Comment #1). They should also make it more clear throughout the manuscript as to whether the plating assays they presented are considered "inducing" conditions. Finally, to better demonstrate the "controllability" of their system... could the authors use their β -carotene and β -ionone assemblies to compare plating in the absence and presence of inducers? I've mentioned some related concerns under Other Comment #5.

3. Page 4 – Figure 1 – This is a central figure that summarizes the COMPASS workflow. Could the authors be more clear in either the figure or its legend about relevant yeast manipulations carried out at "Level 1"? As I understand from the legend, the primary manipulations used at "Level 1" were in vitro assemblies with NEBuilder HiFi, cloned in E. coli. Is the yeast step for "Multi-locus integration" an optional procedure? This doesn't seem to be discussed until much later in the text, and at that point the authors never provided a reference back to Figure 1.

4. The authors, understandably, use a lot of acronyms and shorthand throughout their manuscript. Overall I found their terminology to be sufficiently defined at one point or another, but on a few occasions the definition was not immediately provided. It would greatly help if their acronyms were always defined upon first encounter – regardless of chronology in the body of the main text. For example, the term "homology regions (HRs)" is defined for the first time in the main text on page 9, but I've already encountered this acronym in the legend of Supplementary Figure 2 (which is first referenced on page 5). The meaning can be inferred from the Supp. Fig. 2 legend because they talk about "homology", but ideally the definition is provided verbatim. Another example was "TAR" on page 9 under "Level 2", which I could not find a definition for, other than in the title of primary reference #27.

5. In some cases (e.g., pDI 5 from Figure 5), the same two regulators seem to have been used to control two different CDSs. Have the authors assessed whether cross-talk is negligible (for example, does this result in lower maximal activation at either of the two loci appreciably, given that there could be competition for binding sites? A priori, it is not clear whether additional ATF copies would compensate for the extra binding sites in each unit)? Perhaps the authors could perform one additional validation experiment where they substitute their yEGFP into each loci's CDS region independently to compare fluorescence outputs, or at least compare the fluorescence output from a strain that has yEGFP controlled by a uniquely occurring ATF/BS unit versus when the yEGFP is controlled by a ATF/BS unit that occurs more than once (where additional copies of the ATF/BS are driving expression of non-yEGFP CDSs).

6. Page 24/25 – References to "Supplementary Figure 14" are made, but it's no where to be found. Are they referring to the current Supplementary Figure 12?

Other Comments:

1. Page 6 – Fig 2a – Should the bottom “BS_for” primer read “BS_rev”?
2. Page 15 – Main text – There was a mention of “previous chapters” which seems out of place. Also, there’s a reference to “Supplementary Figure 12” that doesn’t seem to coincide with what the authors are talking about.
3. Pages 15, 18, 19 – Main text – When the authors discuss results from their characterization of ~20 Narion isolates, it was confusing that they provide their results in terms of a % breakdown for the “library members”, “Narion strains”, and “library strains” (as if to imply that they’ve performed bulk analyses on pooled populations). To reduce ambiguity, they could simply refer to them as library/Narion *isolates* when describing the breakdown of their results. Of relevance here, the authors should also explain whether the 20 colonies were selected at random (which would in turn help to justify claims about library sampling), or whether they gave some consideration to colony color during the screen (as they did for previous figures). Finally, it’s my understanding that the theoretical complexity of a library pool is hard-capped by the number of transformants obtained upon transformation. Regardless of the theoretical complexity limit that they calculated, the authors could also note the actual (or at least the estimated) library size as determined by the total number of transformants produced from their transformation. Some relevant principles about libraries are outlined in this paper, PMID: 15857784.
4. Page 17-18 – Figure 6 – Methods section seems to indicate (on page 38) that the “Non-inducing medium” measurements displayed in Figure 6c were taken directly from precultures, but that “Inducing medium” measurements were taken after subculture? For a fair comparison, the non-inducing measurements should be taken from non-inducing media SUBCULTURES, split from the same precultures that were used for the inducing subculture measurements. Otherwise, it seems rather misleading to plot them on the same graph. Could the authors clarify or address this in some way? And to be clear, was β -ionone production measured only under “inducing” conditions? Authors could note this in the Figure 6 legend or its labels. Related to this, for Figure 5 + Supplementary Figures 9-11: authors could also mention – outside of the methods section – whether they considered the media “inducing” media.
5. Page 40 – Supplementary Figure 3 – If the “BamHI” site is located within LYS2, wouldn’t that be used to facilitate integration at the LYS2 locus? The legend says “NotI”. Whichever is the case, the authors will want to ensure congruency with related descriptions in their main text on page 11, Online Methods section on page 34, and Supplementary Note 1 on page 50. Also... labeling under the plasmid maps: seems to be an extra comma+space before “p5”?
6. Page 40 – Supplementary Figure 4 – Labeling under the plasmid maps: “p1” is defined even though it isn’t present in this figure, “p5” is not defined even though it IS present in this figure, and, there’s an extra comma+space before “p6”.
7. Page 44 – Supplementary Figure 8 – This figure is first referenced in regards to β -carotene production on page 11, but the title and legend mentions only β -ionone; given that the relationship between β -ionone and β -carotene may not be immediately apparent to unspecialized readers, the authors might consider adding a mention of β -carotene to this figure title or legend.
8. Page 52 – Supplementary Note 3 – A reference to “Supplementary Fig. 5e,h; 2” is made, which seems like it should be “Figure 5e,h; 2”.
9. Pages 55-56 – Supplementary Note 8 – Various figure references seem to be mis-labeled.

10. Page 57 – Supplementary Protocols – Authors switch from talking about molar ratios to concentration ratios within the first protocol... Was this intended to imply that the fragments generated from multiplex PCRs have heterogeneous/unpredictable molarity? If not, comprehensibility could be improved by sticking to a single descriptor, molarity. On a related note, do either of the “Multiplex-PCR-amplified ATFs,” and “their corresponding BS fragments” amplicons that the authors refer to correspond to POOLS of DNA fragments? The first paragraph of their Online Methods section specifies that “Amplified DNA parts were gel-purified prior to further use”... if they’re pools, what was the procedure to ensure that all fragment sizes were captured? Are primers designed to ensure that all amplicons are a similar length so that a single band can be excised? If not, are different length fragments excised+purified and then combined manually? The authors could certainly add some statements to the protocol to clarify these points – particularly in regards to which components represent pools and which do not. Related explanations under “Level 1” on Page 7 could perhaps also be improved... For example, was “five freely selected CDS units” meant to imply that a pool of the five CDS units were added in equal amounts together with the pool of 9 ATF/BS units and the linearized backbone?

RESPONSE TO REVIEWERS

In the following we provide a point-by-point response to the reviewer's comment. Changes made in the manuscript are highlighted in BLUE.

Reviewers' comments:

Reviewer #1 (Remarks to the Author):

The novelty of this work is related with an inducible combinatorial system, called COMPASS, that allows to rapidly generate different constructs and express them in yeast, by-passing in this way the eventual growth inhibition by metabolic burden.

There is a tremendous amount of work to demonstrate the performance of the system, with production of β -carotene and β -ionone developed as proofs of concept.

I have several concerns and questions that are required to fully appreciate the benefits of the technique:

The method employed for constructing the vectors is, in my opinion, similar to Golden Gate, a well known and widely used combinatorial method; therefore the system is not really new. The authors do not mention Golden Gate (GG) and do not compare their technique with this method. Furthermore, GG does not need to use PCR, and this technique profusely uses PCR.

RESPONSE:

DNA assembly strategies are categorized into two groups, each with outstanding benefits:

(1) Overlap-based assembly methods such as Gibson¹ and SLICE² assemblies and transformation associated recombination (TAR) allowing assembly directly in yeast. These methods are sequence independent, *i.e.*, they do not rely on the use of restriction enzymes (RE) and their recognition sequences, and do not require REs and DNA ligation to assemble neighboring parts. They provide flexibility as the order and orientation of the DNA parts may be changed easily, although this requires new primers and PCR.

(2) RE-based methods, such as Golden Gate (GG), require pre-existing or designed sequences in the neighboring DNA parts. The most significant limitation of GG assembly is that it is more sequence-dependent than SLICE, NEBuilder HiFi, and TAR, as the selected type IIs RE recognition site (*e.g.* *BsaI*) must be absent from the internal portions of all DNA fragments to be assembled. Therefore, in such methods DNA parts must often be 'domesticated' (to remove RE recognition sites) prior to their inclusion in multi-fragment cloning procedures.

We did not include such enzyme sites in COMPASS. The 11 core COMPASS vectors contain one or more recognition sites for *BsaI* or *BsaBI* (another type IIs RE). Similarly, Destination vector I containing the *McrII*, *McrIYB*, and *BTS1* coding sequences downstream of the JUB1-derived ATF has multiple *BsaI* and *BsaBI* recognition sites. However, this is not a limitation of our COMPASS method as it entirely relies on homology-dependent gene assembly, *e.g.* performed in yeast.

Of note, COMPASS utilizes overlap-based assembly (*e.g.* SLICE, Gibson, NEBuilder HiFi, and TAR). To perform such a highly efficient overlap-based cloning, we utilized high-fidelity DNA polymerases such as Phusion, PrimeSTAR GXL or Q5 DNA polymerases with accuracies of 255118, 1870763 and 118467, respectively (accuracy is defined as the number of bases over which one substitution error is expected to occur), facilitating PCR amplification of various DNA fragments (*i.e.* products ≥ 30 kb in length, GC-rich templates), while maintaining exceptionally high fidelity³. We now provide information about the polymerases used in Online Methods (chapter: 'General').

In COMPASS, the DNA parts are initially amplified by PCR and each CDS-terminator sequence is flanked by a unique RE cleavage site. Once the library of ATF/BS-CDS modules (at Level 1 or Level 2) for the pathway of interest is built, the CDS-terminator sequence can be replaced with any other CDS-terminator of interest. Therefore, COMPASS embraces a flexible approach. We now used this approach to construct new plasmids required to perform the experiment requested in question 1 of reviewer 2 (**Supplementary Note 1**, and **Supplementary Figure 2**).

For cloning methods utilizing RE-based methods, such as MoClo⁴, enzymes are used for the restriction of inserts to perform the next level of cloning. One technical problem, especially in the context of combinatorial libraries, is the low concentration of cut-out inserts which might impair cloning efficacy.

Moreover, large plasmids cannot be easily amplified in *E. coli*. Additionally, some assembly products are not stable or clonable in *E. coli* due to, for example, high AT content or Z-DNA-like structures, or the outer membrane of *E. coli* is not easily permeable for macromolecules⁵, COMPASS vectors are well suited to perform *in vivo* cloning in yeast using its native homologous recombination capacity. The vectors employed in COMPASS are equipped with yeast selection markers allowing positive selection for TAR cloning. While NEBuilder HiFi is

preferred for generating the ATF/BS-CDS modules, TAR is the desired approach for combining multi-ATF/BS-CDS modules of pathways.

As COMPASS vectors carry open reading frames of different *E. coli* and yeast selection markers, it allows assembling different inserts in different backbones in a single cloning reaction tube at Level 1. Therefore, COMPASS is a fast method compared to Golden Gate and VEGAS, a combinatorial variant of GG, for pathway optimization in yeast⁶.

In the Introduction of our manuscript, we now provide a more detailed explanation of the benefits of COMPASS (page 2, lines 40 – 43, 49 - 52, pages 2 - 3, lines 54 – 65, 72, 73, 75 – 78, page 4, 105 – 106, page 11, lines 246 – 248, page 23, lines 530 – 534, pages 23 - 24, lines 563 – 569). In addition, we added **Supplementary Table 1** that compares the most common combinatorial cloning approaches.

The system requires a construction step in *E. coli*. In this sense, it looks to be more related with MoClo.

RESPONSE:

MoClo⁴ is based on *in vitro* cloning and transformation of the assembled products into *E. coli*.

In the current realization of COMPASS, Level 1 assemblies are obtained using *in vitro* cloning followed by *E. coli* transformation. Using the same approach, Level 1 constructs (in addition to Level 2 constructs) can be assembled in yeast using inexpensive TAR. This is due to the fact that the different vectors carry different yeast selection markers, in addition to *E. coli* selection markers.

Their inducible system is based on the expression of artificial transcription factors, (ATF) that control the activity of the heterologous genes to be expressed. The ATF constructs are very long, therefore, the expression of these ATFs might also generate an extra metabolic demand for the cell. There is no reference in the text to this subject. In order to fully appreciate the performance of COMPASS, compared to other published well known methods, I propose that the authors report the growth titers obtained with their system - not only yields; and compare them with those achieved in an inducible system for comparison, not only with a constitutive system, where the metabolic burden might be higher.

RESPONSE:

Thanks to the reviewer for addressing this point. To check whether a growth penalty is associated with an overexpression of artificial transcription factors (ATFs) in the engineered strains, we used a modified version of a previously reported high-throughput fitness assay in which flow cytometry is used to monitor growth competition of fluorescently labeled strains⁷. In this assay, AcGFP1-expressing wild-type (WT) cells (parental strain) and DsRED-expressing modified cells (containing the ATF) were co-cultured in (i) non-inducing, and (ii) in inducing medium.

The co-cultures were maintained by serial dilution. At each time point, samples were removed for analysis by flow cytometry to determine the ratio of DsRED-positive to AcGFP1-positive cells. Rare events that appear to be both DsRED-positive and AcGFP1-positive represent instances in which a modified cell and a wild-type cell are misidentified by the cytometer as a single cell, and we took this into account in our analyses. Because a large number of individual cells (20,000 each) can be measured by flow cytometry, this assay allows calculating relative growth rates by determining the ratio of ATF containing cells to WT cells over the course of the competition.

For growth competition experiments, DsRED-labeled cells (*i.e.*, PCB, CB1, MXintpDI 1, and RC1) and corresponding WTs cells labeled with AcGFP1 were mixed in single vessels. We inoculated co-cultures at a ratio of 8 : 2 (modified strain : WT) at OD₆₀₀ ~0.1 in 4 ml YPDA medium with and without inducer. Cells were grown for 48 h in a rotary shaker at 30°C and 230 rpm. At time points 6, 24, and 48 h, samples were removed for OD₆₀₀ measurements and analysis (by flow cytometry) to determine the ratio of DsRED-positive to AcGFP1-positive cells. At each time point, cells were diluted 100-fold into fresh liquid medium for growth until the next time point was reached. We counted DsRED- and AcGFP1-labeled cells, see **Online Methods**, page 45 - 46, lines 1360 - 1388 of the revised manuscript. We now included these data in our manuscript, see **Results**, page 15, lines 358 – 373, page 16, lines 392 - 394, page 19, lines 448 – 450, and **Discussion**, page 24, lines 551 - 569), and **Supplementary Note 3** and **Supplementary Figure 12**.

Our results demonstrate that growth of strains CB1, MXintpDI 1, and RC1 is reduced by 16%, 7% and 13%, respectively, in inducing medium compared to non-inducing medium. Interestingly, there was no difference in the growth rate of these three strains and their respective control strains (Gen 0.1 for CB1, and IMX672.1 for MXintpDI 1 and RC1) in non-inducing medium, while strain PCB, which constitutively expresses the β -carotene pathway, showed 3.6% reduction in growth rate compared to the control strain (Gen 0.1) in non-inducing medium.

Additionally, we now investigated the productivity of the IPTG-inducible system utilized in the current version of COMPASS in response to question 3 of reviewer 2. As shown in **Supplementary Fig. 13** (and **Supplementary Note 4**), induction of the regulators led to 8.5-fold more β -carotene than observed in control medium (*i.e.*, non-inducing medium) (**Supplementary Fig. 13**) in strain CB1. This is 1.5-fold more than in strain PCB (**Fig. 6i** and **Supplementary Fig. 13c**). Overall, the growth data (**Supplementary Figure 12**), together with the data referring to the production amount achieved with plant-derived ATF/BSs (**Fig. 6i** and **Supplementary Fig. 13c**) suggests

that in COMPASS carbon sources can be redirected towards the production of high levels of targeted metabolites at a desired time point by decoupling growth and metabolite production phases.

The technique requires several selection markers (antibiotic resistance genes), that are left in the vector and therefore integrate in the yeast genome.

RESPONSE:

The presence of the positive yeast and *E. coli* selection markers in the modules provides a higher level of control for successful integration (a critical issue in combinatorial approaches). Importantly, the *E. coli* selection markers are not integrated into the yeast genome. The donor fragments contain a yeast selection marker, an inducible plant regulator (ATF/BS), a CDS, and a yeast terminator from Level 1 vectors, in addition to 50-bp up- and down-stream homology regions (HRs) supporting the integration into pre-selected genomic loci. The resulting library of yeast strains grows on plates containing appropriate yeast selection markers (auxotrophic and dominant). However, the number of suitable selection markers is currently limited. As now discussed in the manuscript, COMPASS can be further improved by deleting (or mutating) the marker genes embodied in the COMPASS vectors, after their initial integration into the genome. In this way, cells become competent for the next round of pathway engineering, as markers of the COMPASS vectors can be “re-used”.

One way of removing or mutating marker genes relies on CRISPR/Cas9-mediated genome modification. To this end we had already constructed plasmids pCOM001 and pCOM002, based on plasmid pCRCT which harbors iCas9 and tracrRNA⁸, in which multiple donors and corresponding guide sequence cassettes can be inserted to introduce frameshift mutations in the targeted genomic sites. Plasmid pCOM002 leads to the disruption of the five yeast auxotrophic markers employed in our integration cassettes (*URA3*; *LEU2*, *HIS3*, *TRP1*, *LYS2*; see **Supplementary Fig. 9**). Therefore, after the integration of the desired modules into the host genome, plasmid pCOM002 can be transformed into the yeast cells to disrupt the auxotrophic markers.

Moreover, to also allow the disruption of yeast dominant markers, we now developed plasmid pCOM009 (see **Online Methods**, page 31, lines 834 – 842, and **Results**, pages 13 – 14, lines 307 - 321) which encodes multiple donors and corresponding guide sequences for the disruption of the five yeast dominant genes (*ble*, *nat1*, *blp^R*, *neo*, *hph*; see **Supplementary Figure 10**).

Another promising approach for further improvement of COMPASS in the future would be to flank the ten marker genes of the COMPASS vectors with *loxP* sites. In this way, the markers can be removed from the yeast genome after integration of the first ten pathway genes, using Cre recombinase⁹.

Among the combinations generated for the production of β -carotene, and according to my experience in the area, it does not make sense to me the combination of a weak ATF to express the *BTS1* and *CrtI* genes, corresponding to strain G1pDI 1 that produces around 0,81mg/g. This combination in particular is surprising, as *BTS1* enzyme is not the most performant one for producing GGPP and, according to the literature, this is a bottleneck in the carotenoid pathway. Thus, to achieve a limited expression of this gene does not seem to be the best option to overproduce β -carotene.

RESPONSE:

We observed the highest amount of β -carotene accumulation in G1pDI 1. Its parental strain (Gen 0.1) has an optimized biochemical background in which endogenous metabolites were redirected towards the production of farnesyl diphosphate (FPP), a central precursor to nearly all isoprenoid products^{10,11}. We hypothesized that a high production level of GGPP (*BTS1*-derived product), in addition to a high production level of FPP, may be metabolically burdensome for the cell. Therefore, a weak plant regulator (ATF/BS) implemented in the *BTS1* module, in combination with strong plant regulators in the two other modules, leads to a balanced production of β -carotene in the optimized FPP background.

To our knowledge yeast promoters have commonly been used for metabolic engineering purposes. We previously tested 40 different constitutive promoters from *S. cerevisiae*¹². The data demonstrate that transcriptional output of 92% of them was less than 0.5-fold compared to the *TDH3* promoter.

In response to this reviewer and question 3 of reviewer 2, we now performed the following experiment: Plasmid pGN005B-ProTDH3¹³ was integrated into the *ura3-52* locus of the yeast genome. yEGFP expression was then measured in both, inducing and non-inducing medium. The data are now included in the **Results** (page 6, lines 135 – 136 of the revised manuscript), **Supplementary Note 3**, and **Supplementary Figure 1**. We observed that the transcriptional output of the plant regulators employed here is 0.4- to 5-fold of that of the strong *TDH3* promoter. These data suggest that our plant-derived regulators (together with the synthetic promoters they control) are more strongly expressed than other commonly used yeast promoters.

Previously, Ding *et al.* reported that GGPPSbc leads to an improved GGPP supply over *BTS1*¹⁴. However, we observed that *BTS1* expressing yeasts produce more β -ionone than GGPPSbc expressing cells in 20 characterized Narion isolates (**Figure 6d**). Additionally, both the lowest β -ionone producer (Narion 19) as well as one of the top β -ionone producers (Narion 14) contain a weak ATF/BS to drive the expression of *BTS1* (**Figure 6d**).

Overall, our data from the β -carotene and Narion collections suggest that unpredictable combinations of ATF/BSs in different background strains balance the expression of enzymes in a metabolite pathway.

COMPASS system is based in the combinatorial construction of metabolic pathways within the different steps of the technique; the last one requires to integrate the modules into the genome. To validate this technique the authors evaluated three approaches. The first one integrated each module - one ATF associated to one CDS - in one of three different loci; the second one maintained the modules in the destination vectors, without integrating them in the genome; and the third one integrated the destination vector (with the modules) in a unique locus. The authors reported that the best results were for the second approach. This apparently contradicts the first objective of the technique, which is to generate stable strains through integration in the genome of the transcriptional modules.

RESPONSE:

A problem often encountered with episomal expression systems is an unstable product output. As we showed in **Figure 5i**, the standard deviation for β -carotene production was significantly higher for plasmid-based systems than for genome-integrated systems. Moreover, we now observed a low amount of β -carotene production for another colony of G1pDI 1 (see **Supplementary Note 4** and **Supplementary Figure 13**). A likely reason for the quite huge variation of β -carotene production levels in different replicates might be different copy numbers of 2μ -based plasmids in different yeast cells. Plasmids provide limited control of copy number, and segregational instability can be a critical issue even in selective medium.

We strongly favor the chromosome integration-based version of COMPASS as homologous recombination is highly efficient in *S. cerevisiae*. Moreover, it allows generating libraries of stable yeast variants through only four cloning reactions, followed by the decoupled integration of the constructs into the yeast genome. Additionally, in the integration-based system, selection pressure is not required.

It is highlighted that 9 ATFs with well characterized, differential strengths are used (from a total of 106 ATFs in a library described in a previous report). Therefore, there is still plenty of room for fine-tuning the expression of a multiple gene pathway using this technique.

RESPONSE: Yes, indeed. This is a further important aspect of COMPASS and we therefore had mentioned it in our manuscript (page 23, lines 537 - 540 of the revised manuscript).

Reviewer #2 (Remarks to the Author):

In this work, the authors describe the systematic design of their “COMPASS” methodology, a custom cloning workflow that’s intended to facilitate assembly of synthetic transcription units into biochemical pathways for expression in *S. cerevisiae* yeast – particularly when the optimal design for expression of a target pathway is unknown. To this end, their workflow indeed seems to be relatively easy to use and well-suited for generating low-to-moderate genetic diversity which can then be screened in a forward fashion, as they’ve demonstrated with their β -carotene, β -ionone, and Naringenin proof-of-principles. Their proof-of-principles also do well to demonstrate that the synthetic variants which result from their combinatorial assemblies exhibit variable and sometimes improved levels of end-product expression, and, that the regulatory components + delivery mode (multi-locus integration vs. episomal vs. single locus integration en-masse) being utilized can influence end-product expression in seemingly unpredictable ways. This is not entirely unexpected for ectopically expressed synthetic assemblies, but certainly helps to justify efforts such as theirs to streamline combinatorial assembly and in turn facilitate empirical optimization of biochemical pathway expression. As they alluded towards the end of their paper, this sort of workflow could in theory be scaled up, but this was not the focus of their manuscript and in fact they did very little to validate the actual diversity in the higher-complexity assembly that they built (the “Narion library”). Although their claims were generally well-calibrated on this matter, and a rigorous demonstration of COMPASS’ scalability is arguably beyond the scope of this work, I’ve included a specific comment about this later in my review (under “Other Comments” #3).

The authors also describe, in detail, the design/construction of a series of plasmid-based vectors that they’ve tailored for streamlining each of the in vitro and in vivo (*E. coli* + *S. cerevisiae*) manipulations in their COMPASS workflow. Overall, their workflow cleverly integrates pre-existing technologies for DNA assembly, genome editing, and cloning, and I was pleased to see that the authors are quite transparent about this. Their vectors were further designed to be somewhat modular, and this could clearly reduce the labor required to test assemblies for each of their three delivery modes in parallel. However, perhaps the most novel feature of their workflow is that they’ve implemented a subset of their previously-tested (PMID: 28531348) “artificial transcription factor/binding site” (“ATF/BS”) elements to drive expression of transcription units. The relative strength of their nine ATF/BS elements was assessed using a yEGFP reporter, and the data for this is clearly presented in Supplementary Figure 1. Based on these measurements, they draw conclusions throughout the paper about the unpredictability of their assemblies (e.g., optimal expression of a biochemical pathway’s end-product sometimes required a mixture of different ATF/BS elements that scored only weakly or moderately with their yEGFP reporter). This is a compelling idea, but some experimental validation would be needed to demonstrate that their ATF/BS elements (and TDH3 “positive control”) are behaving as expected in the context of actual assemblies.

Given, also, that they designed their assemblies for “controllable” expression, I was surprised to find that they had only 1 set of experiments which compared expression outputs in the absence/presence of inducers (and based on their methods section it wasn’t clear to me whether this experiment was conducted in an interpretable manner). I’ve provided some specific comments below to elaborate on these points, and other issues I found with the manuscript (mostly minor). There also appear to be some general issues with grammar and syntax throughout the work, although I have not attempted to itemize them.

RESPONSE: See our responses in the following.

Major Comments:

The authors’ justification for implementing ATF/BS units (and hopefully the general appeal of the paper) would be greatly strengthened by an experimental demonstration that their yEGFP reporter behaves as expected when plugged into different ATF/BS within their assemblies. It would also help to clarify whether their claims throughout the paper are justified in assuming that the ATF/BS output strengths (as determined with their yEGFP reporter in a separate set of experiments) remain constant. Generally, their assertions seemed to imply that pathway expression overall is unpredictable because the optimal titration of individual components is inherently unpredictable. While this is certainly plausible, an alternative explanation for some of their results follows that transcriptional outputs from their ATF/BS units are unpredictable to begin with because of cryptic context-dependent effects that result from assembly (perhaps related to local chromatin structure), and this in turn is what drives the differences in pathway output. Validation experiments with yEGFP would help to clarify this, and are expected to be straightforward (given their streamlined methodology). For example, the authors could create a mock β -carotene pathway assembly into their strain of choice, where one of the β -carotene CDSs is replaced with yEGFP under control of each of their nine ATF/BS units. Do the 9x strain assemblies, once verified, generally recapitulate the yEGFP output hierarchy of Supplementary Figure 1? Ideally they would measure this for each of their three delivery modes, but 1 alone would go a long way.

RESPONSE:

We performed the following experiment: We inserted yEGFP downstream of the nine plant regulators (ATF/BS units) in Acceptor vector A. To this end, nine Acceptor vectors A-ATF/BS-BTS1 (**Supplementary Figure 11** and **Supplementary Table 5**) were digested with *I-CeuI* to remove the *BTS1-Ter_{TDH3}* fragment. Next, the *yEGFP-Ter_{CYC1}* fragment was inserted in each of the nine linearized backbones to generate nine Acceptor vectors A-ATF/BS-yEGFP. Each of the nine ATF/BS-yEGFP modules, together with NLS-ATAF1-GAL4AD/2X-McrtI and NLS-DBD_{JUB1}-GAL4AD/2X-McrtYB modules (**Figure 5h**), were integrated into the *XII-5*, *X-3* and *XI-3* loci, respectively, to generate the nine yeast strains YEGFP 1- YEGFP 9; see **Online Methods** (page 44 - 45, lines 1319 - 1337), and **Supplementary Table 5**. The yEGFP outputs were tested for all strains in the absence and presence of inducer. Our results, shown in **Results** (page 6, lines 137 – 145), **Discussion** (pages 23 - 24, lines 563 – 569), **Supplementary Note 1** and **Supplementary Figure 2** demonstrate that in induction medium the strong plant regulators (strains YEGFP1-YEGFP3, **Supplementary Figure 2b**) lead to a higher yEGFP expression than the weak regulators (strains YEGFP7-YEGFP9, **Supplementary Figure 2b**). Moreover, we previously showed that higher fluorescence outputs are achieved for these nine regulators when integrated into locus *ura3-52*¹³ in comparison to the same regulators integrated into the locus *XII-5* (in the β -carotene producing background) (Student’s *t*-test: *p*-value < 0.01; **Supplementary Fig. 2b**). Therefore, we assume that the genomic integration position generally affects the transcriptional output of the plant regulators. However, we also note that the strengths of the different regulators relative to each other remain largely unaffected by the chromosomal locus, at least for the two loci we tested here, *i.e.* *ura3-52* and *XII-5*. This result is in agreement with an earlier study that demonstrated the genomic integration position of promoters significantly affects the expression level of downstream genes and, hence, the proteins they encode¹⁵.

Another concern I had was with the authors’ decision to use the constitutive TDH3 promoter as a “strong” positive control in their proof-of-principle assemblies. In theory this is a satisfactory idea, but the authors showed in their previous work (PMID: 28531348) that the TDH3 promoter does not always activate yEGFP as strongly as some of their ATF/BS units in the presence of inducers. Where does the TDH3 promoter strength fall on their Supplementary Figure 1 hierarchy when hooked up to yEGFP? Ideally they would include TDH3yEGFP as a 10th strain to assemble during the validation experiment I proposed above (under Major Comment #1). They should also make it more clear throughout the manuscript as to whether the plating assays they presented are considered “inducing” conditions. Finally, to better demonstrate the “controllability” of their system... could the authors use their β -carotene and β -ionone assemblies to compare plating in the absence and presence of inducers? I’ve mentioned some related concerns under Other Comment #5.

RESPONSE:

In response to this question and question 5 of reviewer 1, we now performed the following experiment: *PmeI*-digested pGN005B-Pro_{TDH3} (*TDH3* promoter upstream of yEGFP)¹³ was integrated into locus *ura3-52* of yeast strain YPH500, and yEGFP output was measured in both, inducing and non-inducing medium using flow cytometry¹³. The data are now included in **Results** (page 6, lines 135 - 136) and **Supplementary Figure 1**. The transcriptional outputs of the plant regulators were 0.4- to 5-fold compared to that of the strong *TDH3* promoter.

They should also make it more clear throughout the manuscript as to whether the plating assays they presented are considered “inducing” conditions. Finally, to better demonstrate the “controllability” of their system... could the authors use their β -carotene and β -ionone assemblies to compare plating in the absence and presence of inducers? I’ve mentioned some related concerns under Other Comment #5.

RESPONSE:

We always plate yeast cells (i) transformed with episomal plasmids (approach 2), or (ii) containing expression cassettes integrated into the genome (approaches 1 and 3) on selective medium containing 2% (w/v) glucose (non-inducing medium). Thereafter, we perform the induction experiments: cells are plated on inducing SC medium containing 20 mM isopropyl- β -D-thiogalactopyranoside (IPTG), 2% (w/v) galactose, 1% (w/v) raffinose, and the appropriate selection markers (inducing and selective medium) for plasmid-based expression, and inducing YPDA medium plates (20 mM IPTG, 2% (w/v) galactose) in the case of the genomic expression cassettes (induction experiment, **Online Methods**).

In response to this question, we now performed the following experiment: We selected the best β -carotene producers of each of the three approaches (**Figure 5**): PCB, CB1, CC1 (approach 1), G1pDI 1 and MXpDI 1 (approach 2), G1intpDI 1 and MXintpDI 1 (approach 3). The cells were grown in SC-Leu/-Ura/His medium and subsequently inoculated in inducing and non-inducing medium; see **Online Methods**, pages 46, lines 1391 - 1397. The results are presented in **Results** (page 15, lines 366 - 373), **Supplementary Note 4** and **Supplementary Figure 13**. In induction medium, the strains produced a high level of the product, while in the absence of inducer only a low to slightly elevated level (above background) was observed (**Supplementary Fig. 13a** and **13b**).

We furthermore quantified β -carotene production in strains PCB, CB1, G1pDI 1, G1intpDI 1 using the method reported by Lian *et al.* (2017)¹⁶ (see **Online Methods**, pages 47 - 48, lines 1438 - 1448). To this end, β -carotene producing strains were pre-cultured in SC medium with appropriate selection markers for 2 days, inoculated into 5 mL of both, fresh induction and non-induction medium with an initial OD₆₀₀ of 0.1 in 15-mL culture tubes, and cultured (30°C, 250 rpm) for 5 days. The stationary phase yeast cells were collected by centrifugation at 13,000 \times g for 1 min and cell precipitates were resuspended in 1 mL of 3N HCl, boiled for 5 min, and then cooled in an ice-bath for 5 min. The lysed cells were washed with ddH₂O and resuspended in 400 μ L acetone to extract β -carotene. The cell debris was removed by centrifugation and the β -carotene containing supernatant was analyzed for its absorbance at 454 nm (OD₄₅₄). The amount of β -carotene level was normalized to the cell density. The results are now presented in **Results** (page 15, lines 366 - 373), **Supplementary Note 4** and **Supplementary Figure 13c**. Our results demonstrate that integrating genes into multiple loci of the genome (strain CB1) results in a high level of β -carotene (8.5-fold higher in induction medium compared to non-induction medium). Moreover, strain CB1 produced 1.7-fold more β -carotene than strain PCB (containing the β -carotene pathway genes under the control of the *TDH3* promoter). We also observed a high level of β -carotene (2.3-fold more than in strain PCB), when a Destination vector harboring the β -carotene pathway genes under the control of the plant regulators was integrated into a single locus of the Gen 0.1 background (strain G1intpDI 1).

Page 4 – Figure 1 – This is a central figure that summarizes the COMPASS workflow. Could the authors be more clear in either the figure or its legend about relevant yeast manipulations carried out at “Level 1”? As I understand from the legend, the primary manipulations used at “Level 1” were in vitro assemblies with NEBuilder HiFi, cloned in *E. coli*. Is the yeast step for “Multi-locus integration” an optional procedure? This doesn’t seem to be discussed until much later in the text, and at that point the authors never provided a reference back to Figure 1.

RESPONSE:

In the legend, we briefly mention this point.

Page 5, lines 122 - 123: “Thereafter, the ten groups of ATF/BS-CDS modules of Level 1 are integrated into ten defined loci of the genome.” We slightly rephrased the sentence to emphasize this more.

Page 9, line 198: Description for construction of ATF/BS-CDS modules at Level 1: we now included a reference to **Figure 1**.

Page 11, lines 265, page 12, 269: Description for assembly of the pathway library at Level 2: we now included a reference to **Figure 1**.

The authors, understandably, use a lot of acronyms and shorthand throughout their manuscript. Overall I found their terminology to be sufficiently defined at one point or another, but on a few occasions the definition was not immediately provided. It would greatly help if their acronyms were always defined upon first encounter – regardless of chronology in the body of the main text. For example, the term “homology regions (HRs)” is defined for the first time in the main text on page 9, but I’ve already encountered this acronym in the legend of Supplementary Figure 2 (which is first referenced on page 5). The meaning can be inferred from the Supp. Fig. 2 legend because they talk about “homology”, but ideally the definition is provided verbatim. Another example was “TAR” on page 9 under “Level 2”, which I could not find a definition for, other than in the title of primary reference #27.

RESPONSE:

Definition for HR was added to the text that refers to **Supplementary Figure 3** of the revised manuscript; page 7, lines 165 - 166.

Definition for TAR was added (page 11, line 250).

In some cases (e.g., pDI 5 from Figure 5), the same two regulators seem to have been used to control two different CDSs. Have the authors assessed whether cross-talk is negligible (for example, does this result in lower maximal activation at either of the two loci appreciably, given that there could be competition for binding sites? A priori, it is not clear whether additional ATF copies would compensate for the extra binding sites in each unit)? Perhaps the authors could perform one additional validation experiment where they substitute their yEGFP into each loci's CDS region independently to compare fluorescence outputs, or at least compare the fluorescence output from a strain that has yEGFP controlled by a uniquely occurring ATF/BS unit versus when the yEGFP is controlled by a ATF/BS unit that occurs more than once (where additional copies of the ATF/BS are driving expression of non-yEGFP CDSs).

RESPONSE:

It was previously reported that the transcriptional output of a TF is largely affected by the specificity of the TF for binding to its promoter rather than by the concentration of the TF¹⁷. We recently showed that increasing the binding site copy number often leads to a higher transcriptional output of plant regulators¹³. This suggests that the amount of the plant-derived ATF expressed from an IPTG inducible promoter does generally not limit its transcriptional output.

In response to this question, we now performed the following experiment; (i) We constructed Acceptor vector A containing NLS-JUB1-EDLLAD-EDLLAD-2X-AcGFP1 (**Online Methods**, page 45, lines 1340 - 1358, pAVA-GREEN). The NLS-JUB1-EDLLAD-EDLLAD-2X-AcGFP1 module was integrated into locus *XII-5* to generate yeast strain AGREEN (**Supplementary Figure 14a**). (ii) We constructed Acceptor vector B containing NLS-JUB1-EDLLAD-EDLLAD-2X-DsRED (**Online Methods**, page 45, lines 1340 - 1358, pAVB-RED). The NLS-JUB1-EDLLAD-EDLLAD-2X-DsRED module was integrated into locus *XI-3* to generate strain BRED (**Supplementary Figure 14a**). (iii) We integrated the NLS-JUB1-EDLLAD-EDLLAD-2X-AcGFP1 and NLS-JUB1-EDLLAD-EDLLAD-2X-DsRED modules into the *XII-5* and *XI-3* locus, respectively, to generate strain AB-GREEN-RED. Finally, we determined the AcGFP1 and DsRED outputs using flow cytometry. The data are shown in **Results** (page 16, lines 398 - 413), **Supplementary Note 8**, and **Supplementary Figure 14b**. Our results demonstrate that the expression of AcGFP1 (strain AGREEN) and DsRED (strain BRED) are 34% and 21%, respectively, higher than those in the strain co-expressing AcGFP1 and DsRED (each under the control of the same regulator, strain AB-GREEN-RED). This suggests that overproduction of regulators might create an increased metabolic burden resulting in growth inhibition⁸⁻¹³.

Page 24/25 – References to “Supplementary Figure 14” are made, but it’s no where to be found. Are they referring to the current Supplementary Figure 12?

RESPONSE:

Supplementary Figure 9 (previously **Supplementary Figure 12**) is correct; we corrected the text.

Other Comments:

1. Page 6 – Fig 2a – Should the bottom “BS_for” primer read “BS_rev”?

RESPONSE:

The reviewer is right. We corrected the figure.

2. Page 15 – Main text – There was a mention of “previous chapters” which seems out of place. Also, there’s a reference to “Supplementary Figure 12” that doesn’t seem to coincide with what the authors are talking about.

RESPONSE:

The text was corrected. **Supplementary Figure 9** (previously **Supplementary Figure 12**) explains the working procedure employed to generate strain MXFde0.2.

3. Pages 15, 18, 19 – Main text – When the authors discuss results from their characterization of ~20 Narion isolates, it was confusing that they provide their results in terms of a % breakdown for the “library members”, “Narion strains”, and “library strains” (as if to imply that they’ve performed bulk analyses on pooled populations). To reduce ambiguity, they could simply refer to them as library/Narion *isolates* when describing the breakdown of their results. Of relevance here, the authors should also explain whether the 20 colonies were selected at random (which would in turn help to justify claims about library sampling), or whether they gave some consideration to colony color during the screen (as they did for previous figures). Finally, it’s my understanding that the theoretical complexity of a library pool is hard-capped by the number of transformants obtained upon transformation. Regardless of the theoretical complexity limit that they calculated, the authors could also note the actual (or at least the estimated) library size as determined by the total number of transformants produced from their transformation. Some relevant principles about libraries are outlined in this paper, PMID: 15857784.

RESPONSE:

After integration of all nine modules into the desired loci, the transformed cells are grown in 1 ml medium with appropriate selection markers for 3 d that select for the presence of the Cas9 and sgRNA-encoding plasmids. Subsequently, 1 ml of the transformed cells are spread (after a 10^2 dilution) on five 100 x 15 mm plates. Typically, 80 colonies are grown on each plate, i.e., 40,000 (10^2 (dilution factor) x 5 (number of plates) x 80 (number of colonies on each plate)) colonies in total of which 20 were randomly taken and replicated on non-selective induction plates. Thereafter, colonies were scraped of the plates and used to establish liquid-media cultures thereby generating a library of isolates (called Narions) producing β -ionone and naringenin at different levels. Therefore, the actual complexity of the library is 0.05%.

We corrected the text:

1. Page 19, lines 468 - 470 of the revised manuscript: "Twenty colonies (representing 0.0000025% of the theoretical complexity of the library and ~0.05% of its actual complexity) were randomly selected to identify ATF/BS sequences driving the expression of the CDSs (using primers listed in **Supplementary Table 8**)."

2. Page 42 - 43, lines 1261 - 1264: "To select for β -carotene producing strains, cells were inoculated in 1 ml liquid culture medium for 3 d, followed by plating 1 ml of 10^2 -diluted cells on media that selected for the presence of the sgRNA and Cas9 plasmids pTAJAK-92⁴ and pCOM005 which harbor the G418 and LYS2 selection marker genes, respectively."

3. Page 43, lines 1291 - 1293: "Cells were inoculated in 1 ml liquid culture medium for 3 d, followed by plating 1 ml of 10^2 -diluted cells on media that selected for the presence of the sgRNA plasmids which encodes the *LYS2* selection marker."

4. Page 44, lines 1305 - 1306: "Cells were inoculated in 1 ml liquid culture medium for 3 d, followed by plating 1 ml of 10^2 -diluted cells on media that selected for the presence of the sgRNA plasmids."

5. Page 19, line 473, page 20, line 488, and page 44, line 1309: "Narion strains" changed to "Narion isolates".

4. Page 17-18 – Figure 6 – Methods section seems to indicate (on page 38) that the "Non-inducing medium" measurements displayed in Figure 6c were taken directly from precultures, but that "Inducing medium" measurements were taken after subculture? For a fair comparison, the non-inducing measurements should be taken from non-inducing media SUBCULTURES, split from the same precultures that were used for the inducing subculture measurements. Otherwise, it seems rather misleading to plot them on the same graph. Could the authors clarify or address this in some way? And to be clear, was β -ionone production measured only under "inducing" conditions? Authors could note this in the Figure 6 legend or its labels. Related to this, for Figure 5 + Supplementary Figures 9-11: authors could also mention – outside of the methods section – whether they considered the media "inducing" media.

RESPONSE:

We corrected the legend to **Figure 6c**.

"(c) Screening of NG production. Twenty colonies were pre-cultured in SC medium with appropriate selection markers and subsequently used to monitor yEGFP output in the absence (YPDA, 2% (w/v) glucose) and presence of inducer (YPDA, 2% (w/v) galactose and 20 μ M IPTG)."

5. Page 40 – Supplementary Figure 3 – If the "BamHI" site is located within LYS2, wouldn't that be used to facilitate integration at the LYS2 locus? The legend says "NotI". Whichever is the case, the authors will want to ensure congruency with related descriptions in their main text on page 11, Online Methods section on page 34, and Supplementary Note 1 on page 50. Also... labeling under the plasmid maps: seems to be an extra comma+space before "p5"?

RESPONSE:

1) Both, *NotI* and *BamHI* can be used to digest Destination vector I. We corrected the legend to **Supplementary Figure 3 (Supplementary Figure 4** of the revised manuscript): "Destination vector I is equipped with *NotI* and *BamHI* sites allowing integration of the plasmid into *URA3* and *LYS2* loci of the yeast genome, respectively."

2) Extra comma and space before "p5" were deleted.

6. Page 40 – Supplementary Figure 4 – Labeling under the plasmid maps: "p1" is defined even though it isn't present in this figure, "p5" is not defined even though it IS present in this figure, and, there's an extra comma+space before "p6".

RESPONSE:

1) "p1" label under the plasmid map was deleted and "p5" was added.

2) Extra comma and space before "p6" were deleted.

7. Page 44 – Supplementary Figure 8 – This figure is first referenced in regards to β -carotene production on page 11, but the title and legend mentions only β -ionone; given that the relationship between β -ionone and β -carotene may not be immediately apparent to unspecialized readers, the authors might consider adding a mention of β -carotene to this figure title or legend.

RESPONSE:

β -Carotene was added to the title and legend of **Supplementary Figure 8 (Supplementary Figure 11 of the revised manuscript): “Supplementary Fig. 11. Construction of library of modules for ATF/BS upstream of β -carotene and β -ionone pathway genes.**

(a) Schematic overview of modules of ATF/BS library and pathway genes. Nine ATF/BS control units and β -carotene (*Mcr1l*, *BTS1*, *Mcr1YB*), and *RiCCD1* CDS units (*RiCCD1* converts β -carotene to β -ionone) were assembled in Destination vector I, Acceptor vectors A, B, and C, respectively.”

8. Page 52 – Supplementary Note 3 – A reference to “Supplementary Fig. 5e,h; 2” is made, which seems like it should be “Figure 5e,h; 2”.

RESPONSE:

The text was corrected (page 71, line 1975 of the revised manuscript): “**Supplementary Fig. 5e,h; 2**” was changed to “**Fig. 5e, h; 2**”.

9. Pages 55-56 – Supplementary Note 8 – Various figure references seem to be mis-labeled.

RESPONSE:

The corrections were done. **Supplementary Figure 12** was changed to **Supplementary Figure 17** of the revised manuscript.

10. Page 57 – Supplementary Protocols – Authors switch from talking about molar ratios to concentration ratios within the first protocol... Was this intended to imply that the fragments generated from multiplex PCRs have heterogeneous/unpredictable molarity? If not, comprehensibility could be improved by sticking to a single descriptor, molarity.

RESPONSE: “Concentration ratio” was changed to “molar ratio”.

On a related note, do either of the “Multiplex-PCR-amplified ATFs,” and “their corresponding BS fragments” amplicons that the authors refer to correspond to POOLS of DNA fragments? The first paragraph of their Online Methods section specifies that “Amplified DNA parts were gel-purified prior to further use”... if they’re pools, what was the procedure to ensure that all fragment sizes were captured? Are primers designed to ensure that all amplicons are a similar length so that a single band can be excised? If not, are different length fragments excised+purified and then combined manually? The authors could certainly add some statements to the protocol to clarify these points – particularly in regards to which components represent pools and which do not. Related explanations under “Level 1” on Page 7 could perhaps also be improved... For example, was “five freely selected CDS units” meant to imply that a pool of the five CDS units were added in equal amounts together with the pool of 9 ATF/BS units and the linearized backbone?

RESPONSE:

We clarified this in the text by modifying the **Online Methods** as indicated in the following:

1. Page 29, lines 773 - 775: “Amplified DNA parts were gel-purified prior to further use, except when noted otherwise. Moreover, multiplex PCR-amplified fragments were not gel-purified (**Supplementary Protocols**).”

2. Pages 37 - 38, lines 1082 - 1085: “Coding sequences of ATFs were obtained by PCR using appropriate expression plasmids⁵ as templates and the respective forward (ATF-for) and reverse (ATF-rev) primers (see **Supplementary Protocols**). The corresponding binding sites (*JUB1 2X*, *JUB1 4X*, *ANAC102 4X*, *ATAF1 2X*, *RAV1 4X*, and *GRF7 4X*) fused upstream to the yeast minimal *CYC1* promoter were obtained by PCR using appropriate reporter plasmids⁷ as templates and the respective forward (BS-for) and reverse (BS-rev) primers (**Supplementary Protocols**).”

3. Page 38, lines 1094 - 1097: “For *JUB1*, three expression plasmids containing NLS-*JUB1*-*GAL4AD*, NLS-*DBDJUB1*-*GAL4AD*, and NLS-*JUB1*-*EDLLAD*-*EDLLAD* coding sequences were mixed in 1:1:1 molar ratio, and two reporter plasmids harboring two and four copies of the *JUB1* BS, respectively, were mixed in 1:1 molar ratio.”

4. Page 39, line 1134 - 1138: “Nine PCR-amplified ATF/BS fragments (primers *X0_for* and *Z0_rev*, PCR performed on the Entry vectors-nine ATF/BS) were mixed (see **Supplementary Protocols**). The *Mcr1l*, *BTS1*, *Mcr1YB*, and *RiCCD1* coding sequences and their downstream terminators and the promoters of the *E. coli* selection marker genes were PCR-amplified from Entry vectors using appropriate pairs of primers (**Supplementary Protocols**).”

5. Page 40, lines 1159 -1163: “Nine PCR-amplified ATF/BS fragments were mixed (see **Supplementary Protocols**). The *AtC4H:L5:AtATR2*, *PhCHI*, *HaCHS*, *At4CL-2*, and *AtPAL-2* coding sequences and their downstream terminators and the promoters of the *E. coli* selection marker genes were PCR-amplified from Destination vector II and Acceptor vectors E - H, respectively (**Supplementary Protocols**).”

6. Page 40, lines 1180 - 1181: “Therefore, PCR-amplified fragments contain nine different modules differing in their ATF/BS units (see **Supplementary Protocols**).”

References

1. Gibson, D.G. Enzymatic assembly of overlapping DNA fragments. *Methods Enzymol* **498**, 349-361 (2011).
2. Zhang, Y., Werling, U. & Edlmann, W. SLiCE: a novel bacterial cell extract-based DNA cloning method. *Nucleic Acids Res* **40**, e55 (2012).
3. Potapov, V. & Ong, J.L. Examining sources of error in PCR by single-molecule sequencing. *PLoS One* **12**, e0169774 (2017).
4. Werner, S., Engler, C., Weber, E., Gruetzner, R. & Marillonnet, S. Fast track assembly of multigene constructs using Golden Gate cloning and the MoClo system. *Bioeng Bugs* **3**, 38-43 (2012).
5. Kouprina, N. & Larionov, V. Transformation-associated recombination (TAR) cloning for genomics studies and synthetic biology. *Chromosoma* **125**, 621-632 (2016).
6. Mitchell, L.A. *et al.* Versatile genetic assembly system (VEGAS) to assemble pathways for expression in *S. cerevisiae*. *Nucleic Acids Res* **43**, 6620-6630 (2015).
7. Breslow, D.K. *et al.* A comprehensive strategy enabling high-resolution functional analysis of the yeast genome. *Nat Methods* **5**, 711-718 (2008).
8. Bao, Z. *et al.* Homology-integrated CRISPR-Cas (HI-CRISPR) system for one-step multigene disruption in *Saccharomyces cerevisiae*. *ACS Synth Biol* **4**, 585-594 (2015).
9. Karpinski J, *et al.* Directed evolution of a recombinase that excises the provirus of most HIV-1 primary isolates with high specificity. *Nat Biotechnol* **34**, 401-409 (2016).
10. Lopez, J. *et al.* Production of beta-ionone by combined expression of carotenogenic and plant CCD1 genes in *Saccharomyces cerevisiae*. *Microb Cell Fact* **14**, 84 (2015).
11. Scalcinati, G. *et al.* Combined metabolic engineering of precursor and co-factor supply to increase alpha-santalene production by *Saccharomyces cerevisiae*. *Microb Cell Fact* **11**, 117 (2012).
12. Hochrein, L. *et al.* AssemblX: a user-friendly toolkit for rapid and reliable multi-gene assemblies. *Nucleic Acids Res* **45**, e80 (2017).
13. Naseri, G. *et al.* Plant-derived transcription factors for orthologous regulation of gene expression in the yeast *Saccharomyces cerevisiae*. *ACS Synth Biol* **6**, 1742-1756 (2017).
14. Ding, M.Z. *et al.* Biosynthesis of Taxadiene in *Saccharomyces cerevisiae* : selection of geranylgeranyl diphosphate synthase directed by a computer-aided docking strategy. *PLoS One* **9**, e109348 (2014).
15. Chen, X. & Zhang, J. The genomic landscape of position effects on protein expression level and noise in yeast. *Cell Syst* **2**, 347-354 (2016).
16. Lian, J., Hamedirad, M., Hu, S. & Zhao, H. Combinatorial metabolic engineering using an orthogonal tri-functional CRISPR system. *Nat Commun* **8**, 1688 (2017).
17. Zabet, N.R. & Adryan, B. Estimating binding properties of transcription factors from genome-wide binding profiles. *Nucleic Acids Res* **43**, 84-94 (2015).

Reviewers' Comments:

Reviewer #1:

Remarks to the Author:

The authors have responded appropriately to all my concerns and comments.

Reviewer #3:

Remarks to the Author:

In general, the authors did a nice job of addressing my concerns, particularly with their experimental revisions. The newly added Supplementary Figures (2, 13, and 14) do well to showcase the advantages and inducibility of their ATF/BS architecture, but also shed light on some of COMPASS' limitations as a whole. This is especially appropriate given that they've produced a methodological study; the strengths and weaknesses should both be presented (and in most cases it's clear that the weaknesses reflect universal challenges encountered with *S. cerevisiae*).

Their implementation of ATF/BS units is still, in my opinion, the most impactful aspect of the work. The fact that they have almost entirely eliminated the need for restriction enzymes is perhaps also to be applauded, because overlap-based methods and PCR are intrinsically more flexible/modular for the purpose of combinatorial assembly (and high-fidelity polymerases such as Phusion, when used properly, ensure sufficient sequence preservation and cost-effectiveness for typical assembly purposes). The editors should exercise their discretion as to whether or not these or other aspects of the work constitute sufficient impact to justify publication in Nature Communications. My remaining specific comments are itemized below (mostly minor issues). In addition, the authors are encouraged to correct misspellings and instances of awkward syntax or grammar that were either not addressed initially or were inadvertently introduced with their revisions. Again, however, I have not attempted to itemize the latter points.

Comments:

1. The authors adjusted their Fig. 6c legend in response to one of my previous comments ("Other Comments" #4), in an attempt to clarify their assay design. Unfortunately, there is now an apparent incongruency between that legend (line 509) and the relevant section of the methods, which states that single colonies were initially inoculated into YPDA (lines 1452-1453). I presume those are the authors' "precultures"; do the precultures in fact utilize SC medium or YPDA? Also, that section within the methods currently only states that the precultures were used to inoculate "main cultures" in inducing media – were the same precultures also used to inoculate "main cultures" in non-inducing media? Finally, use of the terminology "subcultures" is strongly preferred to "main cultures".

2. The authors incorporated new yEGFP data for the constitutive TDH3 promoter into Supplementary Figure 1. Given the newly added results of Supplementary Figure 2, which seem to indicate that integration sites can have a crucial influence on overall expression, the authors should explicitly state in the legend or graphic of Supplementary Figure 1 where the yEGFP cassettes were integrated for those experiments (regardless of the fact that they provide a reference to their previous work). For example, the *ura3-52* locus?

3. The authors added additional yEGFP data for each of the ATF/BS units as their new Supplementary Figure 2. Despite the apparent location-dependent effects observed overall, this data seems to indicate that the relative activity of ATF/BS outputs generally (although not always) falls in line with the hierarchy they've presented in Supplementary Figure 1. Given that the TDH3 promoter falls fairly low in the activity hierarchy of Supplementary Figure 1, and that expression from the XII-5 locus was generally reduced across the board relative to the *ura3-52* locus, it would have made sense to include measurements with the TDH3 promoter integrated at the XII-5 locus in Supplementary Figure 2. This would have also provided immediate validation that, under THOSE

assay conditions, the inducibility we're observing for each ATF/BS promoter is a specific effect of IPTG (which should have no effect on the constitutive TDH3 promoter). Regardless, assuming no outstanding experimental flaws regarding comment #1, the experiments they've added do well to showcase the variable expression and inducibility that might be expected from a bona-fide COMPASS assembly.

4. Related to my comments #2 & #3 above, I am generally still critical of the authors' decision to use the TDH3 promoter as their "strong" positive control throughout the manuscript, as they've CLEARLY implemented much stronger promoters that could have been tested in their control assemblies. They did however seem to confirm with their newly added Supplementary Figure 13 that the TDH3 promoters are constitutive (albeit indirectly, since multiple instances of the promoter are present in that assembly). At the very least, the authors should adjust their textual descriptions concerning this promoter throughout the manuscript to emphasize only that it's constitutive (rather than strong). It seems misleading—if not merely confusing—to keep referring to TDH3 as a "strong" promoter, given their data in Supplementary Figure 1.

The following lines have instances of "strong" that should be adjusted accordingly:
Line 136, Line 338, Line 1154, Line 1931, Line 1967, Line 2001, Line 2089, Line 2121

5. Page 16 – Lines 398-413 – The authors added Supplementary Figure 14 in response to one of my previous comments, along with some relevant textual explanations in the results section. I am in general agreement with the rationale they've provided, for example: "Moreover, we previously showed that a higher transcriptional output is often obtained by increasing the copy number of a binding site of a plant regulator, suggesting that the abundance of the ATF expressed from an IPTG inducible promoter is likely sufficient to target more than a single copy of the binding site." However, I found it confusing that they begin the next sentence (which introduces Supplementary Figure 14) with "Furthermore", as if to imply that the results of Supplementary Figure 14 were completely consistent with the rationale they've just proposed above. Rather, the new results in fact demonstrate some mild deviation from expectations. Accordingly, I feel that they should change "Furthermore" to "However" (or equivalent), on line 406.

6. Page 5 – Lines 122-123 – The authors adjusted this sentence in the legend in response to one of my previous comments but failed to explicitly clarify that the integration step is carried out in yeast (as opposed to all the other Level 1 manipulations which use E. coli), and that it's optional (if a COMPASS user prefers to follow only the single-locus integration pathway at Level 2, they still need to construct the Level 1 vectors but can skip the "multi-locus integration" steps). The references to Fig. 1 that they added on page 9 and page 11 were appreciated, but did nothing to address this particular point of confusion for me. Therefore, given that it's a key figure and presented so early in the paper, the legend would immediately benefit from more explicit wording: e.g., "Thereafter, the ten groups of ATF/BS-CDS modules of Level 1 may be integrated into ten defined loci of the yeast genome."

7. The authors successfully adjusted some terminology to "isolates" in the final section of their results ("Proof of concept: Co-biosynthesis of β -ionone and biosensor-responsive naringenin"), and this has substantially reduced the writing's ambiguity. With that said, two instances of the original terminology remain unchanged ("40% of the library members" on line 477; and, "15% of the library strains" on lines 479-480), and the authors should consider adjusting those as well.

Responses to REVIEWERS' COMMENTS:

Reviewer #1 (Remarks to the Author):

The authors have responded appropriately to all my concerns and comments.

Reviewer #3 (Remarks to the Author):

In general, the authors did a nice job of addressing my concerns, particularly with their experimental revisions. The newly added Supplementary Figures (2, 13, and 14) do well to showcase the advantages and inducibility of their ATF/BS architecture, but also shed light on some of COMPASS' limitations as a whole. This is especially appropriate given that they've produced a methodological study; the strengths and weaknesses should both be presented (and in most cases it's clear that the weaknesses reflect universal challenges encountered with *S. cerevisiae*).

Their implementation of ATF/BS units is still, in my opinion, the most impactful aspect of the work. The fact that they have almost entirely eliminated the need for restriction enzymes is perhaps also to be applauded, because overlap-based methods and PCR are intrinsically more flexible/modular for the purpose of combinatorial assembly (and high-fidelity polymerases such as Phusion, when used properly, ensure sufficient sequence preservation and cost-effectiveness for typical assembly purposes). The editors should exercise their discretion as to whether or not these or other aspects of the work constitute sufficient impact to justify publication in Nature Communications. My remaining specific comments are itemized below (mostly minor issues). In addition, the authors are encouraged to correct misspellings and instances of awkward syntax or grammar that were either not addressed initially or were inadvertently introduced with their revisions. Again, however, I have not attempted to itemize the latter points.

Comments:

1. The authors adjusted their Fig. 6c legend in response to one of my previous comments ("Other Comments" #4), in an attempt to clarify their assay design. Unfortunately, there is now an apparent incongruity between that legend (line 509) and the relevant section of the methods, which states that single colonies were initially inoculated into YPDA (lines 1452-1453). I presume those are the authors' "precultures"; do the precultures in fact utilize SC medium or YPDA? Also, that section within the methods currently only states that the precultures were used to inoculate "main cultures" in inducing media – were the same precultures also used to inoculate "main cultures" in non-inducing media? Finally, use of the terminology "subcultures" is strongly preferred to "main cultures".

RESPONSE:

The reviewer is right. We always used SC medium for the precultures. The text in the Methods section has now been corrected: "To quantify the yEGFP fluorescence output in the absence of plant-derived ATFs, single colonies of FdeR-based reporter strains were inoculated into 500 μ l non-inducing SC medium with appropriate selection markers in 48-well deep-well plates."

We changed "main cultures" to "subcultures".

2. The authors incorporated new yEGFP data for the constitutive TDH3 promoter into Supplementary Figure 1. Given the newly added results of Supplementary Figure 2, which seem to indicate that integration sites can have a crucial influence on overall expression, the authors should explicitly state in the legend or graphic of Supplementary Figure 1 where the yEGFP cassettes were integrated for those experiments (regardless of the fact that they provide a reference to their previous work). For example, the *ura3-52* locus?

RESPONSE:

We added the information about the locus (*ura3-52*) to the legend of Supplementary Figure 1.

We also clarified this in the main text (Results): "The ATFs span an expressional activity ranging from ~0.4- to ~5-fold of that observed for the strong yeast *TDH3* promoter³², whereby the regulators were all integrated into the *ura3-52* locus of yeast⁴ (Supplementary Fig. 1)."

3. The authors added additional yEGFP data for each of the ATF/BS units as their new Supplementary Figure 2. Despite the apparent location-dependent effects observed overall, this data seems to indicate that the relative activity of ATF/BS outputs generally (although not always) falls in line with the hierarchy they've presented in Supplementary Figure 1. Given that the TDH3 promoter falls fairly low

in the activity hierarchy of Supplementary Figure 1, and that expression from the *XII-5* locus was generally reduced across the board relative to the *ura3-52* locus, it would have made sense to include measurements with the *TDH3* promoter integrated at the *XII-5* locus in Supplementary Figure 2. This would have also provided immediate validation that, under THOSE assay conditions, the inducibility we're observing for each ATF/BS promoter is a specific effect of IPTG (which should have no effect on the constitutive *TDH3* promoter). Regardless, assuming no outstanding experimental flaws regarding comment #1, the experiments they've added do well to showcase the variable expression and inducibility that might be expected from a bona-fide COMPASS assembly.

RESPONSE:

We constructed a control strain to study the position effect of the *XII-5* locus on fluorescence output of the yeast *TDH3* promoter by performing the following experiment: We constructed Acceptor vector A containing the *TDH3* promoter upstream of *yEGFP*. The Pro_{TDH3} -EGFP module was integrated into locus *XII-5*, and NLS-ATAF1-GAL4AD/2X-McrtI and NLS-DBD_{IIIR1}-GAL4AD/2X-McrtYB modules (Figure 5h) were integrated into the *X-3* and *XI-3* locus, respectively, to generate yeast strain YEGFP0; see Methods (page 32, lines 1032 – 1042 in “No Markup”-setting in track change) and Supplementary Table 5. *yEGFP* output was tested in the absence and presence of inducer. The results are now presented in Supplementary Figure 2b and c, Results (page 5, lines 127 – 129 in “No Markup”-setting in track change), Discussion (pages 14, lines 450 – 455 in “No Markup”-setting in track change), and Supplementary Note 1. Our results demonstrate that the transcriptional output of the plant regulators at locus *XII-5* is 0.05- to 4.3-fold greater than that of the yeast *TDH3* promoter. The data also confirm that the *TDH3* promoter is constitutive, as it resulted in similar *yEGFP* output in YPDA inducing medium (20 mM IPTG, 2% (w/v) galactose; *ura3-52* locus: 819 ± 29 AU; *XII-5* locus: 273.35 ± 17.24 AU) and non-inducing medium (2% (w/v) glucose; *ura3-52* locus: 1030.07 ± 28.02 AU; *XII-5* locus: 267.38 ± 12.27 AU).

4. Related to my comments #2 & #3 above, I am generally still critical of the authors' decision to use the *TDH3* promoter as their “strong” positive control throughout the manuscript, as they've CLEARLY implemented much stronger promoters that could have been tested in their control assemblies. They did however seem to confirm with their newly added Supplementary Figure 13 that the *TDH3* promoters are constitutive (albeit indirectly, since multiple instances of the promoter are present in that assembly). At the very least, the authors should adjust their textual descriptions concerning this promoter throughout the manuscript to emphasize only that it's constitutive (rather than strong). It seems misleading—if not merely confusing—to keep referring to *TDH3* as a “strong” promoter, given their data in Supplementary Figure 1. The following lines have instances of “strong” that should be adjusted accordingly:

Line 136, Line 338, Line 1154, Line 1931, Line 1967, Line 2001, Line 2089, Line 2121

RESPONSE:

We now changed “strong yeast *TDH3* promoter” to “constitutive yeast *TDH3* promoter” throughout.

5. Page 16 – Lines 398-413 – The authors added Supplementary Figure 14 in response to one of my previous comments, along with some relevant textual explanations in the results section. I am in general agreement with the rationale they've provided, for example: “Moreover, we previously showed that a higher transcriptional output is often obtained by increasing the copy number of a binding site of a plant regulator, suggesting that the abundance of the ATF expressed from an IPTG inducible promoter is likely sufficient to target more than a single copy of the binding site.” However, I found it confusing that they begin the next sentence (which introduces Supplementary Figure 14) with “Furthermore”, as if to imply that the results of Supplementary Figure 14 were completely consistent with the rationale they've just proposed above. Rather, the new results in fact demonstrate some mild deviation from expectations. Accordingly, I feel that they should change “Furthermore” to “However” (or equivalent), on line 406.

RESPONSE:

We changed “Furthermore” to “However”. (page 10, line 321 in “No Markup”-setting in track change).

6. Page 5 – Lines 122-123 – The authors adjusted this sentence in the legend in response to one of my previous comments but failed to explicitly clarify that the integration step is carried out in yeast (as opposed to all the other Level 1 manipulations which use *E. coli*), and that it's optional (if a COMPASS user prefers to follow only the single-locus integration pathway at Level 2, they still need to construct the Level 1 vectors but can skip the “multi-locus integration” steps). The references to Fig. 1 that they added on page 9 and page 11 were appreciated, but did nothing to address this particular point of confusion for me. Therefore, given that it's a key figure and presented so early in the paper, the

legend would immediately benefit from more explicit wording: e.g., “Thereafter, the ten groups of ATF/BS-CDS modules of Level 1 may be integrated into ten defined loci of the yeast genome.”

RESPONSE:

The reviewer is right.

In legend to Figure 1, we changed “Thereafter, the ten groups of ATF/BS-CDS modules of Level 1 are integrated into ten defined loci of the genome.” to “Thereafter, the ten groups of ATF/BS-CDS modules of Level 1 may be integrated into ten defined loci of the yeast genome.”

7. The authors successfully adjusted some terminology to “isolates” in the final section of their results (“Proof of concept: Co-biosynthesis of β -ionone and biosensor-responsive naringenin”), and this has substantially reduced the writing’s ambiguity. With that said, two instances of the original terminology remain unchanged (“40% of the library members” on line 477; and, “15% of the library strains” on lines 479-480), and the authors should consider adjusting those as well.

RESPONSE:

We corrected the text:

Page 12, line 371 (“No Markup”-setting in track change): “40% of the library members” is now changed to “40% of the library isolates”.

Page 12, line 373 (“No Markup”-setting in track change): “15% of the library strains” is now changed to “15% of the library isolates”.

** See Nature Research's author and referees' website at www.nature.com/authors for information about policies, services and author benefits